

# Evaluating modelled tropospheric columns of CH₄, CO and O₃ in the Arctic using ground-based FTIR measurements

Victoria A Flood[1], Kimberly Strong[1], Cynthia H. Whaley[2], Kaley A. Walker[1], Thomas Blumenstock[3],
James W. Hannigan[4], Johan Mellqvist[5], Justus Notholt[6], Mathias Palm[6], Amelie N. Röhling[3], Stephen
Arnold[7], Stephen Beagley[8], Rong-You Chien[9], Jesper Christensen[10], Makoto Deushi[11], Srdjan
Dobricic[12], Xinyi Dong[9], Joshua S. Fu[9], Michael Gauss[13], Wanmin Gong[8], Joakim Langner[14], Kathy S.
Law[15], Louis Marelle[15], Tatsuo Onishi[15], Naga Oshima[11], David A. Plummer[2], Luca Pozzoli[16,12], Jean-
Christophe Raut[15], Manu A. Thomas[14], Svetlana Tsyro[13], and Steven Turnock[17, 7]

[1]Department of Physics, University of Toronto, Toronto, ON, Canada
[2]Canadian Centre for Climate Modelling and Analysis, Environment and Climate Change Canada, Victoria, BC, Canada
[3]Karlsruhe Institute of Technology, Institute of Meteorology and Climate Research (IMK-ASF), Karlsruhe, Germany
[4]Atmospheric Chemistry, Observations & Modeling, National Center for Atmospheric Research, Boulder, CO, USA
[5]Earth and Space Sciences, Chalmers University of Technology, Gothenburg, Sweden
[6]Institute of Environmental Physics, University of Bremen, Bremen, Germany
[7]Institute for Climate and Atmospheric Science, School of Earth and Environment, University of Leeds, Leeds, United Kingdom
[8]Air Quality Research Division, Environment and Climate Change Canada, Toronto, ON, Canada
[9]University of Tennessee, Knoxville, Tennessee, United States
[10]Department of Environmental Science/Interdisciplinary Centre for Climate Change, Aarhus University, Frederiksborgvej 399, 4000, Roskilde, Denmark
[11]Meteorological Research Institute, Japan Meteorological Agency, Tsukuba, Japan
[12]European Commission, Joint Research Centre, Ispra, Italy
[13]Division for Climate Modelling and Air Pollution, Norwegian Meteorological Institute, Oslo, Norway
[14]Swedish Meteorological and Hydrological Institute, Norrköping, Sweden
[15]Sorbonne Université, UVSQ, CNRS, LATMOS, Paris, France
[16]FINCONS SPA, Via Torri Bianche 10, 20871 Vimercate, Italy
[17]Met Office Hadley Centre, Exeter, UK

*Correspondence to*: Victoria A. Flood (v.flood@utoronto.com)

**Abstract.** Both measurements and modelling of air pollution in the Arctic are difficult. Yet with the Arctic warming at nearly four times the global average rate, and changing emissions in and near the region, it is important to understand Arctic atmospheric composition and how it is changing. This study examines the simulations of atmospheric concentrations of methane, carbon monoxide and ozone in the Arctic by 11 models. Evaluations are performed using data from five high-latitude ground-based Fourier transform infrared (FTIR) spectrometers in the Network for the Detection of Atmospheric Composition Change (NDACC). Mixing ratios of trace gases are modelled at three-hourly intervals by CESM, CMAM, DEHM, EMEP MSC-W, GEM-MACH, GEOS-Chem, MATCH, MATCH-SALSA, MRI-ESM2, UKESM1 and WRF-Chem for the years 2008, 2009, 2014, and 2015. The comparisons focus on the troposphere (0-7 km partial columns) at Eureka, Canada; Thule, Greenland; Ny Ålesund, Norway; Kiruna, Sweden; and Harestua, Norway. Overall, the models are biased



low in the tropospheric column, on average by -9.6% for CH$_4$, -21% for CO and -18% for O$_3$. Results for CH$_4$ are relatively consistent across the four years, whereas CO has a maximum negative bias in the spring and minimum in the summer, and O$_3$ has a maximum difference centred around the summer. The average differences for the models are within the FTIR uncertainties for approximately 15% of the model-location comparisons.

## 1 Introduction

Short Lived Climate Forcers (SLCFs) are a group of greenhouse gases and air pollutants with lifetimes less than two decades (IPCC, 2021). These include methane (CH$_4$), ozone (O$_3$), black carbon, halocarbons, sulfate, nitrate, and organic aerosols. The Intergovernmental Panel on Climate Change (IPCC) reports that in addition to radiative forcing, SLCFs have been found to have negative impacts on air quality, ecosystems, and human health.  Due to their relatively short lifetimes, SLCFs are generally reflective of emission rates, meaning that mitigation can result in near-term impacts. Furthermore, in most cases

anthropogenic sources outweigh natural emissions and for SLCFs with shorter atmospheric lifetimes, less mixing occurs, causing most of the pollutants to remain predominantly localized. Understanding the impact of SLCFs on the future climate will aid in policies and mitigation strategies to stay on track with the Paris Accord and its subsequent amendments. Reductions of SLCFs can be particularly beneficial in the Arctic because models have demonstrated a strong climate response in this region to local and remote forcing by SLCFs (Stohl et al., 2015).

The Arctic Monitoring and Assessment Programme (AMAP) was created by the Arctic Council to provide science-based analysis of Arctic pollution and climate change. AMAP has provided reports on SLCF impacts on the Arctic dating back to 2008. The 2021 AMAP SLCF Assessment Report assesses the impacts of black carbon, CH$_4$, O$_3$ and sulfate aerosols on the air quality, climate and human health in the Arctic region (AMAP, 2021). A key difference from previous AMAP reports is the emphasis on air quality and human health. In addition to these SLCFs, the analysis includes SLCF precursor

gases carbon monoxide (CO), nitric oxide (NO) and nitrogen dioxide (NO$_2$). The report compares the output from 18 models with various historical measurements, including satellite, aircraft, ship, and in situ datasets. A prominent theme in this report is the severity of change happening in the Arctic. This includes the amplification of the pace of change in physical drivers such as temperature and snow cover, and the frequency of extreme events, such as wildfires and incidents of rapid sea-ice loss. These factors contribute to ecosystem disruption, directly affecting local Arctic communities, in addition to having

global repercussions. SLCF reductions are motivated by the near-term (20-30 years) benefits, and by the goal of slowing the warming of the Arctic climate, which results in more wildfires and permafrost melt, and in turn, an increase in SLCF emissions and precursor gases (AMAP, 2021). The projections in this report provide guidance, objectives, and cautions for potential reduction implementation scenarios (AMAP, 2021).

         Using measurements to evaluate model simulations of the Arctic is important because the latter are used to project

future changes in the Arctic, a region that is sensitive to climate change, warming at a rate three to four times the global average (Bush and Lemmen, 2019; NOAA, 2020; AMAP, 2021; IPCC, 2021; Rantanen et al., 2022). However,



measurements in the Arctic are difficult due to the harsh environment, remote locations, and high operating costs. All of these factors lead to a scarcity of monitoring stations and a limited representation of atmospheric vertical information; as such, utilization of all available data sources is valuable in aiding model development.

The Network for Detection of Atmospheric Composition Change (NDACC) has over 70 stations around the globe, collecting high-quality atmospheric composition measurements with ground-based, remote-sensing instruments (Kurylo and Soloman, 1990; De Mazière et al., 2018). The network's objective is to create a long-term database for various studies such as atmospheric trends, assessing links between climate, air quality and composition, and as a resource for other atmospheric investigations such as satellite validation and model development. Atmospheric vertical profiles and trace gas columns are

retrieved from high-resolution Fourier transform infrared (FTIR) spectrometers that record solar spectra featuring characteristic atmospheric absorption lines. Five of the 28 NDACC FTIR stations are located at latitudes north of 60°N, all of which are included in this study: Eureka, Canada; Ny Ålesund, Norway; Thule, Greenland; Kiruna, Sweden; and Harestua, Norway.

    This project examines simulations from 11 models that were run for the 2021 AMAP SLCF Assessment Report, to

assess the agreement between modelled trace gas concentrations and ground-based retrievals from these high-latitude FTIR spectrometers. Specifically, this paper presents comparisons of $CH_4$, CO and $O_3$ partial columns (from 0-7 km) for the years 2008, 2009, 2014, and 2015. The models examined are chemical transport and climate models: CESM, CMAM, DEHM, EMEP MSC-W, GEM-MACH, GEOS-CHEM, MATCH, MATCH-SALSA, MRI-ESM2, UKESM1 and WRF-CHEM. The objective is to utilize the high-quality, long-term Arctic FTIR datasets to assess how well the models perform. This study

builds upon the model-measurement comparisons presented in the 2021 AMAP SLCF Assessment Report using an additional Arctic dataset that was not included in the original report. The remainder of this paper is organized as follows: Sect. 2 provides a description of the datasets used, Sect. 3 describes the analysis methodology, Sect. 4 examines the results and compares them with similar studies, and Sect. 5 presents the summary and conclusions.

## 2 Datasets

### 2.1 FTIR spectroscopy and retrievals

The FTIR measurement sites included in this study are summarized in Table 1. These instruments require sunlight and a clear sight to the sun to make measurements, and so the high-latitude datasets are limited to the sunlit portion of the year at each location. To ensure high data quality and consistency between sites, NDACC has several specialized instrument and theme groups; the instruments used here are part of the Infrared Working Group (IRWG). The ten standard gases reported by

sites participating in the IRWG are $C_2H_6$, $CH_4$, CO, $ClONO_2$, HCl, HCN, HF, $HNO_3$, $N_2O$, and $O_3$, while several other gases are retrieved as research data products, including $C_2H_2$, $CH_3OH$, $H_2CO$, HCOOH and OCS. The FTIR measurements cycle through a series of optical filters covering different spectral regions between approximately 650 and 4500 cm$^{-1}$ for the retrieval of multiple atmospheric gases. Atmospheric trace gas profiles and columns are retrieved with the SFIT4 algorithm,



using optimal estimation to iteratively adjust an a priori profile to match a modelled spectrum to the measured spectrum

within a defined convergence criterion (Rodgers, 2000; IRWG, 2020). The a priori information for the modelled spectra is provided by 40-year-average profiles from the Whole Atmosphere Community Climate Model (WACCM) (Marsh et al., 2013), with spectroscopic absorption parameters from the HITRAN 2008 line-list (Rothman et al., 2009) and pressure and temperature data from the U.S. National Centers for Environmental Prediction (NCEP) (Kalnay et al., 1996). All sites included in this paper use SFIT4, except Kiruna, which uses a comparable program called PROFFIT (Hase et. al, 2004).

Primary references and further details of the sites are presented in Table 1.

**Table 1: Summary of NDACC FTIR sites used in this study.**

| Site | Location | Key References | Operations | Colour Key |
|---|---|---|---|---|
| Eureka, Canada | 80.05°N, 86.42°W 610 masl | Batchelor et. al. (2009) | Late February to Mid-October / Since 2006 | |
| Ny Ålesund, Norway | 78.92°N, 11.93°E 15 masl | Notholt et al. (1997a,b); Notholt et al. (2000) | Mid-March to September / Since 1992 | |
| Thule, Greenland | 76.53°N, 68.74°W 225 masl | Hannigan et al. (2009) | March to October / Since 1999 | |
| Kiruna, Sweden | 67.84°N, 20.41°E 419 masl | Blumenstock et al. (1997, 2009) | Mid-January to November / Since 1996 | |
| Harestua, Norway | 60.2°N, 10.8°E 596 masl | Galle et al. (1999) | All Year / Since 1994 | |

The NDACC FTIR data files include volume mixing ratio (VMR) in parts per million (ppm) and total columns and partial

columns in molecules per centimetre squared (molec/cm$^2$). Other variables include altitude, date/time, pressure, a priori vertical profile, averaging kernel matrix and retrieval uncertainties, both systematic and random. The random uncertainties are determined from the temperature, solar zenith angle, and measurement noise from the signal-to-noise ratio. Systematic uncertainties are determined from temperature and line parameters such as line strength and width.

       The averaging kernel matrix represents the relationship between the retrieved state and the true atmospheric state at

each altitude layer and the sensitivity of a retrieval is calculated by taking the sum of the rows of the averaging kernel. This indicates how much of the information is coming from the a priori profile, and how much comes from the measurement itself (Rodgers, 2003; Vigouroux et al., 2009). The mean column averaging kernel for CH$_4$, CO and O$_3$ for 2008, 2009, 2014 and 2015, are shown in Fig. 1. The plot is focused on the 0-7 km column, where the column averaging kernels for all three species appear smooth around 1.0. By altitude, the sensitivity of each species is >0.5 in the partial column examined,





meaning over half of the retrieved profile information comes from the measurement. The degrees of freedom for signal (DOFS) is calculated by taking the trace of the averaging kernel; this indicates the number of independent pieces of information coming from each retrieval. The random and systematic partial column errors are calculated using the error covariance matrices, following the method outlined in Vigouroux et al. (2009). The square root of the associated error is taken, and this is scaled to a percent error using the corresponding partial column sum. The mean systematic and random

percent errors are added in quadrature to get the overall mean percent error for the species. The mean DOFS and mean percent error of the 0-7 km partial columns of $CH_4$, CO, and $O_3$ for 2008, 2009, 2014, and 2015, for each station are listed in Table 2.

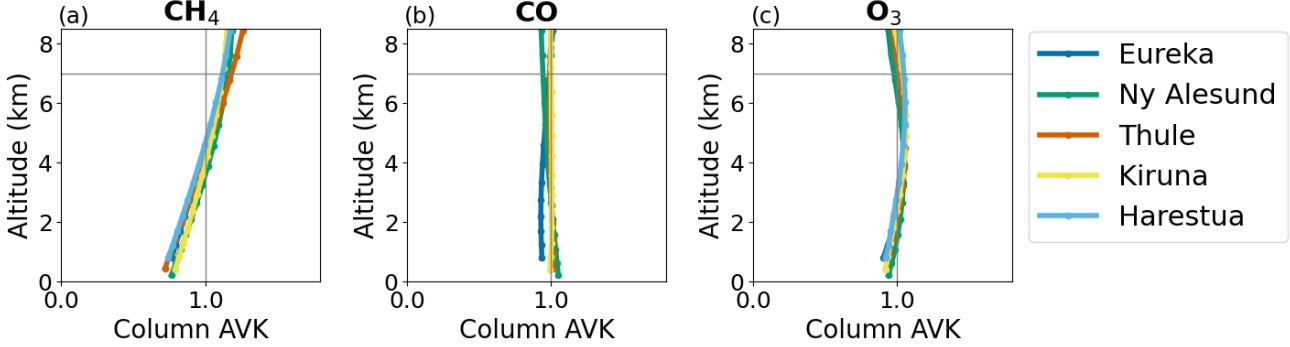

**Figure 1: Mean column averaging kernels for (a) $CH_4$, (b) CO and (c) $O_3$. Means are for 2008, 2009, 2014, and 2015 for all five FTIR sites except Harestua (no 2008 data).**

**Table 2: Summary of FTIR measurement statistics.**

| Site | Number of Measurements (2008, 2009, 2014, 2015) | | | Mean DOFS (0-7 km) | | | Mean Percent Error (0-7 km) | | |
|---|---|---|---|---|---|---|---|---|---|
| | $CH_4$ | CO | $O_3$ | $CH_4$ | CO | $O_3$ | $CH_4$ | CO | $O_3$ |
| Eureka | 754 | 736 | 684 | 0.84 | 1.1 | 0.80 | 4.6 | 3.9 | 8.2 |
| Ny Ålesund | 205 | 128 | 121 | 0.81 | 1.3 | 0.79 | 11.5 | 7.7 | 4.9 |
| Thule | 406 | 459 | 474 | 0.78 | 1.6 | 1.2 | 5.7 | 5.4 | 3.9 |
| Kiruna | 397 | 299 | 322 | 0.96 | 1.6 | 0.86 | 3.6 | 6.4 | 7.2 |
| Harestua | 193 (no 2008) | No CO | 169 (no 2008) | 0.81 | N/A | 1.12 | 5.2 | N/A | 4.1 |





**2.2 Atmospheric models**

The models used in this study provide three-dimensional VMR fields on 3-hour intervals for 2008, 2009, 2014, and 2015.
Note that not every model has provided all three gases. This set of models is a mix of Earth system models, chemical
transport models, global transport models, and chemistry climate models. The models all used the same set of anthropogenic
emissions from ECLIPSE v6b (Evaluating the Climate and Air Quality Impacts of Short-Lived Pollutants) by the IIASA

GAINS (International Institute for Applied Systems Analysis – Greenhouse gas – Air pollution Interactions and Synergies)
model (Amann et al., 2011; Klimont et al., 2017; Höglund-Isaksson et al., 2020). However, the models used a variety of
biogenic and volcanic emissions, different levels of tropospheric gas-phase chemistry complexity, and only some models
simulate the stratosphere (Whaley et al., 2022).  Nine of the 11 models examined use the Global Fire Emissions Database
(GFED, van der Werf et al., 2017) or GFED-based (CMIP6) forest fire emissions, and nine of the 11 exclusively use

ECLIPSEv6b for agricultural waste burning. A summary of the models is presented in Table 3. For a full description of the
models, see Appendix A of Whaley et al. (2022) and the references in Table 3.

**Table 3: Summary of models used in this study.**

| Model | 3-Hourly Outputs | Primary Reference | Colour Key |
|---|---|---|---|
| CESM<br>*Community Earth System Model* | $CO$, $O_3$ | Liu et al. (2016); Danabasoglu et al. (2020) | |
| CMAM<br>*Canadian Middle Atmosphere Model* | $CH_4$, $CO$, $O_3$ | Jonsson et al. (2004); Scinocca et al. (2008) | |
| DEHM<br>*Danish Eulerian Hemispheric Model* | $O_3$ | Christensen (1997); Brandt et al. (2012); Massling et al. (2015) | |
| EMEP MSC-W<br>*European Monitoring and Evaluation System-Meteorological Synthesizing Center - West* | $CO$, $O_3$ | Simpson et al. (2012, 2019) | |
| GEM-MACH<br>*Global Environmental Multiscale Model - Modelling Air Quality and Chemistry* | $CO$, $O_3$ (only 2015) | Gong et al. (2015); Makar et al. (2015a,b); Moran et al. (2018) | |
| GEOS-CHEM<br>*Goddard Earth Observing System - Chemistry* | $CH_4$, $CO$, $O_3$ | Bey et al. (2001) | |
| MATCH<br>*Multi-Scale Atmospheric Transport Chemistry* | $CO$, $O_3$ | Robertson et al. (1999) | |



| | | | |
|---|---|---|---|
| MATCH-SALSA<br>*Multi-Scale Atmospheric Transport Chemistry - Sectional Aerosol Module for Large Scale Applications* | CO, O$_3$ | Robertson et al. (1999); Andersson et al. (2007); Kokkola et al. (2008) | ■ |
| MRI-ESM2<br>*Meteorological Research Institute - Earth System Model Version 2* | CH$_4$, CO, O$_3$ | Kawai et al. (2019); Yukimoto et al. (2019); Oshima et al. (2020) | ■ |
| UKESM1<br>*U.K. Earth System Model Version 1* | O$_3$ | Kuhlbrodt et al. (2018); Williams et al. (2018); Sellar et al. (2019) | ■ |
| WRF-CHEM<br>*Weather Research and Forecasting Model with Chemistry* | CO, O$_3$ (only 2014 / 2015) | Marelle et al. (2017, 2018) | ■ |

## 3 Methods

The FTIR data are publicly available on the NDACC data repository [https://www-air.larc.nasa.gov/missions/ndacc/data.html], and the model data are available at http://crd-data-donnees-rdc.ec.gc.ca/CCCMA/products/AMAP/. As mentioned, the models provided 3-hourly VMRs on model-specific pressure levels and latitude/longitude grids. The process of aligning the model output to FTIR data is described by the flowchart in Fig. 2.






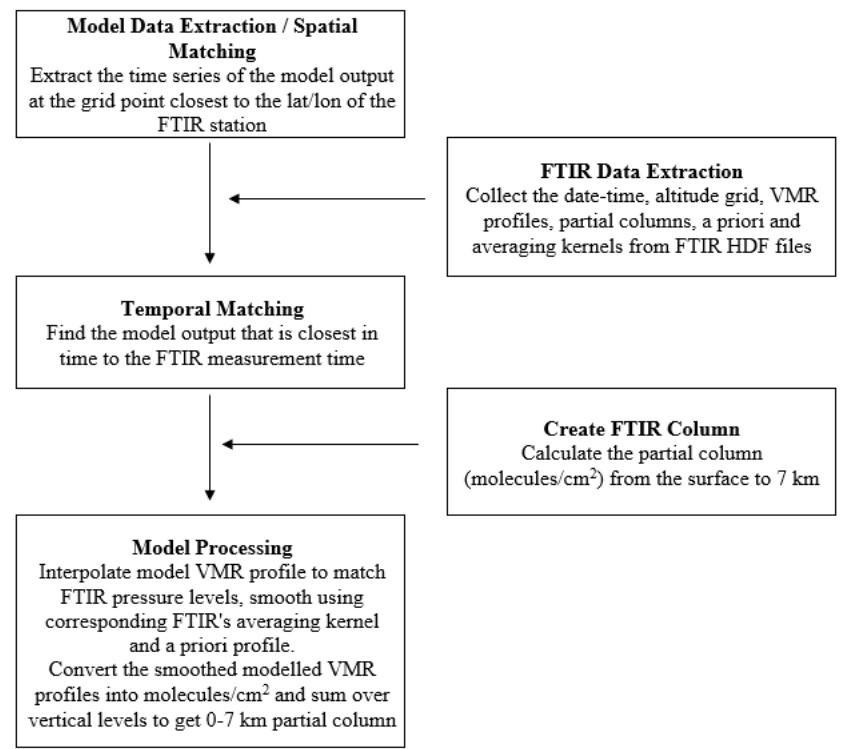

**Figure 2: Flow chart depicting the process of matching model output to FTIR data.**

This procedure modifies the model output to correspond to an FTIR measurement, making the resulting partial
columns equivalent for further comparison. The FTIR measurements are matched with the model measurement closest in
time (±<1.5 hours). If more than one FTIR measurement coincides with a model output (i.e., multiple measurements are
within 1.5 hours of the same model time), the FTIR measurements are averaged. After the model outputs are matched to the
FTIR measurements, they are interpolated onto the pressure grid of the FTIR profile. Then, the model VMR profile is
smoothed using the respective FTIR measurement's averaging kernel and a priori profile. The purpose of smoothing the
model data with the FTIR averaging kernel is to adjust the model to the vertical sensitivity of the FTIR measurement
(Rodgers and Connor, 2003). The calculation for the smoothing is shown in Eq. 1, where $\boldsymbol{x_a}$ is the FTIR a priori VMR
vertical profile, $\mathbf{A}$ is the column averaging kernel matrix from the corresponding FTIR measurement, and $\boldsymbol{x_{model}}$ is the
modelled VMR vertical profile:

$$\boldsymbol{x_{smooth}} = \boldsymbol{x_a} + \boldsymbol{A} \times [\boldsymbol{x_{model}} - \boldsymbol{x_a}]. \tag{1}$$


The model VMR profile is then transformed to a layer profile in units of molecules per centimeter squared using the ratio
between the VMR and molecules per centimeter squared in the retrieved FTIR profile as the conversion factor. At this point,





the model output has the same altitude grid and units as the FTIR, which allows for partial columns to be summed. Partial columns from 0-7 km were calculated given the focus on SLCFs in the troposphere. Note that "0 km" is used as proxy for
the minimum altitude, but this varies, based on location, with the altitude of each instrument listed in Table 1.

To compare the model and FTIR partial columns, a percent difference is calculated, as defined by Eq. 2, where $C_m$ and $C_F$ are the 0-7 km partial columns for the model and FTIR, respectively:

$$Percent\ Difference = \left(\frac{C_m - C_F}{C_F}\right) \times 100 \ . \tag{2}$$

The correlation coefficient, $R^2$, is calculated based on the line of best fit of the model results versus FTIR measurements
using all the available data points. The normalized root mean percent error (NRMSE) is presented for each model and location, given by Eq. 3 (Kärnä and Baptista, 2016). The root mean percent error is normalized to the standard deviation of the FTIR data in ($\sigma_F$) used in the respective analysis.

$$NRMSE = \frac{1}{\sigma_F}\sqrt{\left[\sum_{i=1}^{N}(C_{m,i} - C_{F,i})^2\right]} \ . \tag{3}$$

In addition to evaluating the models using every available FTIR data point in the analysis years, the monthly mean annual cycles are also presented. The mean partial columns are calculated by taking the mean of every point in a month, for 2008, 2009, 2014, and 2015. The model means are made in the same manner, using only the smoothed partial columns that have a corresponding matching FTIR measurement, as defined above. The mean percent difference is the mean value from Eq. 2 for each month across the years, where error bars represent the standard deviation of this mean. The mean of these monthly
mean differences is used to calculate the overall mean percent difference for each model, where the uncertainty is the standard deviation. Finally, the multi-model mean (MMM) is calculated by taking the mean of the monthly means of all models.

## 4 Results and discussion

This section presents the analyses described above, for $CH_4$, CO and $O_3$, and discusses the findings in the context of the 2021
AMAP SLCF Assessment Report, and other related literature. Given the volume of data (three species, five locations, and 11 models), only selected plots are shown in the main text, with the remaining figures provided in Appendices A-C. The results from Eureka, Canada are shown in the main text as this location is at the highest latitude and has the most measurement points for all species. All the comparisons shown are for a 0-7 km partial column, where the model output is smoothed as described by Eq. 1.

### 4.1 $CH_4$

$CH_4$ is the second most important greenhouse gas (GHG) after $CO_2$, and its emissions are expected to increase in the Arctic due to melting permafrost (IPCC, 2021). $CH_4$ is also involved in the formation of tropospheric $O_3$, which is the third




strongest anthropogenic GHG and an air pollutant at the surface. Therefore, it is important for both air quality and climate models to represent $CH_4$ accurately.

It should be noted that the $CH_4$ concentrations in these models have been prescribed and only three of the 11 participating models provided 3-hourly $CH_4$ concentrations (Whaley et al., 2022). The prescribed concentrations are input at the bottom model layer and all come from the same dataset (Prather et al., 2012; Olivié et al., 2021), but the resulting $CH_4$ concentrations differ based on the processes within each model. The $CH_4$ plots for Ny Ålesund, Thule, Kiruna, and Harestua are provided in Appendix A, following the same order discussed here for Eureka.

Figure 3 shows the Eureka time series for the FTIR and models, with (a) the partial columns and (b) the percent difference between the model and measurement. Figure 3 (and Figs. A1-A4) shows that the models simulate a consistent seasonal cycle from year to year for each model/location, with little variance, apart from a few outliers. Figure 4 (and Figs. A5-A8) shows the monthly means of all the years combined; broadly, the pattern of the seasonal cycle of $CH_4$ is consistent, although the amplitude is underestimated. The uniformity seen in the comparisons is likely a consequence of the longer

lifetime of $CH_4$, relative to the other SLCFs, in addition to being prescribed in the models. This is also seen in Fig. 5 (and Figs. A9-A12), where the model and FTIR columns are compared, with the line of best fit and $R^2$ are indicated in the legend.

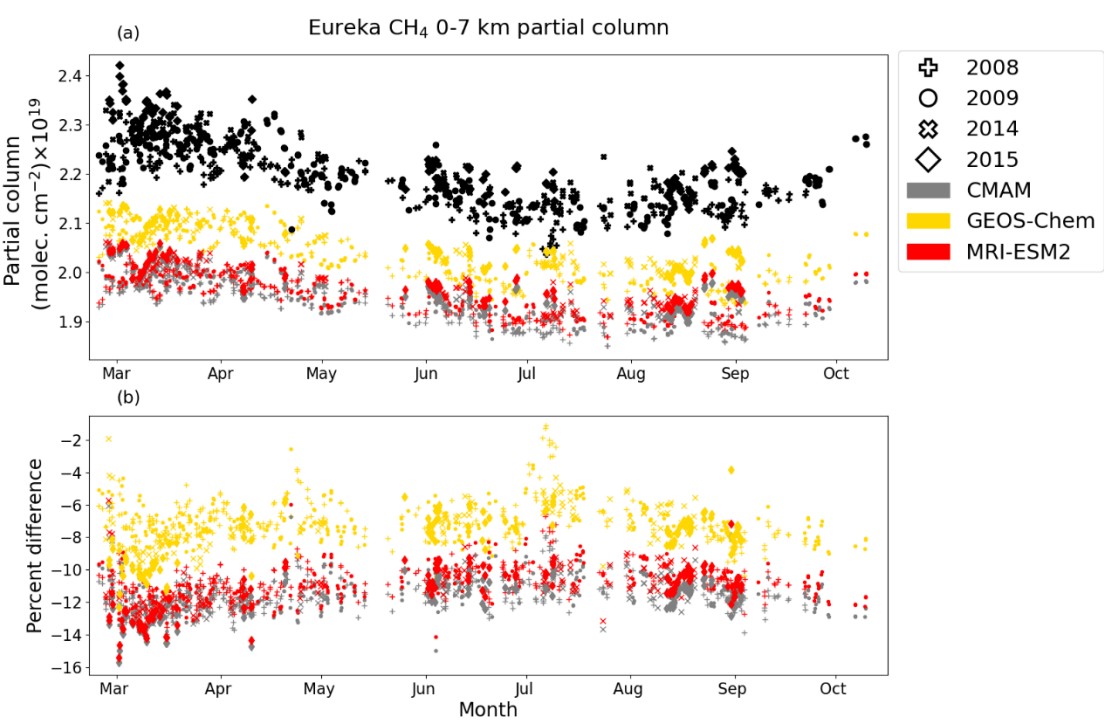

**Figure 3: (a) FTIR (black) and modelled (colour) partial columns of $CH_4$ by day of year, from Eureka. Model data are the nearest in time to each FTIR measurement. (b) Percent difference from Eq. 2 by day of year. Each year is**
**indicated by a different marker.**





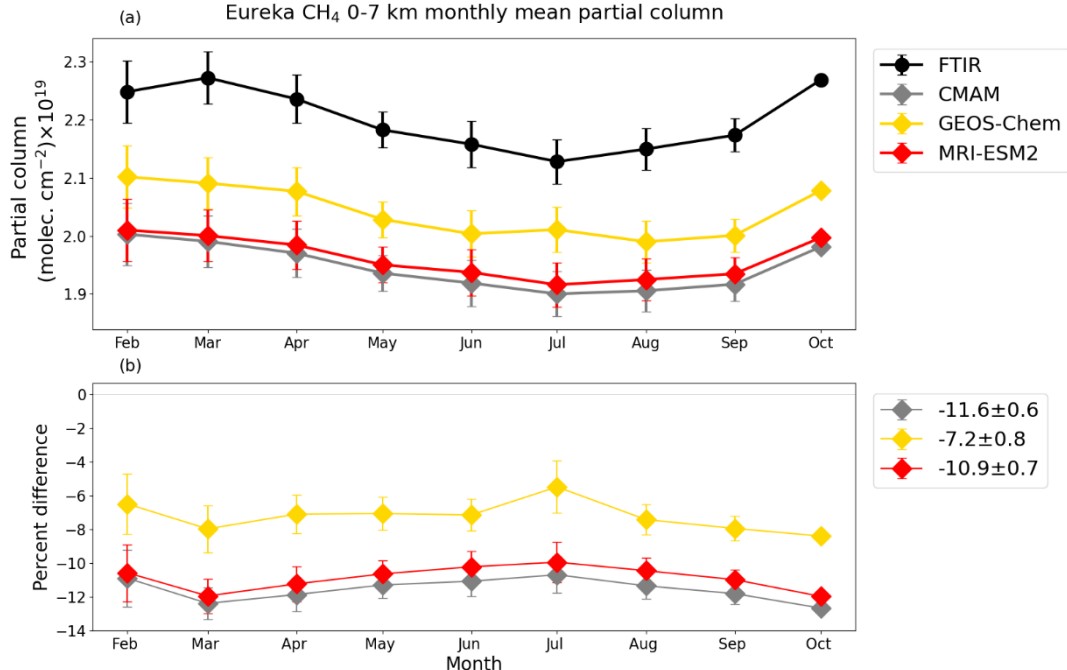

**Figure 4: (a) Monthly mean FTIR (black) and modelled (colour) partial columns of CH₄, from Eureka using model data that are the nearest in time to each FTIR measurement shown in Figure 3. Error bars represent the standard deviation of the monthly mean. (b) Mean percent difference by month. Error bars represent standard deviation of the monthly mean percent difference. The legend on panel (b) shows the overall mean percent difference and its standard deviation.**

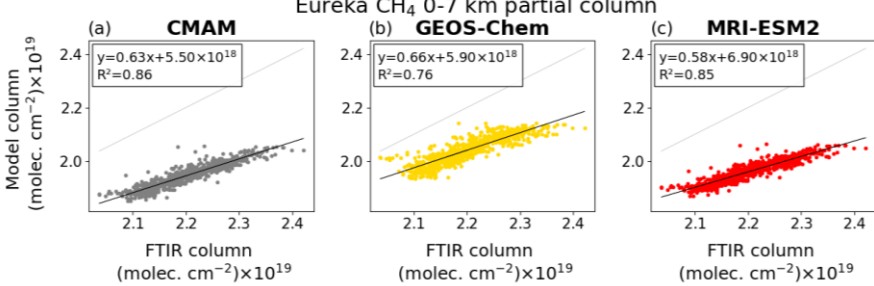

**Figure 5: Model vs. FTIR partial column of CH₄ for Eureka, showing all available model-FTIR corresponding data. The black line is the line of best fit, where the equation and $R^2$ are noted in the legend. The 1:1 line is shown in light grey.**





A summary of the overall mean difference, $R^2$, and the normalized root-mean-square error for each location is
shown in Fig. 6. Across all three models, Arctic $CH_4$ is underpredicted compared to the FTIR measurements. The low bias
we find in this study for the Arctic sites is consistent with the global comparisons of these models to satellite measurements
in Whaley et al. (2022), which found that some models did not distribute $CH_4$ with an accurate north-south gradient,
resulting in low biases in the Arctic and high biases in lower latitudes. Eureka, Thule, and Kiruna all have $R^2$ values nearest
to 1 across the models, and these sites also have the largest number of $CH_4$ measurements. For all models, the $R^2$ values for
Ny Ålesund and Harestua are significantly smaller, while the overall mean percent difference is comparable to the other
locations. The discrepancy is likely attributed to the smaller number of measurement points, causing outliers to have more
weight in the linear regression, which is reflected in the elevated NRMSE for Ny Ålesund across all models. The mean
difference for GEOS-Chem is within the uncertainty of the FTIR measurements for Ny Ålesund, and Thule, as is the mean
difference for MRI-ESM2 at Ny Ålesund (see Table 4). This shows that the models can represent the temporal variability
well, despite low biases in magnitude.

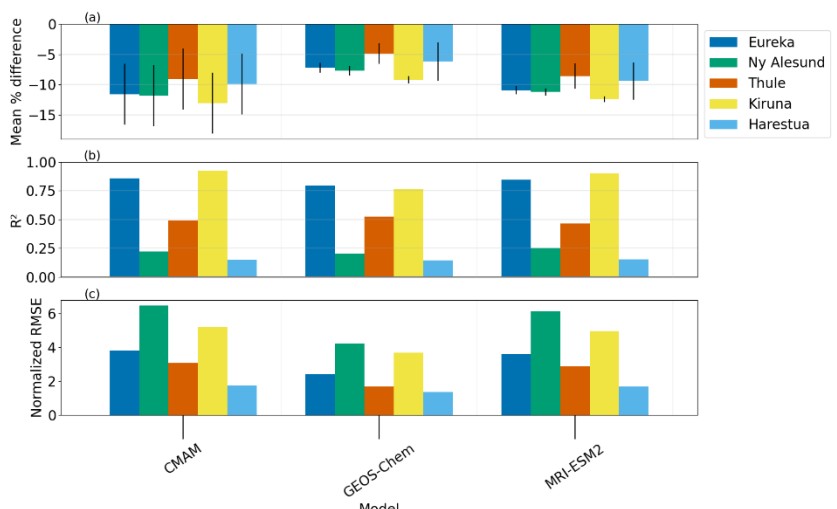

**Figure 6: By model and location: (a) Overall mean percent difference for $CH_4$ 0-7 km partial columns, with error
bars that represent the standard deviation of the mean, as shown in the legend of Figs. 4b, and A5-A8. (b) $R^2$ as
shown in Figs. 5 and A9-A12. (c) Normalized root-mean-square error.**

Figure 7 shows the multi-model mean (MMM) for each location, and the percent difference compared to the
monthly mean FTIR. The error bars and shading represent the standard deviation of the mean. The AMAP SLCF Assessment
Report compares the models with surface $CH_4$ measurements and finds that the MMM bias for Arctic $CH_4$ is +1.3% (AMAP,
2021). When comparing with 0-7 km FTIR partial columns, the MMM bias ranges from -5 to -15% (Fig. 7(f)) and unlike the
results in the AMAP Report, the comparisons are not improved by choosing a multi-model mean because all three models



have a negative bias. The FTIRs show good sensitivity to surface $CH_4$; however, as these column measurements average out $CH_4$ biases over the tropospheric column, they are not expected to exactly match the surface measurement comparisons. The AMAP Report also includes a comparison with upper-troposphere/lower-stratosphere (UTLS) $CH_4$ VMRs as measured by
the ACE-FTS (Atmospheric Chemistry Experiment - Fourier Transform Spectrometer) satellite instrument and finds that the models are biased low in the vicinity of the tropopause (300hPa) (Whaley et al., 2022). This is consistent with the results found here, suggesting that the model simulations of both the lower troposphere (0-7 km partial columns) and the UTLS are biased low.

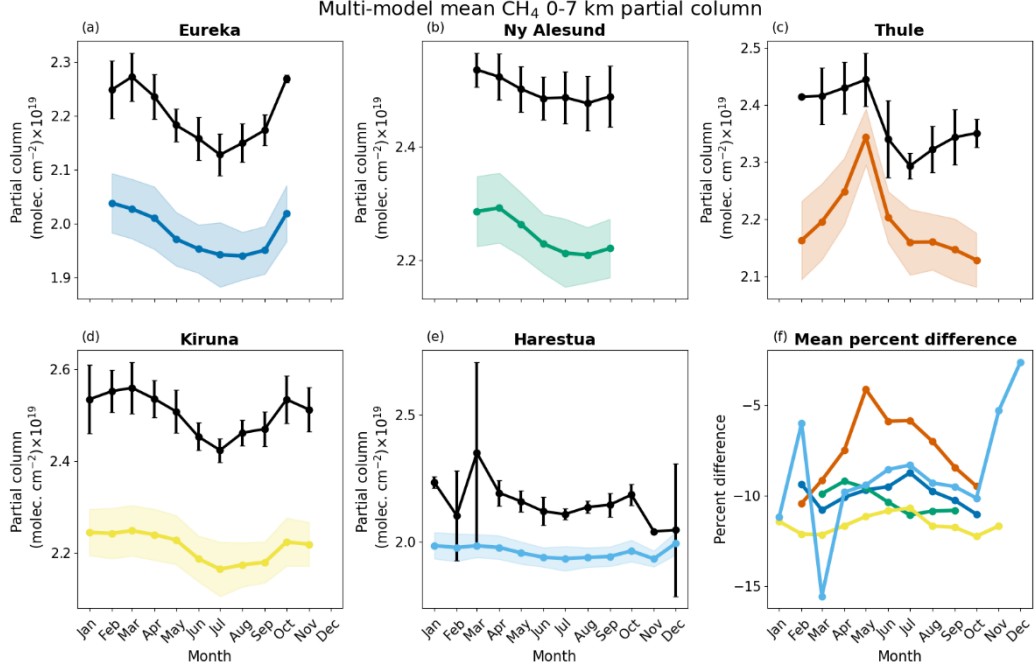

**Figure 7: (a-e) Monthly mean FTIR (black) and multi-model mean (coloured) partial columns of CH₄, with error bars and shaded areas representing the standard deviation of the mean. (f) Monthly mean percent difference of the MMM for all locations.**

**4.2 CO**

Like $CH_4$, CO is involved in tropospheric $O_3$ formation in the presence of $NO_x$. Thus, in order to properly simulate tropospheric $O_3$, it is important for models to accurately simulate CO. In the Arctic, CO is used as a tracer for identifying and quantifying influences from biomass burning and lower latitude anthropogenic emissions (e.g., Fisher et al., 2010; Monks et al., 2015; Viatte et al., 2015; Lutsch et al., 2020)






Nine of the 11 models examined in this study provided 3-hourly outputs for CO; WRF-Chem only has outputs for 2014 and 2015, and GEM-MACH only has data for 2015 (Table 3). The results for CO at Eureka are provided below with Ny Ålesund, Thule, and Kiruna results provided in Appendix B. Figure 8 (and Figs. B1-B3) shows the CO seasonal cycle with all available measurement points; evidence of biomass burning events can be observed in the summer months, where there are sporadic increases in the measured CO. Seven of the nine CO models examined use GFED-based fire emissions. The remaining models are EMEP MSC-W which uses FINN fire emissions and GEM-MACH which uses CFFEPS fire emissions (Whaley et al., 2022). The GFED-based models may overestimate CO from biomass burning as their bias shifts positive in the summertime relative to the rest of the timeseries. This feature is absent for GEM-MACH which does not have a consistent trend between sites during the summer (although results are only available for one year), and for EMEP MSC-W which shifts more negatively in the summertime. It is well known that the fire emissions inventories vary greatly from each other (AMAP, 2021), causing these differences in model results.

Figure 9 (and Figs. B4-B6) shows (a) the monthly mean partial columns and (b) percent differences between the models and the FTIR measurements. This allows for an overview of the mean percent difference and how the model biases change over the year. For example, MATCH exhibits a positive shift in bias from the end of summer to the fall in all locations. GEM-MACH is the only model that has a positive mean difference in all locations. The year-round difference is likely due to the fact that this model used anthropogenic emissions produced locally for most of its regional domain, instead of the ECLIPSEv6B anthropogenic emissions that all of the other models used, and lateral regional boundary conditions provided from MOZART4 (Model for Ozone and Related Chemical Tracers, version 4) global simulations (Emmons et al., 2010; Gong et al., 2018; AMAP, 2021)  Further, Fig. 10 (and Figs. B7-B9) shows the comparison of the modelled and FTIR partial columns, with the line of best fit and $R^2$ indicated in the legend. For many models, the 1:1 comparison (and Figs. 10, B7-B9) shows that models have better agreement with the FTIR for low CO values and the disparity increases as CO increases, i.e. the line of best fit and 1:1 line diverge. The points with the maximum CO VMRs correspond to the FTIR springtime peak in the CO cycle (since wintertime CO measurements are not possible during polar night).



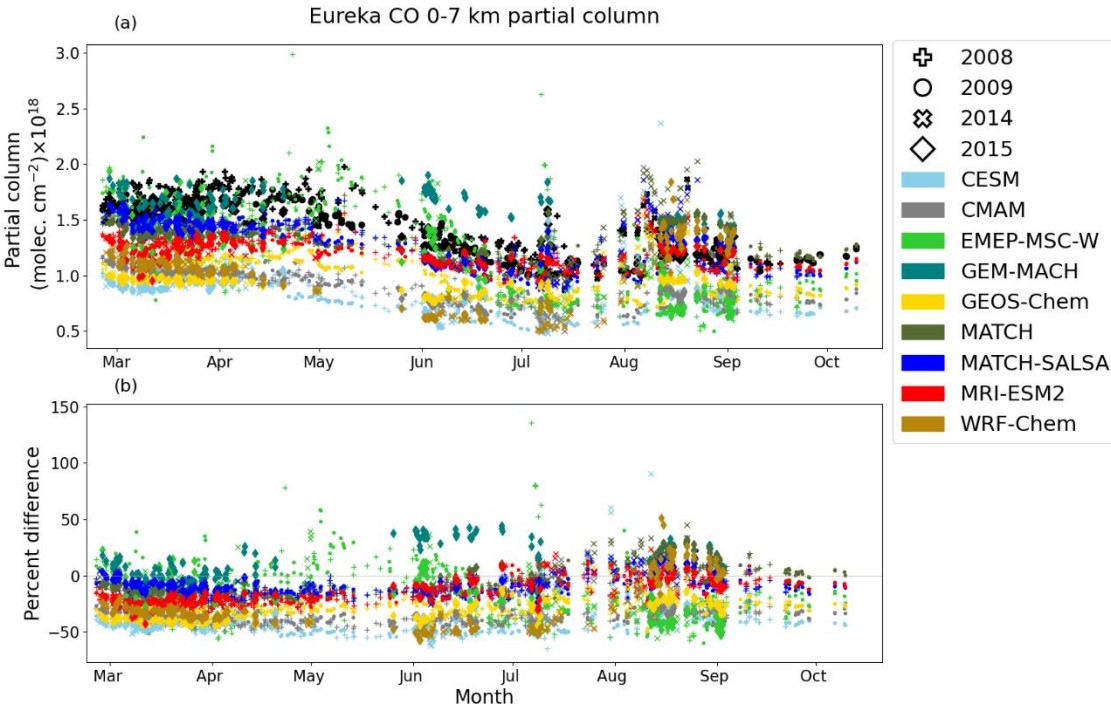


**Figure 8: (a) FTIR (black) and modelled (colour) partial columns of CO by day of year, from Eureka. Model data are the nearest in time to each FTIR measurement. (b) Percent difference from Eq. 2 by day of year. Each year is indicated by a different marker.**



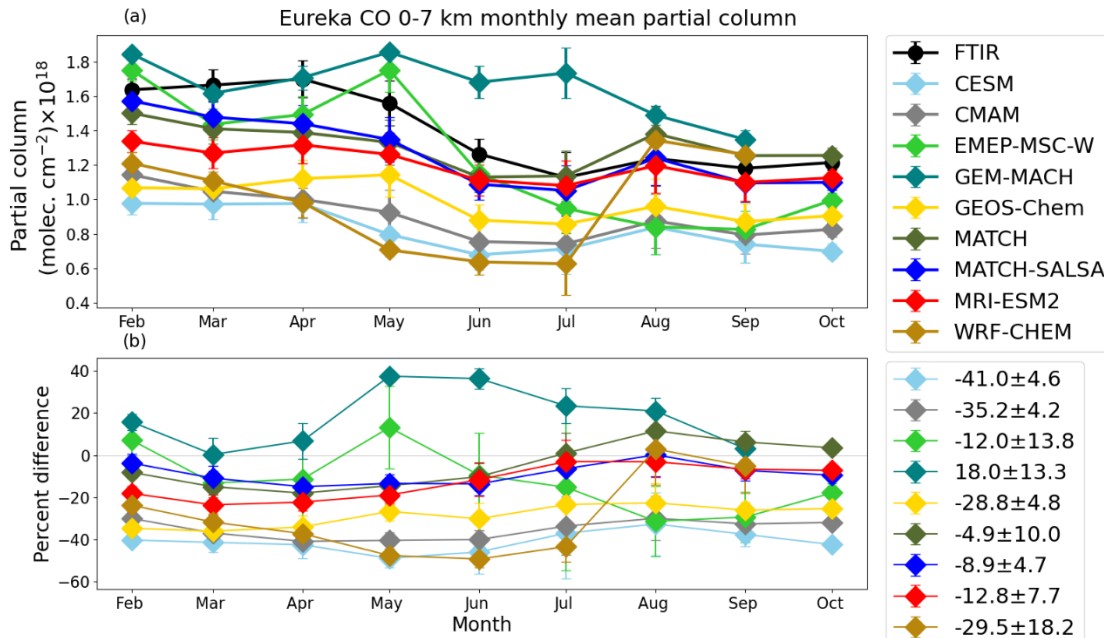


**Figure 9: Monthly mean FTIR (black) and modelled (colour) partial columns of CO, from Eureka using model data that are the nearest in time to each FTIR measurement shown in Figure 8. Error bars represent the standard deviation of the monthly mean. (b) Mean percent difference by month. Error bars represent standard deviation of the monthly mean percent difference. The legend on panel (b) shows the overall mean percent difference and its standard**
**deviation.**



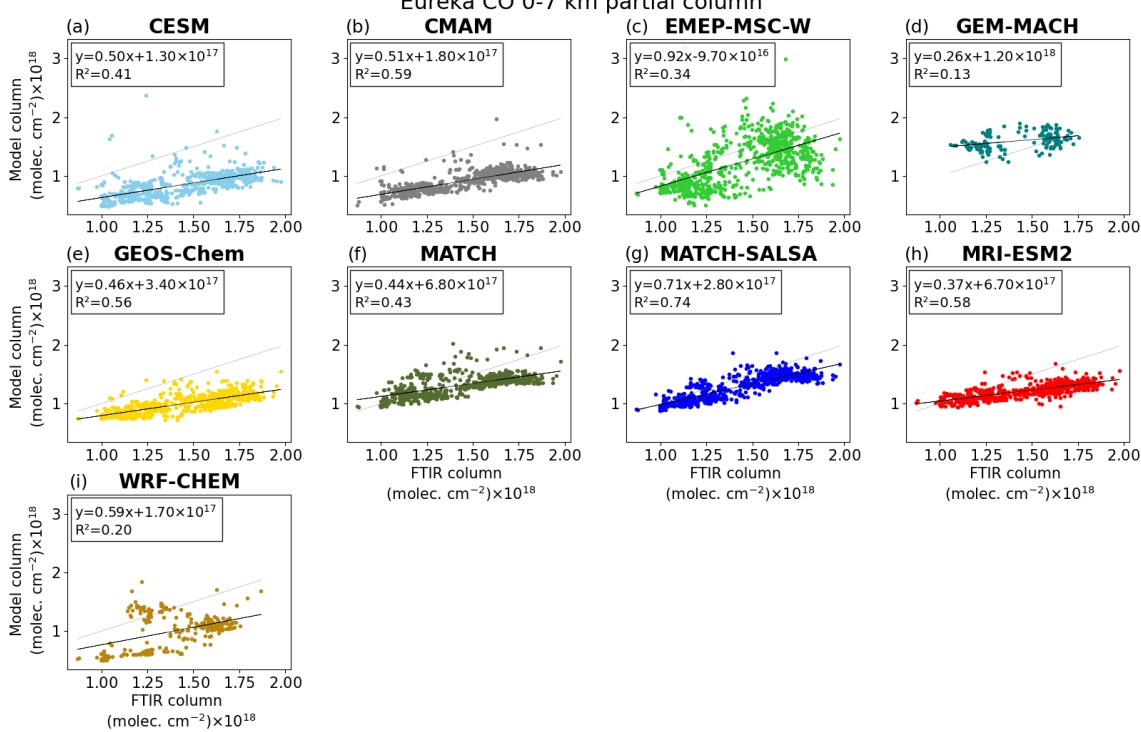

**Figure 10: Model vs. FTIR partial column of CO for Eureka, showing all available model-FTIR corresponding data. The black line is the line of best fit, where the equation and $R^2$ are noted in the legend. The 1:1 line is shown in light grey.**

Figure 11 summarizes the overall mean percent difference, $R^2$, and normalized root-mean-square error for all locations. GEM-MACH has a mean percent error that is within the FTIR error for Thule and Kiruna, EMEP MSC-W and MATCH are simulated within the average error for Ny Ålesund (see Table 4). MATCH-SALSA and MRI-ESM2 exhibit high $R^2$ and low percent difference across all locations, relative to the other models' values, although their columns do not fall within the FTIR uncertainties. GEM-MACH and MATCH have NRMSE comparable to MATCH-SALSA and MRI-ESM2, despite generally lower $R^2$. WRF-Chem shows a better agreement with the FTIR comparisons for Eureka, where the NRMSE is comparable to CESM, CMAM and GEOS-Chem. This is likely a result of the increased density of measurement points in August and September, when WRF-Chem exhibits a minimum bias compared to the FTIR data. Overall, four model-location pairs have a mean difference within the average FTIR 0-7 km partial column uncertainty (see Table 2), and when including the standard deviation of the mean difference, an additional eight pairs meet this criterion.



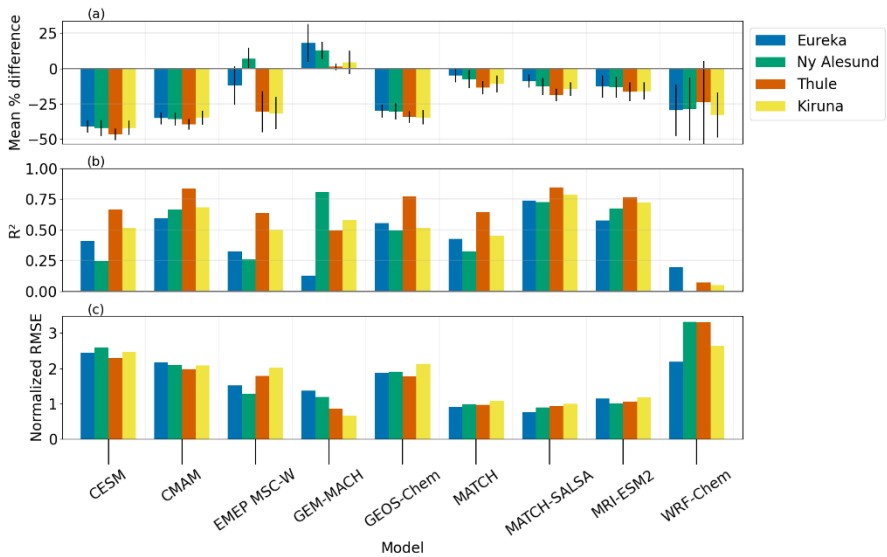

**Figure 11: By model and location: (a) Overall mean percent difference for CO 0-7 km partial columns, with error bars that represent the standard deviation of the mean, as shown in the legend of Figs. 9b andB6. (b) $R^2$ as shown in Figs. 10 and B7-9. (c) Normalized root-mean-square error.**


Figure 12 shows the monthly MMM for CO at each location, with the percent difference in the last panel (f). This highlights the general tendency of the models to underpredict tropospheric CO more in the spring than in the summer. Similar trends have been found in other Arctic model-measurement comparison studies. The AMAP SLCF Assessment Report found that compared to CO from various surface networks, the models had a greater bias than for the other SLFCs

examined, underestimating CO in the spring and overestimating CO in the summer (AMAP 2021). The same pattern was observed when comparing with MOPITT (Measurements of Pollution In The Troposphere) satellite CO at the 600 hPa level (Whaley et al., 2022). The change from a negative winter-spring bias to a positive summer bias was observed in model comparisons to surface CO measurements at two additional Arctic sites, Zeppelin, Norway and Utqiagvik/Barrow, USA (Whaley et al., 2023).

The POLARCAT (Polar Study using Aircraft, Remote Sensing, Surface Measurements and Models, of Climate, Chemistry, Aerosols and Transport) Model Intercomparison Project (POLMIP) examined 11 atmospheric models in relation to a variety of Arctic observations taken in 2008 (Emmons et al., 2015). In that study, models were run for 2008 with a standardized emissions inventory; there is some overlap of models examined here, although a different emissions input was used (see Emmons et al., 2015 for full project description). Similar to the results presented here, the POLMIP study found

that relative to surface, airborne, and satellite Arctic tropospheric measurements, CO was underpredicted by the models (MMM gross error 9-12%), with a more negative bias in the winter/spring compared to the summer, although the models still broadly captured the seasonal cycle, (Monks et al., 2015). Using an idealized tracer, POLMIP examined anthropogenic



and biomass burning influences in Arctic regions, demonstrating a seasonal dependence of transport efficiency. It was shown that for anthropogenic emissions, Europe influences the surface CO, while Asia and North America have more influence

higher in the troposphere (Monks et al., 2015). Further, the tracer investigation shows that OH differences account for more variability between the models than the transport mechanisms within the individual models. However, it can be noted that although models may reduce negative biases through better OH chemistry, this alone will not resolve the differences between the model and measurements (Monks et al., 2015).

The current study, the POLMIP study, and the AMAP Report exhibit similarities in the model-measurement

comparisons of CO, most notably, all three studies show negative biases early in the year, which shift positively in the summer; the model-FTIR comparisons become less negative, while the AMAP-surface measurement comparisons change to a positive bias. Lutsch et al. (2020) also reported a low bias in GEOS-Chem lower tropospheric CO columns compared with measurements from 10 FTIR stations, including four sites in this study, although they found a greater underestimation for Eureka and Thule in July and August due to transported boreal wildfire emissions not fully captured by the model,

particularly for years after 2015 not included in the present study. Previously published studies point to underestimated emission fluxes as a source of the discrepancies (Monks et al., 2015, Whaley et al., 2022; 2023). The results of the model-FTIR comparisons presented here support this reasoning, as the model with a positive bias (GEM-MACH) has a different emissions input, with possibly more complete emissions in the Arctic, as this was a high-resolution Arctic version. The models may be improved with more refined OH chemistry, although it is unlikely to completely resolve the inconsistencies

(Monks et al., 2015); improvements to long-range transport and biomass burning inventories could also reduce the differences between model results and measurements.



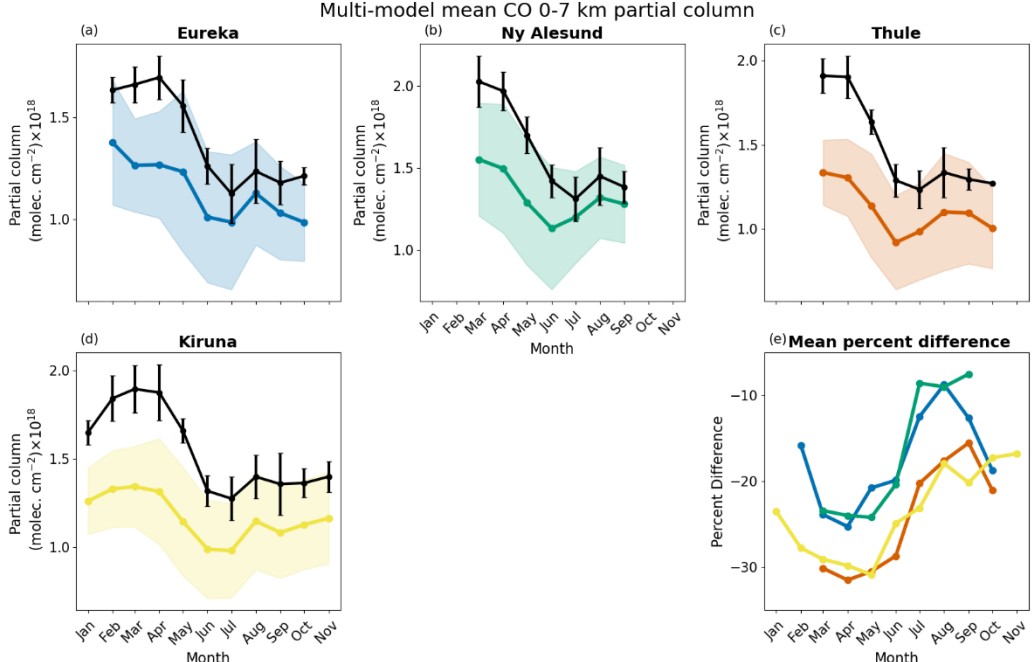

**Figure 12: (a-d) Monthly mean FTIR (black) and multi-model mean (coloured) partial columns of CO, with error bars and shaded areas representing the standard deviation of the mean. (e) Monthly mean percent difference of the MMM for all locations.**

### 4.3 $O_3$

Tropospheric $O_3$ is both a significant anthropogenic GHG and an air pollutant that has impacts on human health and ecosystems. It is a secondary pollutant, produced by photochemical oxidation of volatile organic compounds in the presence of $NO_x$. In addition to atmospheric chemistry, its production is highly sensitive to meteorological conditions. Therefore, it is difficult for models to accurately simulate tropospheric $O_3$. Yet, $O_3$ is often quite well reproduced, possibly due to compensating biases in its precursors (Whaley et al., 2022). Although progress has been made, sparse observations, Arctic amplification, and a changing global climate hinder the understanding and modelling of $O_3$ in Arctic regions (Whaley et al., 2023). For a summary of the current understanding of Arctic tropospheric $O_3$, see Whaley et al. (2023).

All 11 of the models examined in this study provide 3-hourly $O_3$ concentrations. Results for Eureka are shown below and the associated figures for $O_3$ at Ny Ålesund, Thule, Kiruna, and Harestua are provided in Appendix C. Figure 13 (and Figs. C1-C4) show the entire time series of 0-7 km $O_3$ partial columns by time of year on panel (a), and the percent difference between the model and FTIR in panel (b). These plots demonstrate the variation between the models and throughout the year, which is likely a by-product of the complexity in modelling tropospheric $O_3$. Figure 14 (and Figs. D5-D8) reduce the previous figure(s) to the monthly mean (a) partial columns and (b) percent differences to highlight the parts



of the year which are over or underpredicted. For example, springtime $O_3$ is of interest in the Arctic due to the springtime maximum in its seasonal cycle, and the potential for both stratospheric ozone intrusions into the upper (mid) troposphere and surface $O_3$ depletion events (ODEs) due to bromine explosions and halogen chemistry. However, the FTIR $O_3$ seasonal cycle does not have a springtime minimum from surface ODEs, as one might expect from surface measurements (Solberg et al., 1996; Berg et al., 2003; Skov et al., 2006; Eneroth et al., 2007; Whaley et al, 2023). The Arctic surface ODE features are primarily limited to the near surface/lower boundary layer (<2 km), whereas the 0-7 km partial column is dominated by the free troposphere (Zhao et al., 2016). It can be noted that all of the models in this study lack the necessary halogen chemistry needed to simulate ODEs in the high Arctic (Whaley et al., 2023). Across all locations, MATCH-SALSA overpredicts $O_3$ by 35-75% in winter, which gradually declines until May, after which the bias becomes negative. UKESM1, GEM-MACH, and GEOS-Chem underestimate springtime $O_3$ most substantially across all sites, which may be attributed to a low bias in the models' lateral boundary condition, inaccuracies in model water vapor and/or a lack of $O_3$ transported from mid-latitudes. In the case of the regional GEM-MACH model, low biases in the lateral boundary conditions may also be contributing. CESM, CMAM and DEHM demonstrate reasonable agreement with measured springtime $O_3$ across locations, in addition to a smaller overall mean percent difference, relative to other models. EMEP MSC-W and WRF-Chem simulate springtime $O_3$ comparable to the aforementioned models, although negative biases later in the year lead to a larger overall mean percent difference. This may indicate that these models have too much photochemical $O_3$ loss in the summer months.

Figure 15 (and Figs. C9-C12) shows the model versus FTIR $O_3$ 0-7 km partial columns, with the line of best fit and $R^2$ shown in the legend, along with the 1:1 line. As with CO, many of the models have a better agreement for smaller $O_3$ partial columns and fail to capture larger values. This could be related to the underestimation in precursor species (such as CO or $NO_x$), a lack of long-range transport, an underestimation of ozone production in air masses during long-range transport to the Arctic, or a combination thereof. Using a MOZART-4 tagged tracer simulation of $O_3$, Wespes et al. (2012) examined source attributions of the tropospheric $O_3$ columns measured by the FTIR instruments at Thule and Eureka. Their analysis shows that the retrievals have minimal contribution from the a priori (~1%), resulting in high vertical sensitivity throughout the troposphere. The tropospheric column source contributions were estimated, where over half was attributed to anthropogenic sources, followed by stratospheric influence and lastly lightning and biomass burning emissions (Wespes et al., 2012). The seasonal cycle of Arctic $O_3$ has been shown to vary based on geographical conditions, such as if the site is costal, inland or at a high elevation (Whaley et al., 2023). Moreover, $O_3$ partial columns can be variable because they depend on the vertical distribution of $O_3$, which is determined by a combination of emissions, chemistry, dynamics, and radiation, all of which vary with altitude (Rap et al., 2015). Notably, Arctic $O_3$ columns have strong gradients in the influences on the vertical profile from mid-latitude regions (Europe, North America and Asia), which also vary with season (Monks et al., 2015). The combination of these factors leads to an increasingly complex series of difficult to model processes, which can also result in compounding errors. Without sensitivity simulations, like those carried out in Monks et al. (2015) and Rap et al. (2015), it is difficult to definitively say which of these processes are responsible for the underestimations found in this study, and as such is recommended for future work.



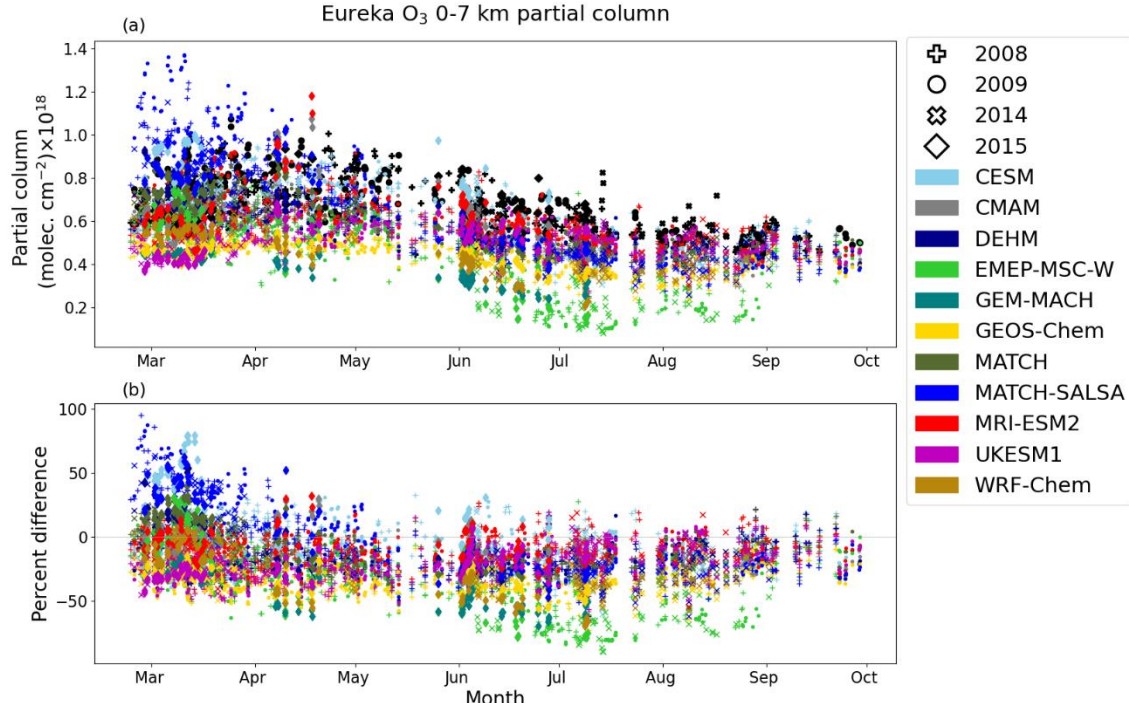

**Figure 13: FTIR (black) and modelled partial columns of $O_3$ by day of year, from Eureka. Model data are the nearest in time to each FTIR measurement. (b) Percent difference from Eq. 2 by day of year. Each year is indicated by a different marker.**




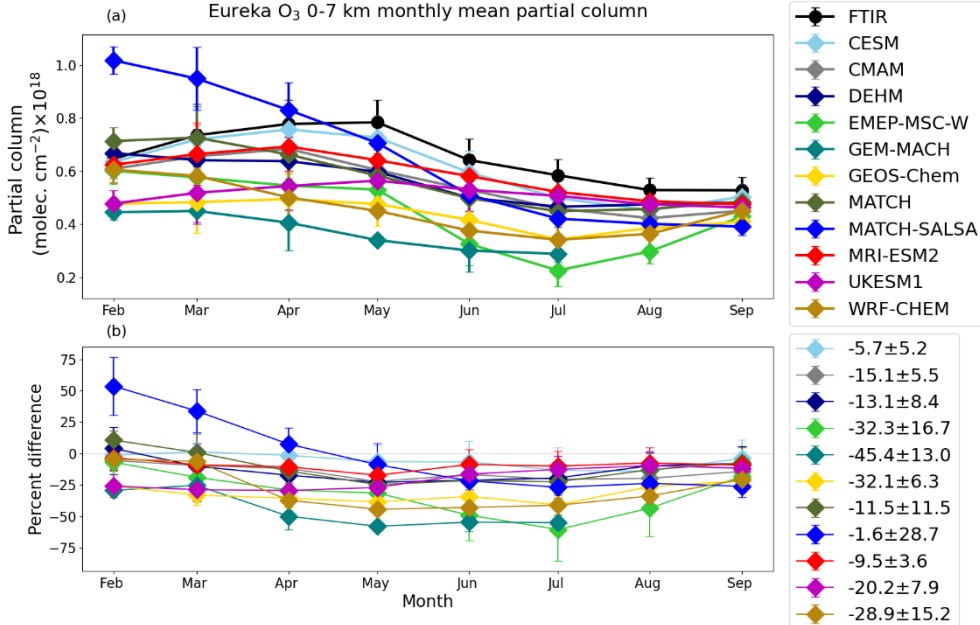

**Figure 14 (a) Monthly mean FTIR (black) and modelled partial columns of O₃, from Eureka using model data that are the nearest in time to each FTIR measurement shown in Figure 13. Error bars represent the standard deviation of the monthly mean. (b) Mean percent difference by month. Error bars represent standard deviation of the monthly mean percent difference.**






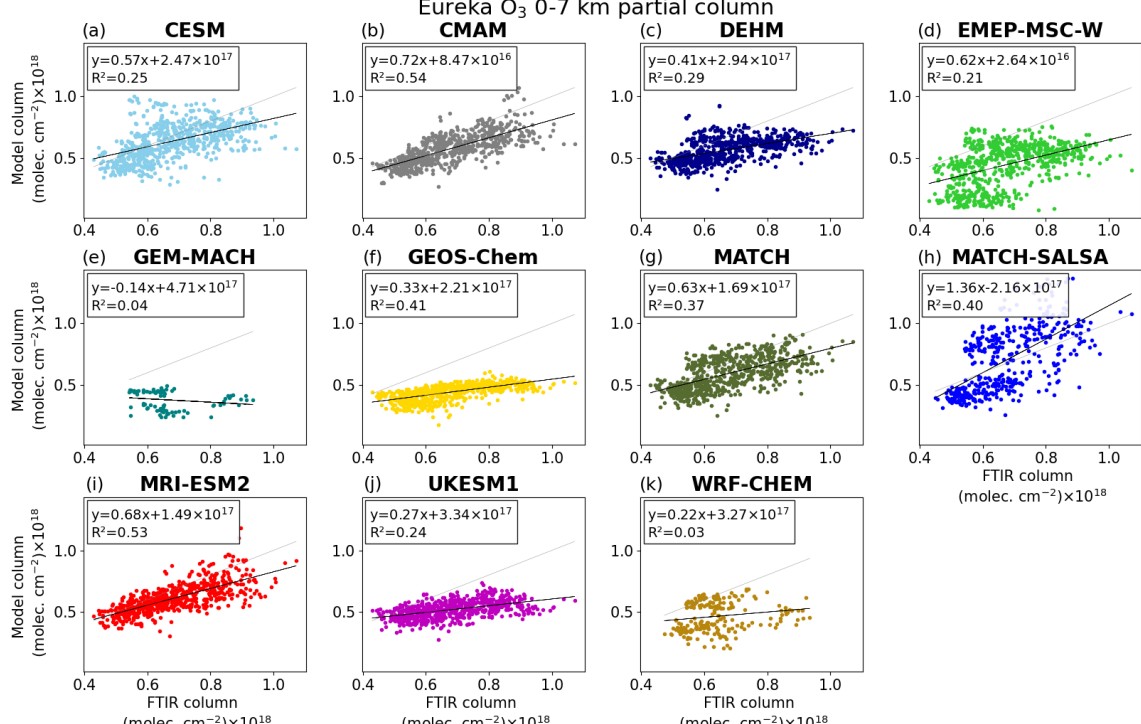

**Figure 15: Model partial column vs. FTIR partial column of O₃ for Eureka, showing all available model-FTIR corresponding data. The black line is the line of best fit, where the equation and R² are noted in the legend. The 1:1 line is shown in light grey.**


Figure 16 shows the summary of $O_3$ mean percent differences, $R^2$, and normalized root-mean-square error. The model-FTIR comparisons reveal that the level of chemistry and spatial resolution of the models does not necessarily improve results. For example, WRF-Chem, EMEP MSC-W and GEM-MACH show a low $R^2$ and higher NRMSEs (varying between sites and models). These models have detailed chemistry and were run at higher spatial resolutions, whereas for example

CMAM has a coarser resolution with simplified chemistry and demonstrates larger $R^2$ and smaller mean percent differences (Fig.16) (Whaley et al., 2023). Overall, the $O_3$ partial column comparisons show significant variation, although again are largely underpredicted. The $R^2$, mean percent difference, and NRMSE are relatively consistent, where models with a larger percent difference also have a weaker correlations and higher NRMSEs. An exception to this is CESM, which has one of the smallest overall differences across the models and locations. However, in the model vs. FTIR plot(s) (and Figs. 15, C9-C12),

CESM has considerable scatter above and below the line of best fit, resulting in a decreased mean difference, while also reducing $R^2$, unlike MRI-ESM2, which has a similar mean percent difference and NRMSE, but a stronger linear correlation.

To supplement the aircraft and satellite campaigns undertaken for the POLARCAT study, daily mean $O_3$ measurements from the FTIR instruments at Eureka and Thule were compared to MOZART-4 simulations in Wespes et al.



(2012). When examining a partial column from the ground to 300 hPa (approximately 9 km), the smoothed model showed a
bias of -15% relative to the FTIR. This is consistent with their analysis of aircraft observations, which revealed that the
model underestimated $O_3$ by 5-15%. In the current study, across all the locations and models, 24 of the 55 mean percent
differences are within ±15% (see Table 4). The similarity of these results to those in Wespes et al. (2012) indicates that
despite model updates in the last decade, model biases remain unchanged. The FTIR uncertainty for $O_3$ partial columns
ranges from 3.9% to 8.2%; the overall mean percent difference for MATCH-SALSA falls within these uncertainty bounds
for all locations, and CESM, DEHM, MATCH and MRI-ESM2 are within FTIR uncertainty for all locations but Ny
Ålesund.

The AMAP SLCF Assessment Report finds that the multi-model mean of Arctic $O_3$ has a bias of +11 ± 3% relative
to surface measurements (AMAP, 2021). When partitioning results by region, all the models had positive biases when
compared to the surface measurements in Alaska and negative biases in Northern Europe, resulting in a relatively small
mean bias across the Arctic as a whole (Whaley et al., 2022). Inaccuracies in long-range transport of $O_3$ and its precursors
may have contributed to the increased discrepancy seen in the model-FTIR comparisons of the current study, particularly in
partial columns with larger values. For example, the underestimation of CO may contribute to the negative bias in $O_3$ (see
Figs. 10-11). Most models in AMAP (2021) show negative biases for Greenland and Northern European locations, which
would correspond closer geographically with the FTIR sites examined here. When comparing the AMAP models to TES
(Tropospheric Emission Spectrometer) and ACE-FTS satellite $O_3$ measurements, the biases are negative at lower altitudes,
and become positive at higher altitudes (Whaley et al., 2022). AMAP model vs. ozonesonde comparisons showed similar
elevated positive biases around 6-8 km of up to ±50%, again indicating that the models may produce too much $O_3$ from mid-
latitude anthropogenic emissions or that there may be too much downward transport of $O_3$ from the stratosphere (Whaley et
al., 2023). The best performance in that study came from the multi-model mean, which simulated $O_3$ within ± 8% throughout
the troposphere.





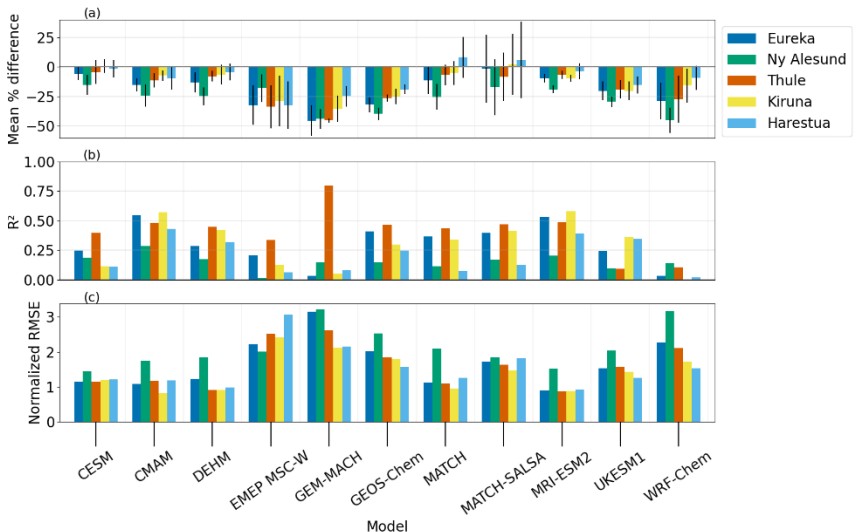

**Figure 16: By model and location: (a) Overall mean percent difference for O₃, error bars represent the standard deviation of the mean, as shown in the legend of Fig.14b, Fig.C5-C8. (b) R² as shown in Fig.15, Fig C9-C12. (c) Normalized root-mean-square error.**


Figure 17 shows the monthly MMM for $O_3$ at all locations, along with the monthly mean FTIR and the associated percent difference. This shows that the models, as a whole, have an increased negative bias in the middle of the year relative to the winter, while still exhibiting a negative bias overall. The longitudinal range of sites examined here may limit biases to be negative, not capturing the positive-negative gradient from west-east in $O_3$ found in the AMAP Report (AMAP, 2021; Whaley et al., 2022). Nonetheless, the model-FTIR $O_3$ comparisons reflect the proclivity of the models to underpredict Arctic $O_3$ in the lower troposphere, as also found in the aforementioned studies. The results of this study agree with results from previous studies and suggest that improvements are still needed for accurate modelling of $O_3$ and CO in the Arctic (Whaley et al., 2023). Models still require improvements in their treatment of stratospheric-tropospheric exchange and Arctic boundary layer processes to better simulate Arctic $O_3$, as well as further improvements and understanding about processes influencing $O_3$ removal through dry deposition and $O_3$ photochemical production from anthropogenic, biomass burning and natural sources in the lower and mid troposphere.





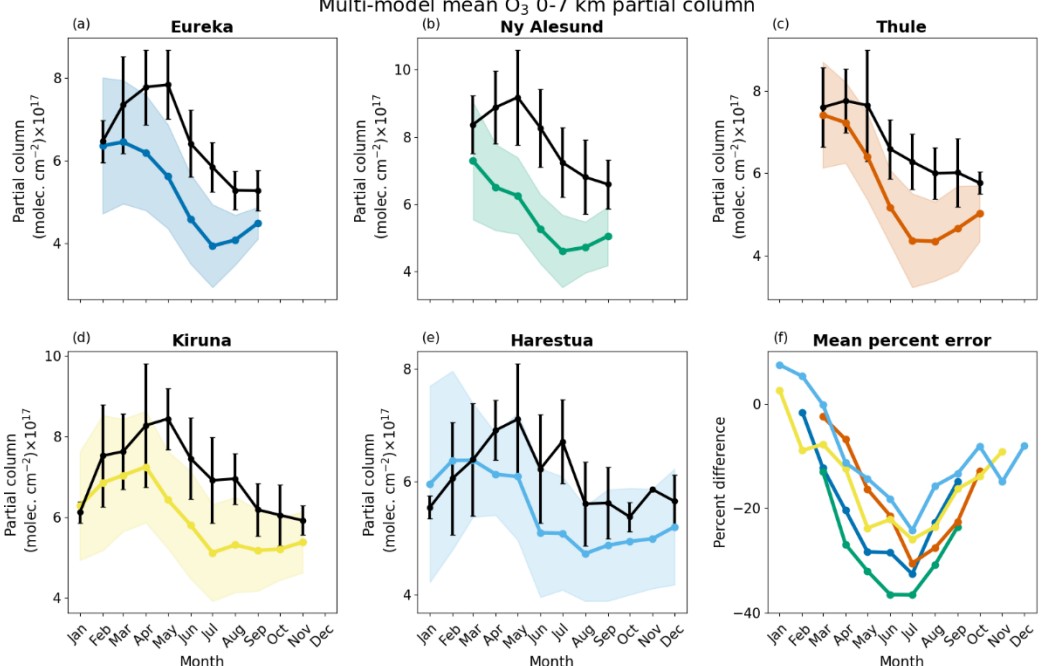

**Figure 17: (a-e) Monthly mean FTIR (black) and multi-model mean (coloured) partial columns of O₃, with error bars and shaded areas representing the standard deviation of the mean. (f) Monthly mean percent difference of the MMM for all locations.**


## 5 Conclusions

This study compares atmospheric models with historical data from five Arctic NDACC ground-based FTIR spectrometers. The models simulate SLCFs and precursor gases with 3-hourly outputs for the years 2008, 2009, 2014, and 2015. Here, a total of three models are evaluated for CH₄, nine for CO and 11 for O₃. The model simulations are compared with FTIR

tropospheric partial column measurements to assess performance throughout the year and across locations.

Generally, across the five locations, the model simulations of 0-7 km partial columns of CH₄, CO and O₃ are underestimated. There were no significant patterns in the biases identified between the sites, species, or models examined. Modelled CH₄ partial columns are relatively consistent across the year, broadly capturing seasonal cycles, with the exception of a few outliers. CO simulations are inconsistent in reproducing the seasonal cycle, underpredicting springtime partial

columns compared to the rest of the year, and skewing differences to be more positive when there are enhancements due to biomass burning events. Similarly, the models underestimated O₃ maxima more than O₃ minima in the troposphere. The multi-model means are reflective of these trends, for which (ignoring outliers), the CH₄ mean percent difference is relatively consistent across the year, CO has a maximum difference in the spring and a minimum in the summer, and O₃ has maximum difference centered around the summer. The AMAP SLCF Assessment Report found the best results using a multi-model



mean for all species when comparing with surface measurements (AMAP 2021; Whaley et al., 2022). However, here, the multi-model means of the tropospheric column for all species are biased low. The average MMM mean difference is approximately -10% for $CH_4$, -21% for CO and -18% for $O_3$ (see Table D1), where the uncertainty of the FTIR 0-7 km partial column is on the order of 6% on average. When examining the models and location pairs individually, the mean difference (inclusive of standard deviation) is within the respective FTIR uncertainty, for seven of 15 model-FTIR

comparisons for $CH_4$, 12 of 34 for CO, and 25 of 55 for $O_3$ (see Table 4).

These evaluations show that models are lacking some degree of transport and/or emissions to accurately reproduce tropospheric columns and seasonal variability in the Arctic. Model evaluation can provide a valuable checkpoint to help improve the representation of the Arctic in atmospheric models. NDACC FTIR spectrometers were selected for this project because of the wide range of species measured, high spectral resolution, multiple high-latitude sites, and publicly available

data. Future work would benefit from the inclusion of sensitivity studies, furthering the model-measurement comparisons with mid-latitude NDACC FTIR sites, and extending comparisons to a longer timeframe, with some models and locations having data from as early as 1990.



**Table 4: Summary of mean percent difference for each model and location by species. MMM is the multi-model mean. The colour scale indicates the mean percent difference relative to the FTIR measurements, from blue (-50%) to red (+50%). A square marker indicates that the mean difference is within the FTIR uncertainty. A triangle marker indicates that the mean difference is within the FTIR uncertainty combined with the standard deviation of the monthly mean percent difference.**

| CH₄ | Eureka | Ny Alesund | Thule | Kiruna | Harestua |
|---|---|---|---|---|---|
| CMAM | | △ | △ | | △ |
| GEOS-Chem | | □ | □ | | △ |
| MRI-ESM2 | | □ | | | |
| MMM | | □ | | | |

| CO | Eureka | Ny Alesund | Thule | Kiruna | Harestua |
|---|---|---|---|---|---|
| CESM | | | | | |
| CMAM | | | | | |
| EMEP-MSC-W | △ | □ | | | |
| GEM-MACH | | △ | □ | □ | |
| GEOS-Chem | | | | | |
| MATCH | △ | □ | | △ | |
| MATCH-SALSA | | △ | | | |
| MRI-ESM2 | | △ | | | |
| WRF-Chem | | △ | △ | | |
| MMM | | △ | | | |

| O₃ | Eureka | Ny Alesund | Thule | Kiruna | Harestua |
|---|---|---|---|---|---|
| CESM | □ | | △ | □ | □ |
| CMAM | | | | □ | △ |
| DEHM | △ | | △ | □ | △ |
| EMEP-MSC-W | | | | | |
| GEM-MACH | | | | | |
| GEOS-Chem | | | | | |
| MATCH | △ | | △ | □ | △ |
| MATCH-SALSA | □ | △ | △ | □ | △ |
| MRI-ESM2 | △ | | △ | △ | □ |
| UKESM1 | | | | | |
| WRF-Chem | | | | △ | △ |
| MMM | | | | | △ |

**colour scale**

| -50 | -40 | -30 | -20 | -10 | 0 | 10 | 20 | 30 | 40 | 50 |
|---|---|---|---|---|---|---|---|---|---|---|

□ model-FTIR mean difference is within FTIR percent error

△ model-FTIR mean difference is within FTIR percent error when including the standard deviation of the mean




# Appendices

## Appendix A – Additional figures for CH4

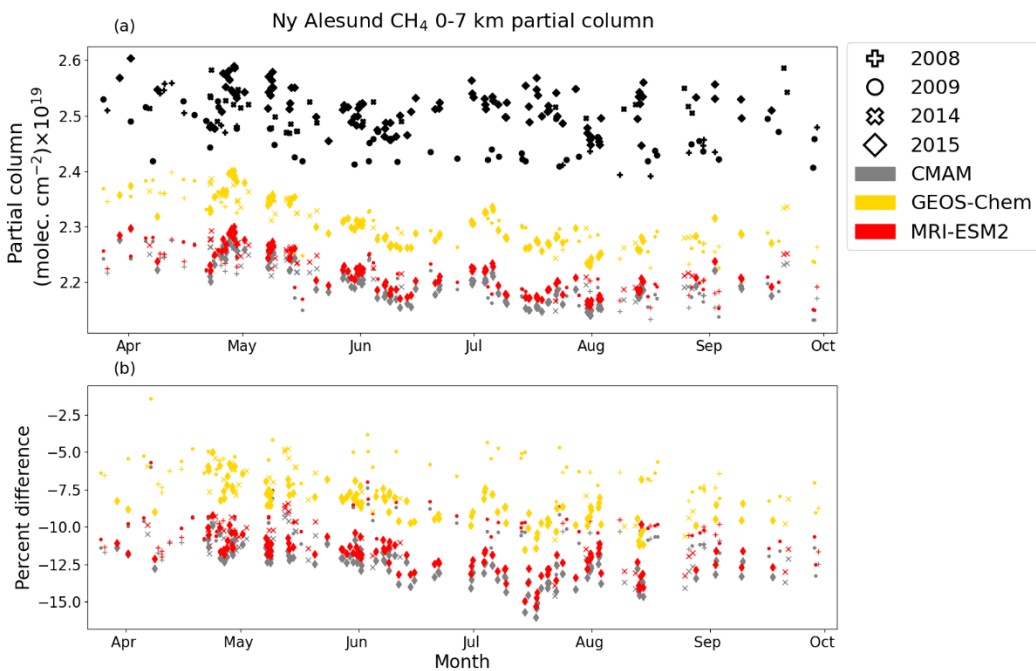

**Figure A1: (a) FTIR (black) and modelled (colour) partial columns of CH4 by day of year, from Ny Ålesund. Model data are the nearest in time to each FTIR measurement. (b) Percent difference from Eq. 2 by day of year. Each year is indicated by a different marker.**






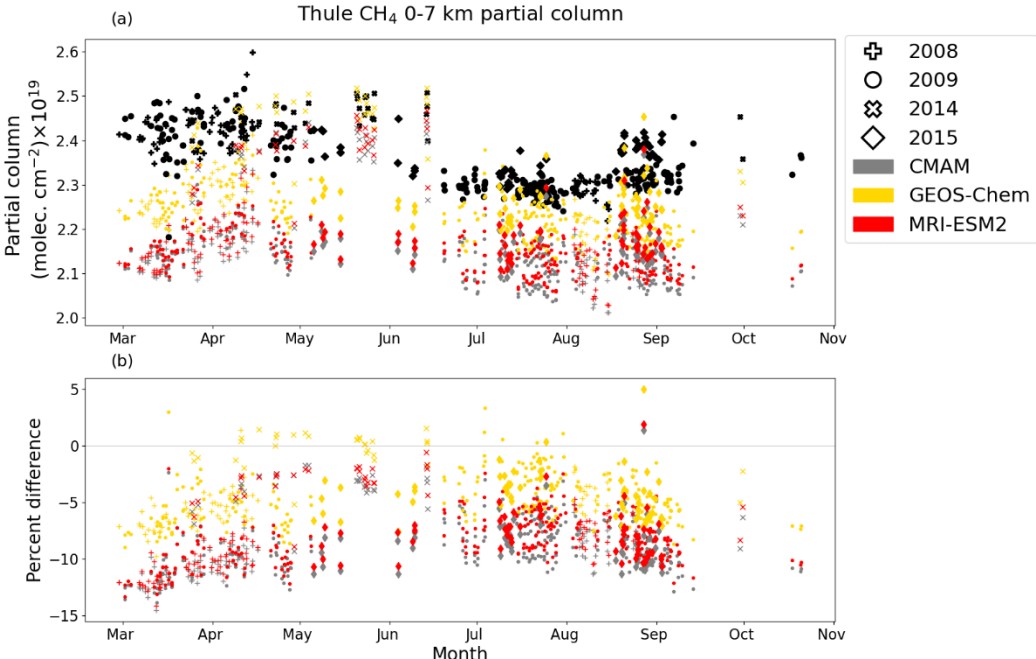

**Figure A2: Same as Fig. A1 but for Thule.**

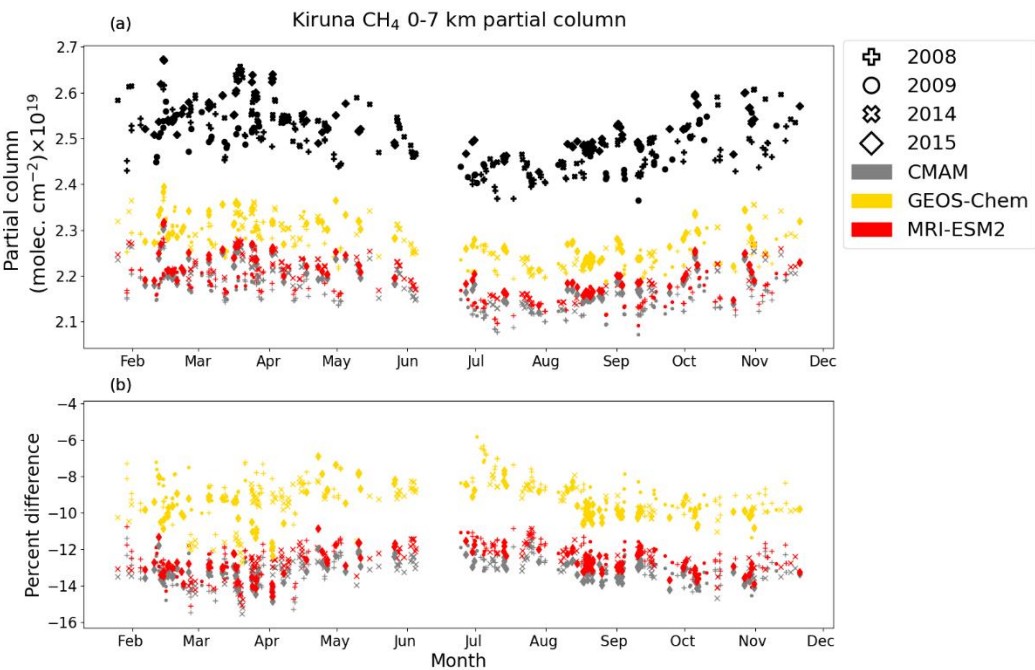

**Figure A3: Same as Fig. A1 but for Kiruna.**



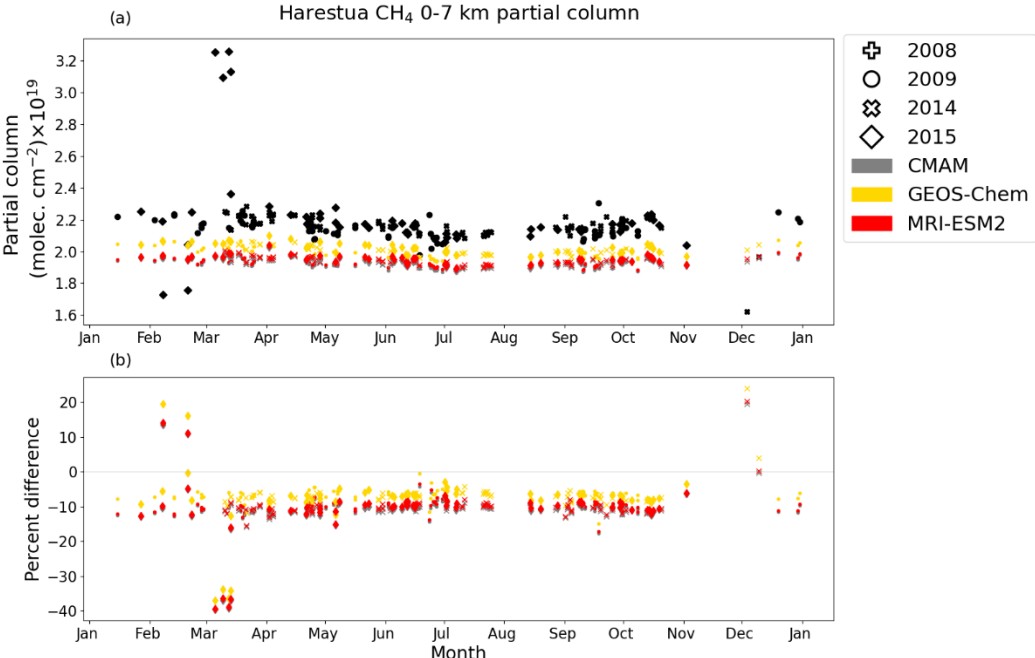


**Figure A4: Same as Fig. A1 but for Harestua.**

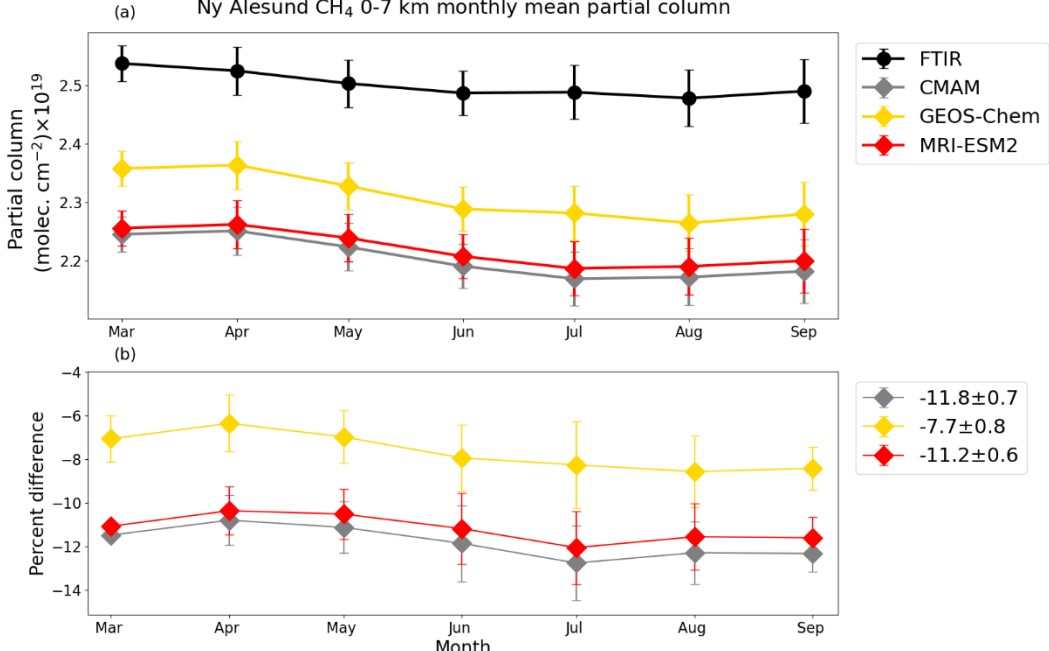

**Figure A5: (a) Monthly mean FTIR (black) and modelled (colour) partial columns of CH₄, from Ny Ålesund using model data that are the nearest in time to each FTIR measurement shown in Figure A1. Error bars represent the standard deviation of the monthly mean. (b) Mean percent difference by month. Error bars represent standard deviation of the monthly mean percent difference. The legend on panel (b) shows the overall mean percent difference with the standard deviation of the overall mean percent difference.**





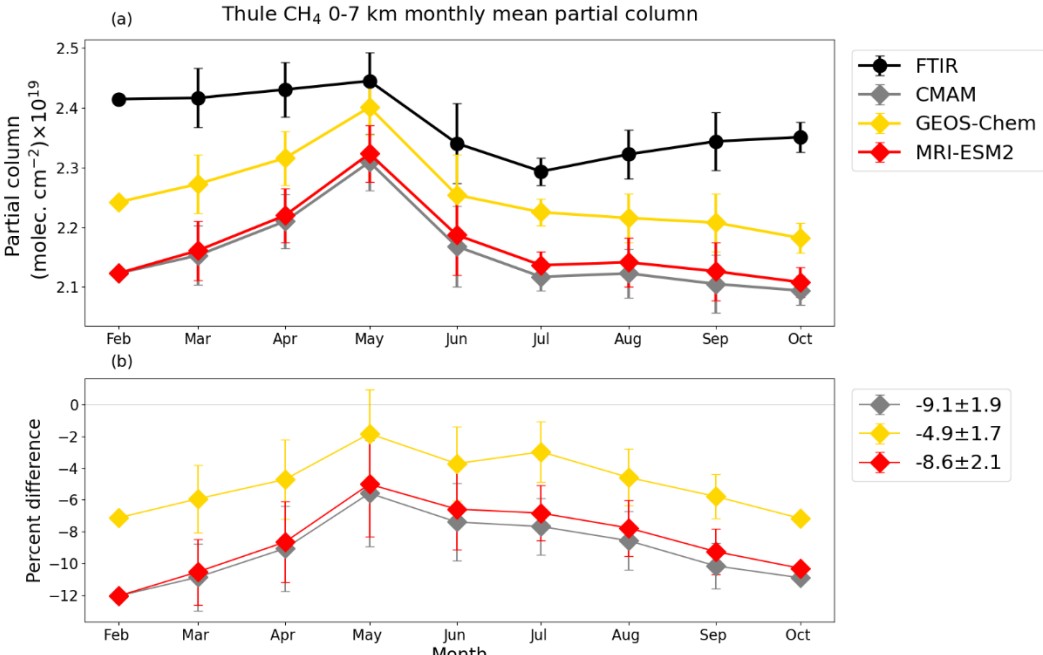


**Figure A6: Same as Fig. A5 but for Thule.**

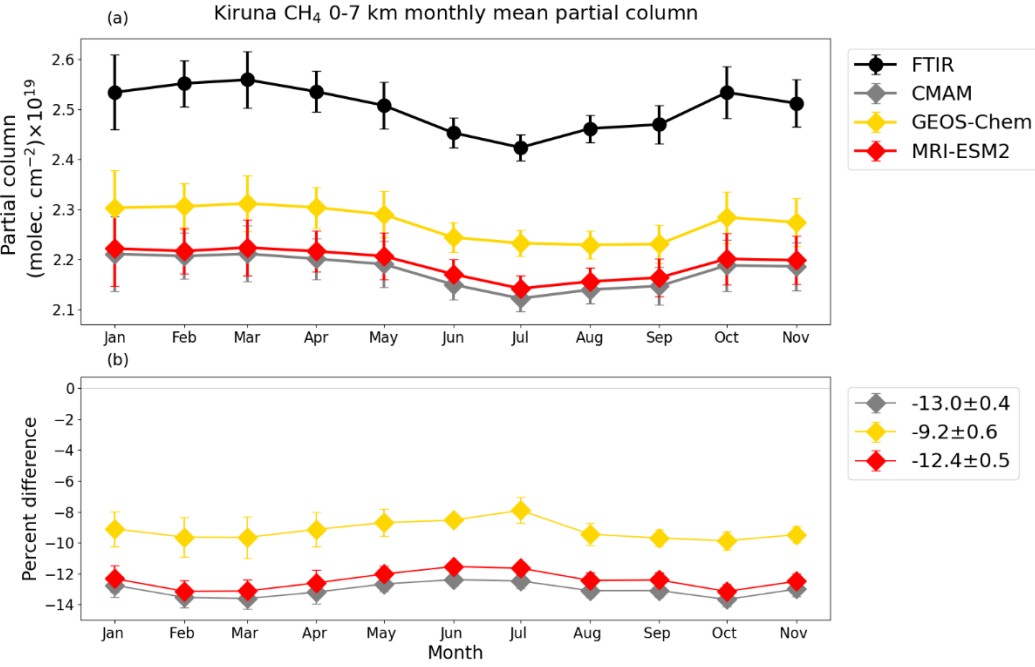

**Figure A7: Same as Fig. A5 but for Kiruna.**





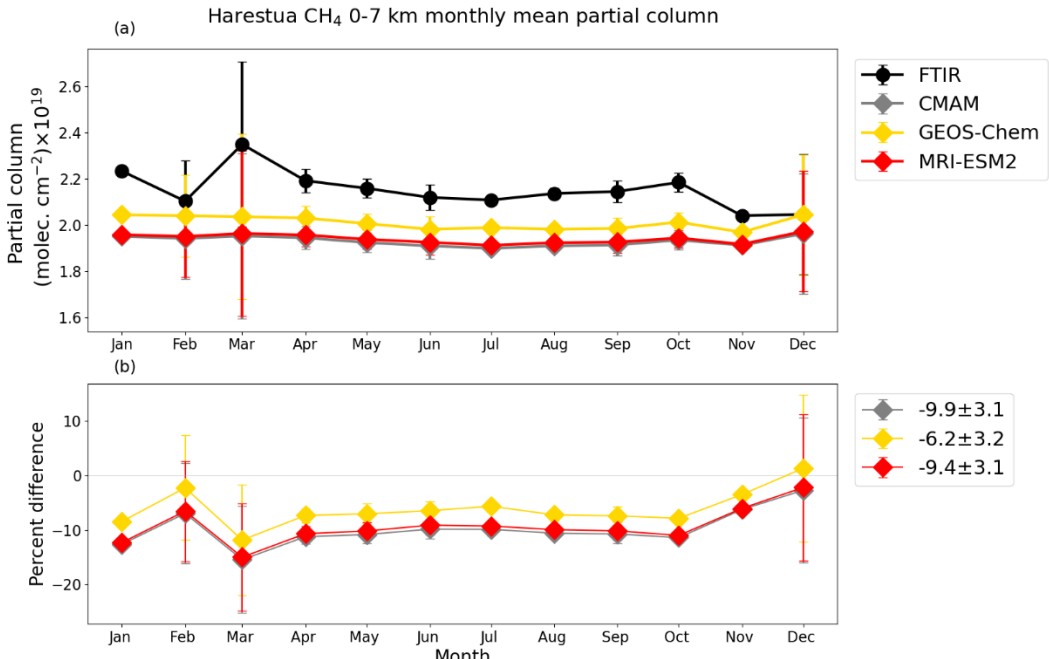


**Figure A8: Same as Fig. A5 but for Harestua.**

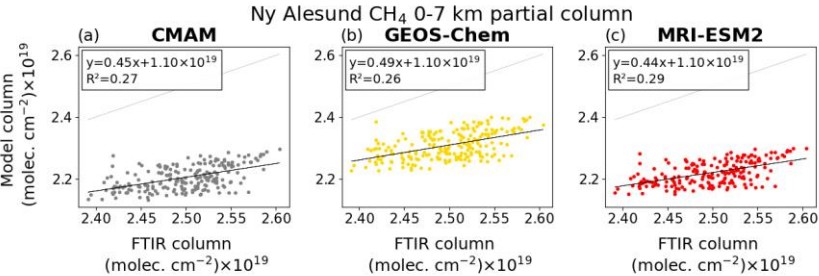

**Figure A9: Model partial column vs. FTIR partial column of $CH_4$ for Ny Ålesund, showing all available model-FTIR corresponding data. The black line is the line of best fit, where the equation and $R^2$ are noted in the legend. The 1:1 line is shown in light grey.**





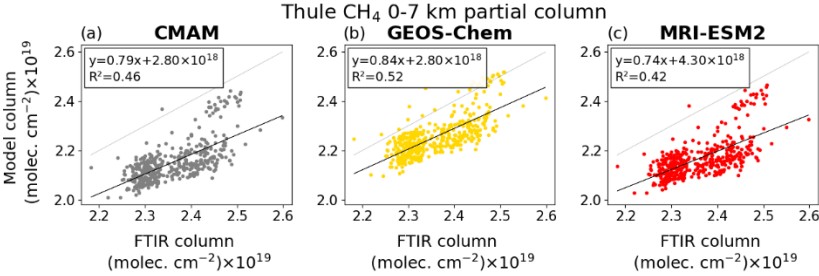

Figure A10: Same as Fig. A9 but for Thule.

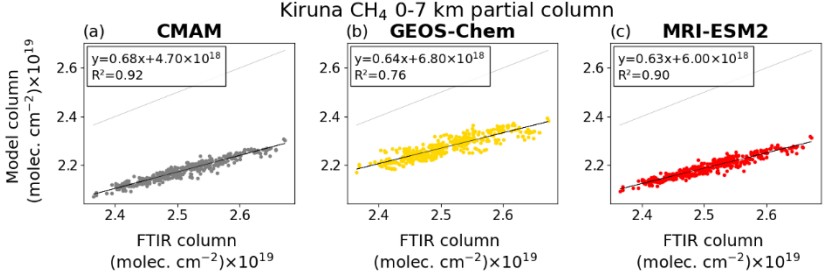

Figure A11: Same as Fig. A9 but for Kiruna.

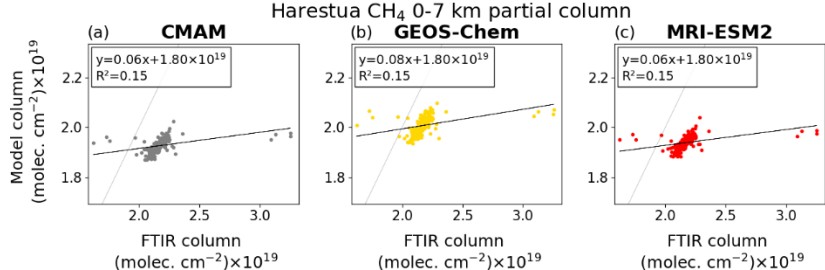

Figure A12: Same as Fig. A9 but for Harestua.



**Appendix B – Additional figures for CO**

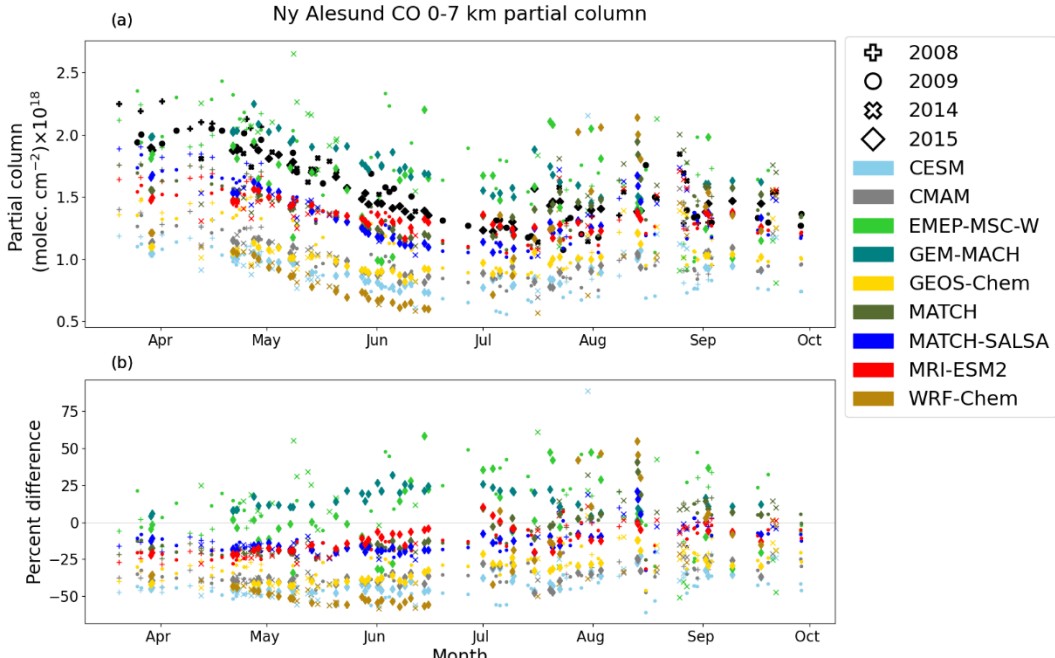

**Figure B1: (a) FTIR (black) and modelled partial columns of CO by day of year, from Ny Ålesund. Model data are the nearest in time to each FTIR measurement. (b) Percent difference from Eq. 2 by day of year. Each year is indicated by a different marker.**




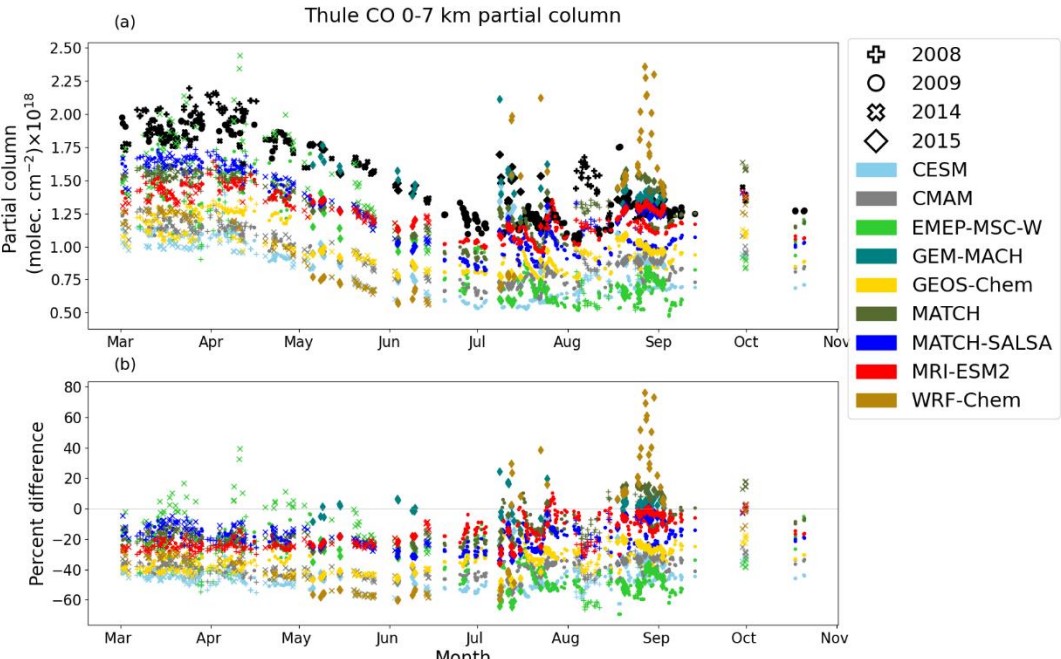

**Figure B2: Same as Fig. B1 but for Thule.**

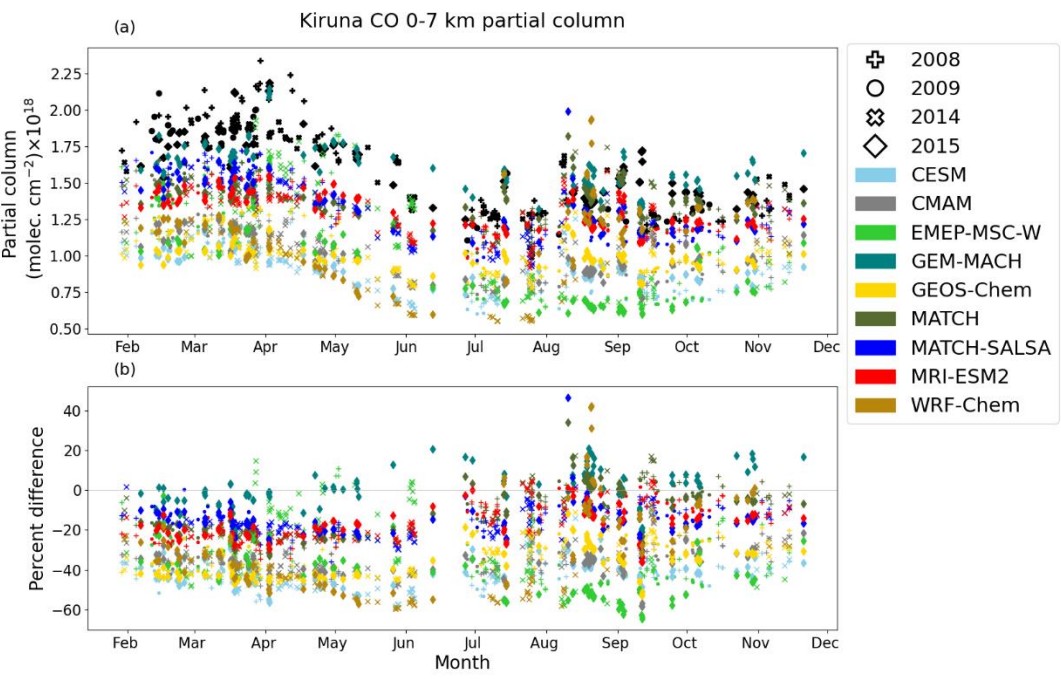


**Figure B3: Same as Fig. B1 but for Kiruna.**





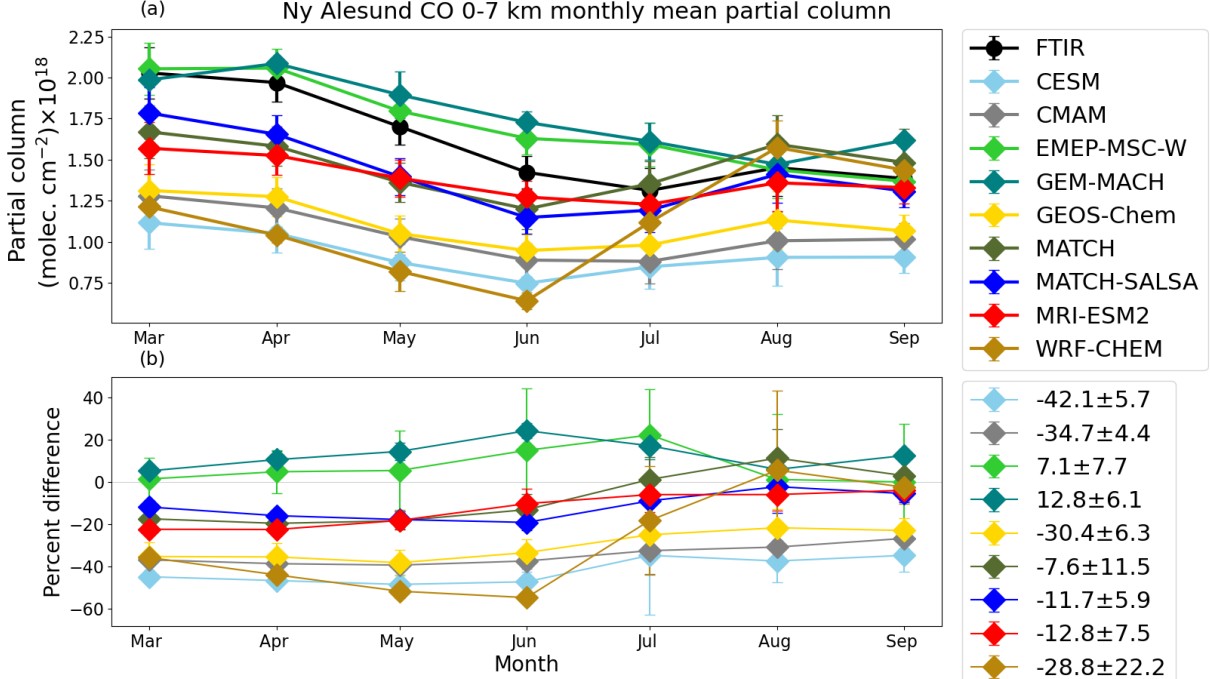

**Figure B4: (a) Monthly mean FTIR (black) and modelled partial columns of CO, from Ny Ålesund using model data that are the nearest in time to each FTIR measurement shown in Figure B1. Error bars represent the standard deviation of the monthly mean.**

**(b) Mean percent difference by month. Error bars represent standard deviation of the monthly mean percent difference. The legend on panel (b) shows the overall mean percent difference with the standard deviation of the overall mean percent difference.**



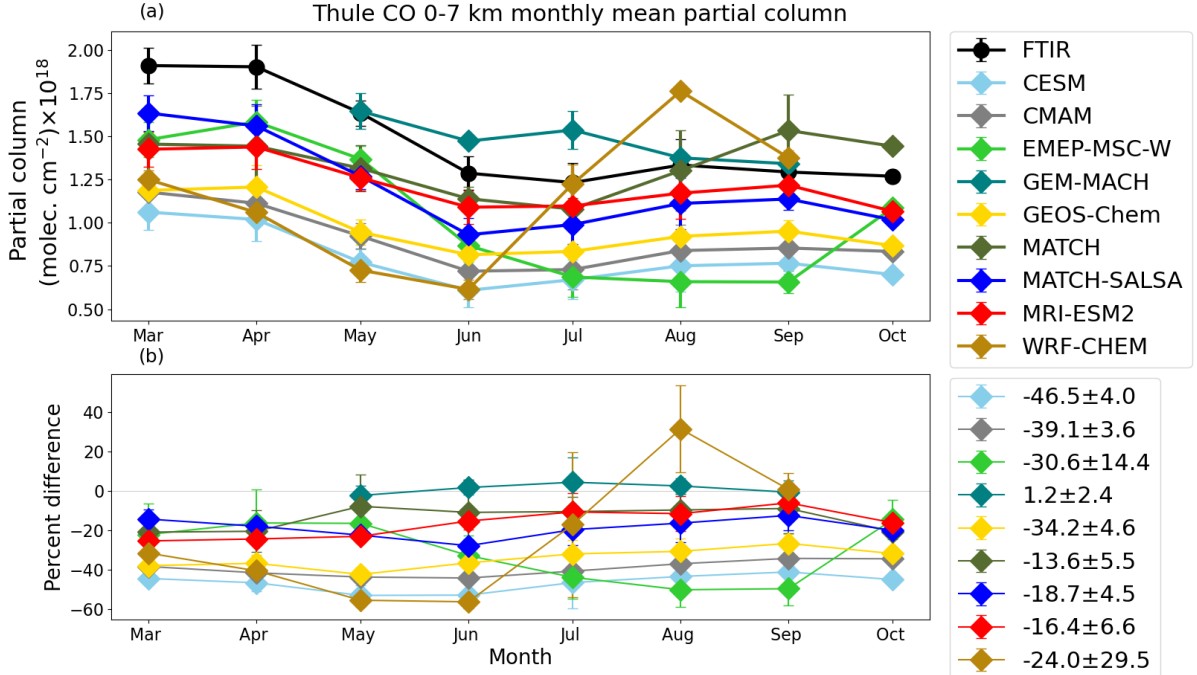

**Figure B5: Same as Fig. B4 but for Thule.**


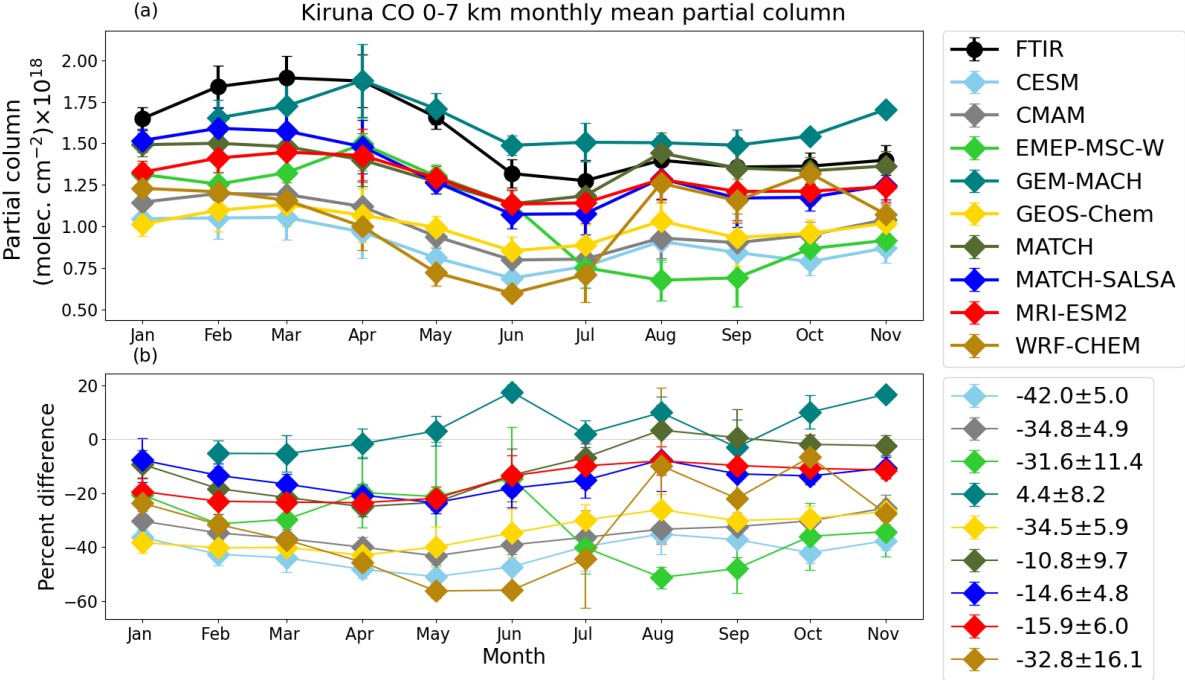

**Figure B6: Same as Fig. B4 but for Kiruna.**




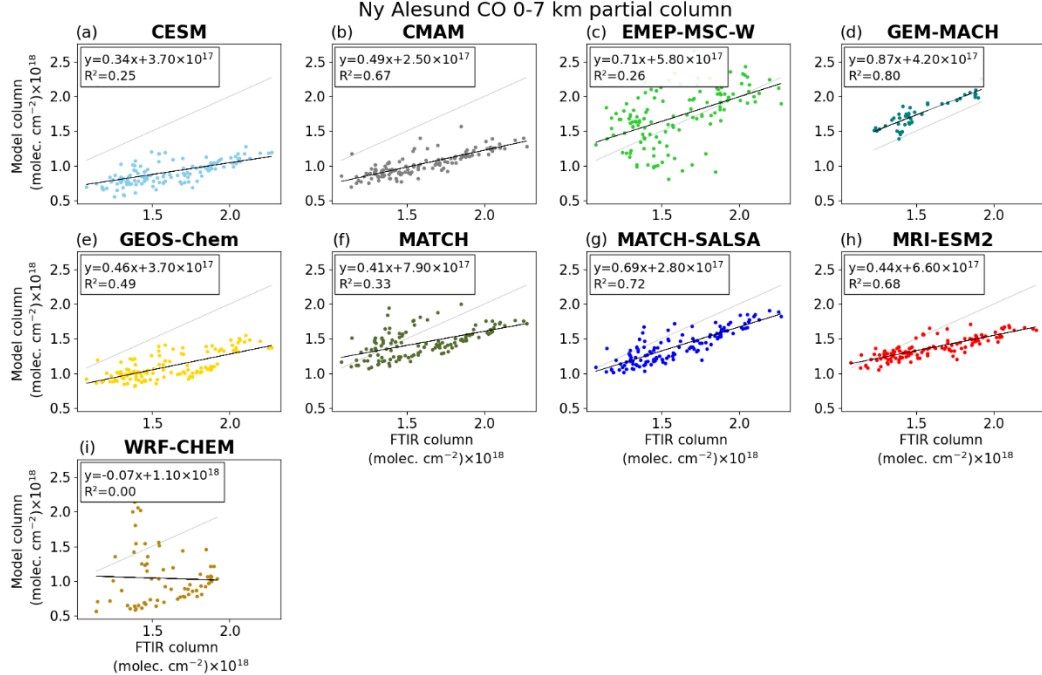

**Figure B7: Model partial column vs. FTIR partial column of CO for Ny Ålesund, showing all available model-FTIR corresponding data. The black line is the line of best fit, where the equation and $R^2$ are noted in the legend. The 1:1 line is shown in light grey.**





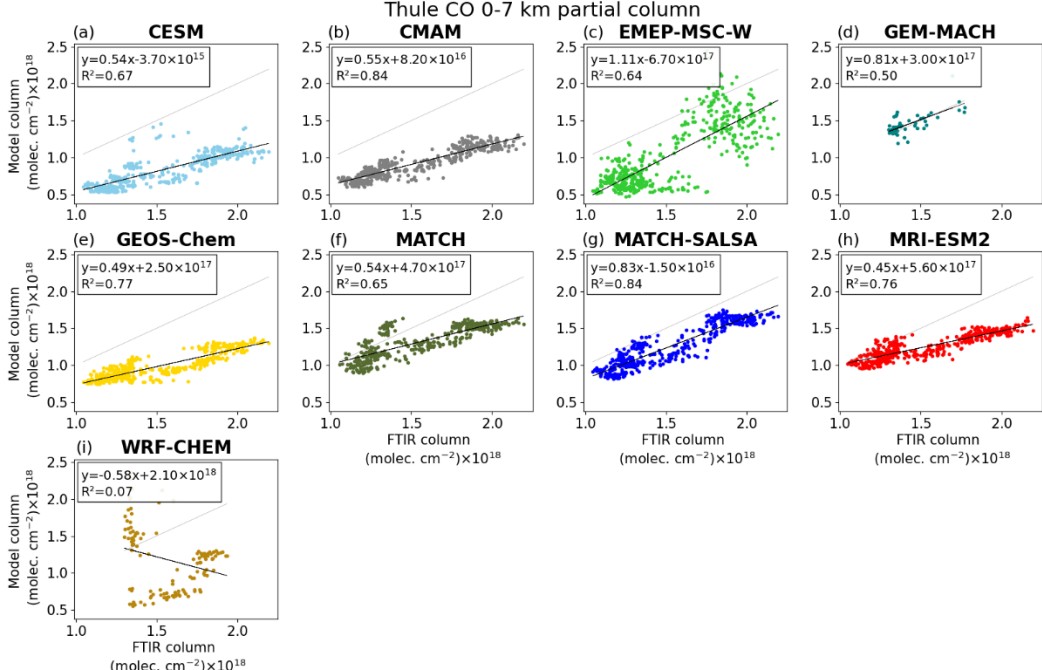


**Figure B8: Same as Fig. B7 but for Thule.**

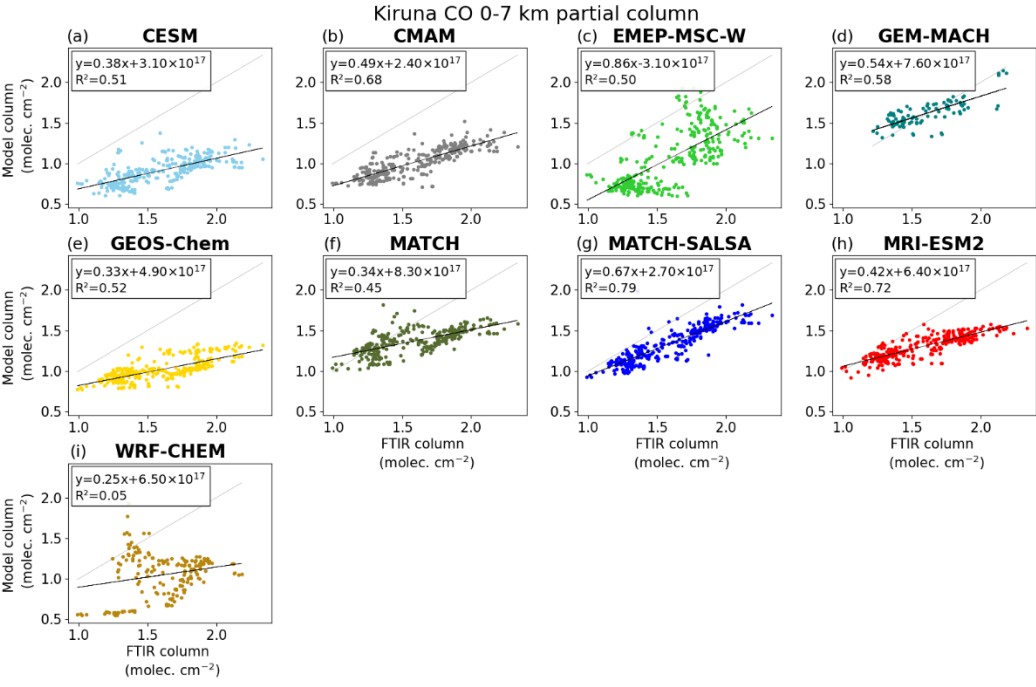

**Figure B9: Same as Fig. B7 but for Kiruna.**



## Appendix C – Additional figures for O₃

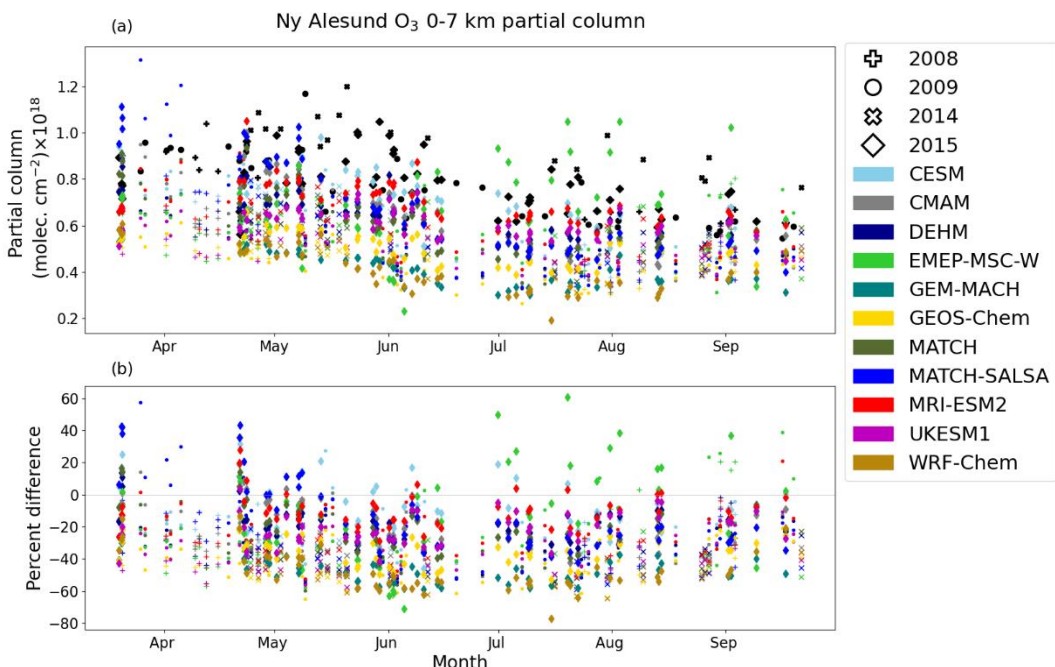

**Figure C1: (a) FTIR (black) and modelled partial columns of O₃ by day of year, from Ny Ålesund. Model data are the nearest in time to each FTIR measurement. (b) Percent difference from Eq. 2 by day of year. Each year is indicated by a different marker.**





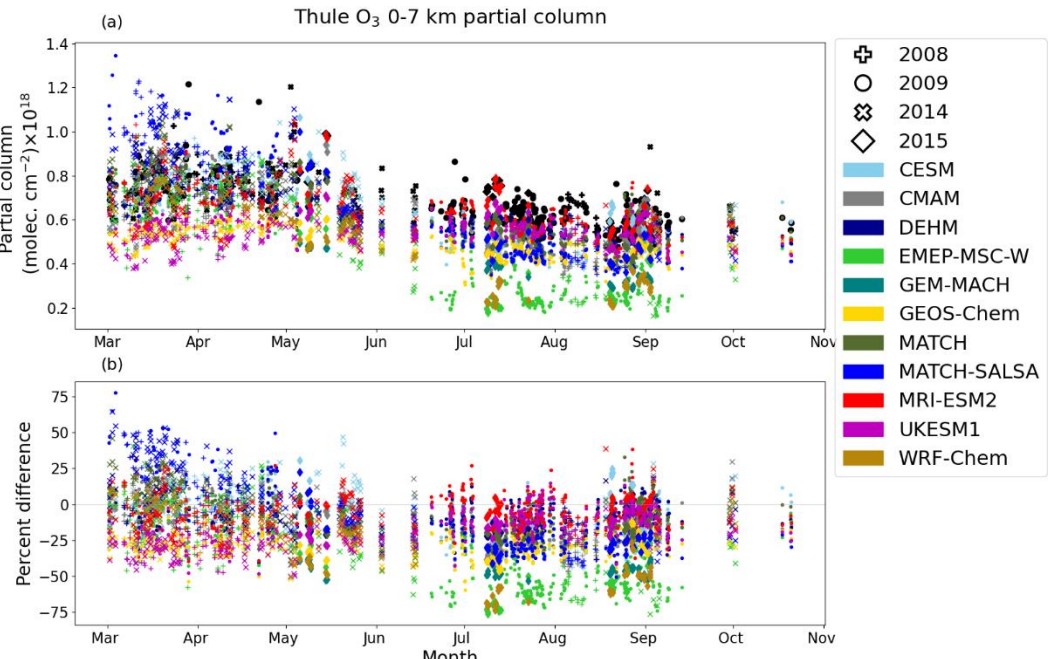

**Figure C2: Same as Fig. C1 but for Thule.**


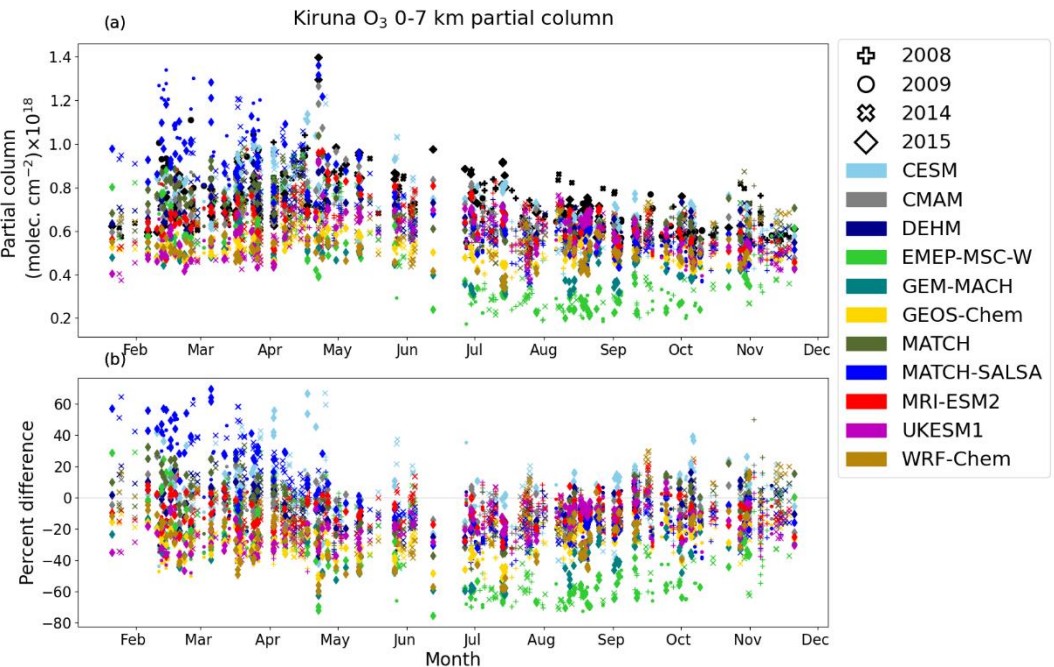

**Figure C3: Same as Fig. C1 but for Kiruna.**



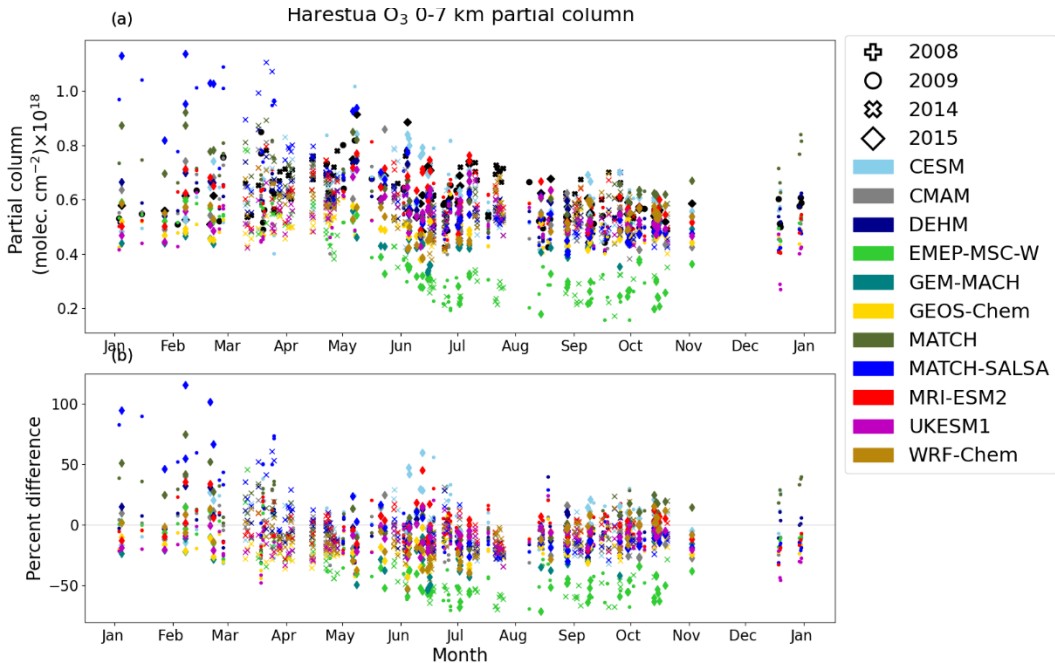

**Figure C4: Same as Fig. C1 but for Harestua.**



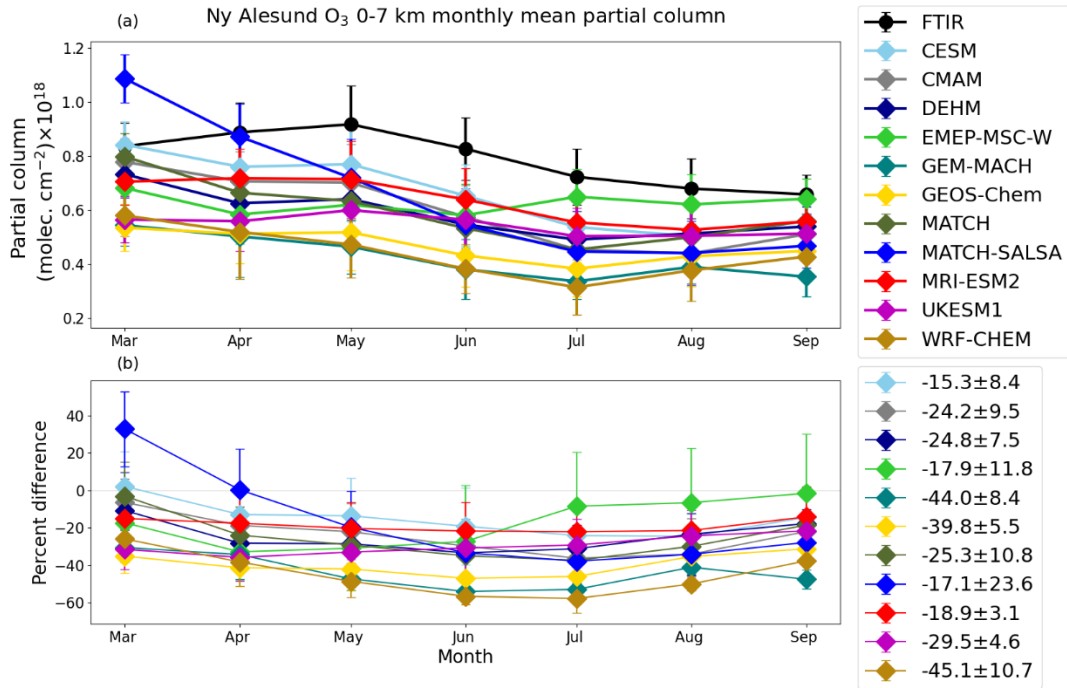

**Figure C5: (a) Monthly mean FTIR (black) and modelled partial columns of O$_3$, from Ny Ålesund using model data that are the nearest in time to each FTIR measurement shown in Figure C1. Error bars represent the standard deviation of the monthly mean. (b) Mean percent difference by month. Error bars represent standard deviation of the monthly mean percent difference. The legend on panel (b) shows the overall mean percent difference with the standard deviation of the overall mean percent difference.**





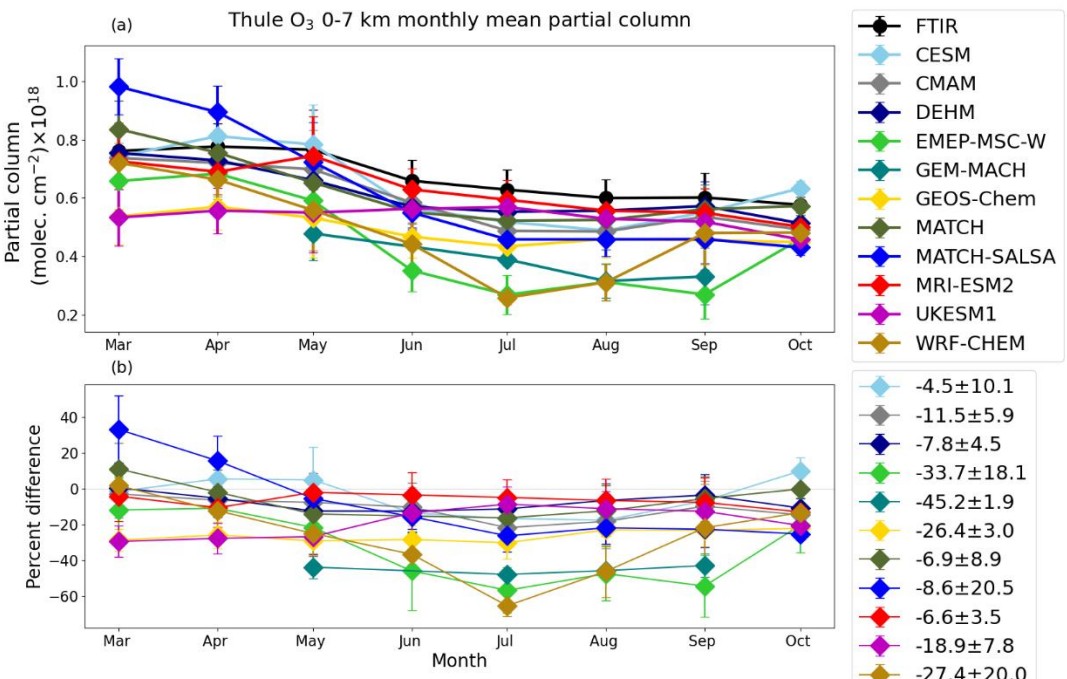

**Figure C6: Same as Fig. C5 but for Thule.**

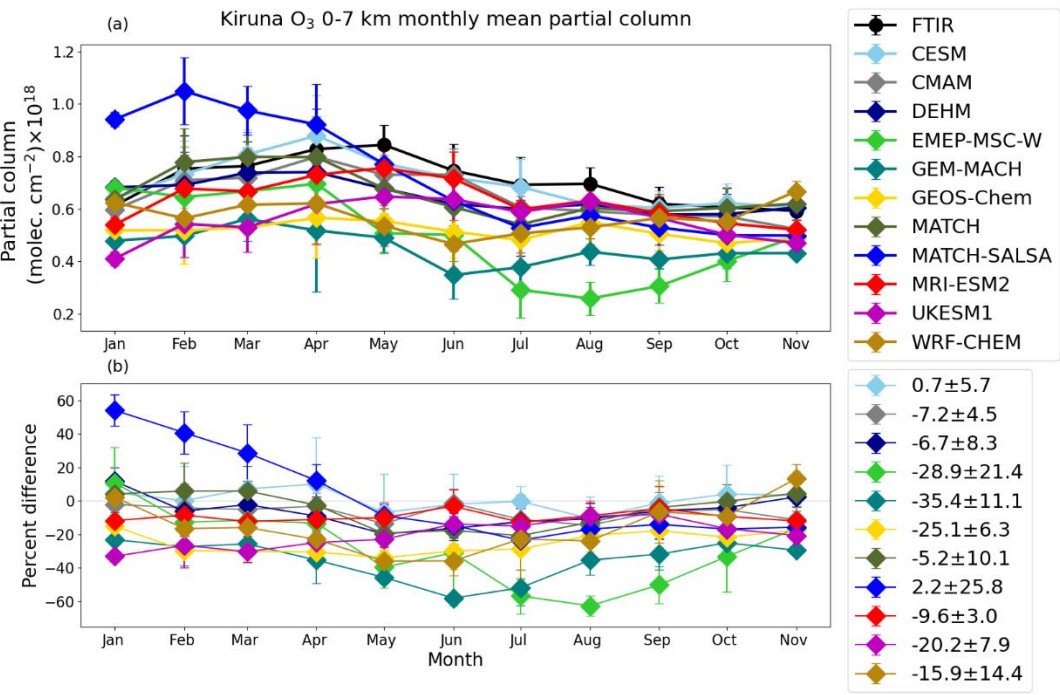

**Figure C7: Same as Fig. C5 but for Kiruna.**



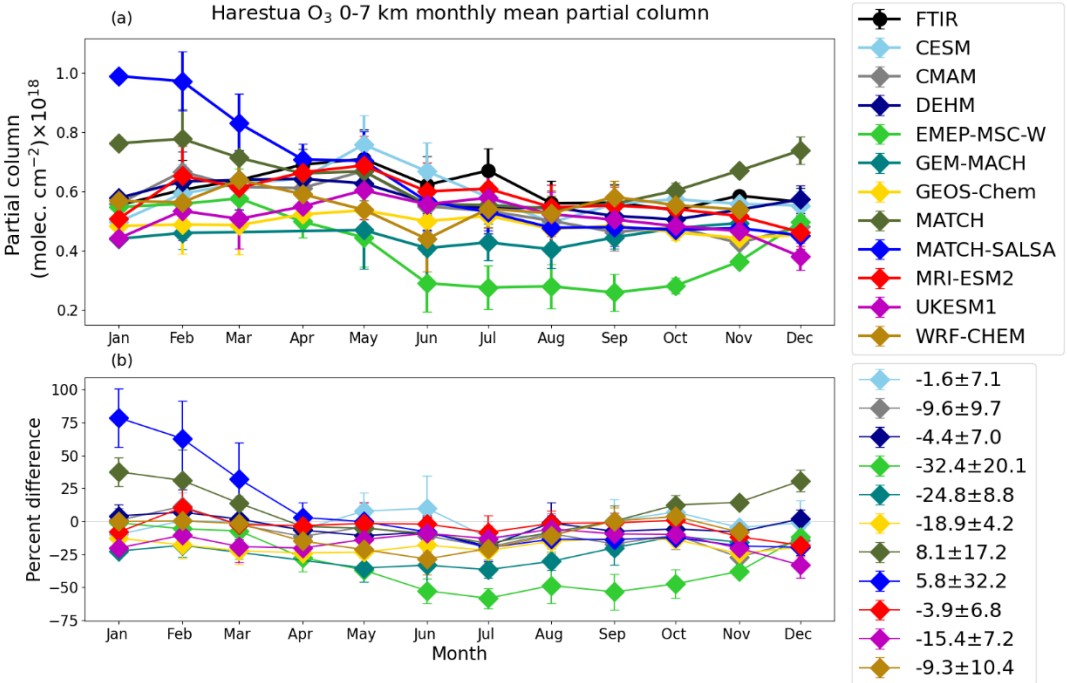

**Figure C8: Same as Fig. C5 but for Harestua.**



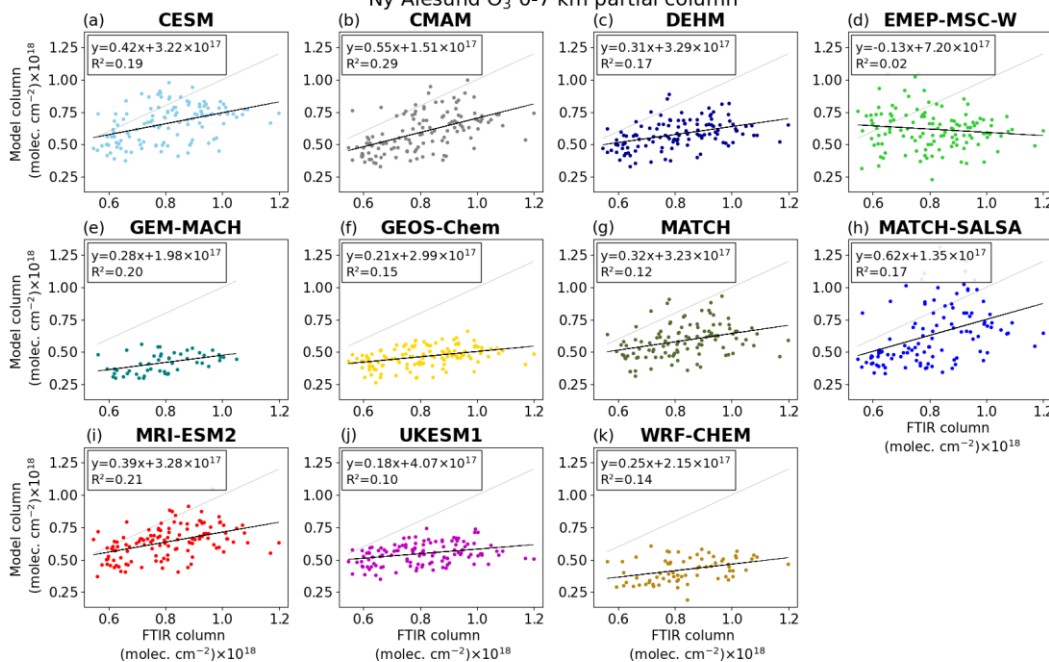

**Figure C9: Model partial column vs. FTIR partial column of $O_3$ for Ny Ålesund, showing all available model-FTIR corresponding data. The black line is the line of best fit, where the equation and $R^2$ are noted in the legend. T The 1:1 line is shown in light grey.**





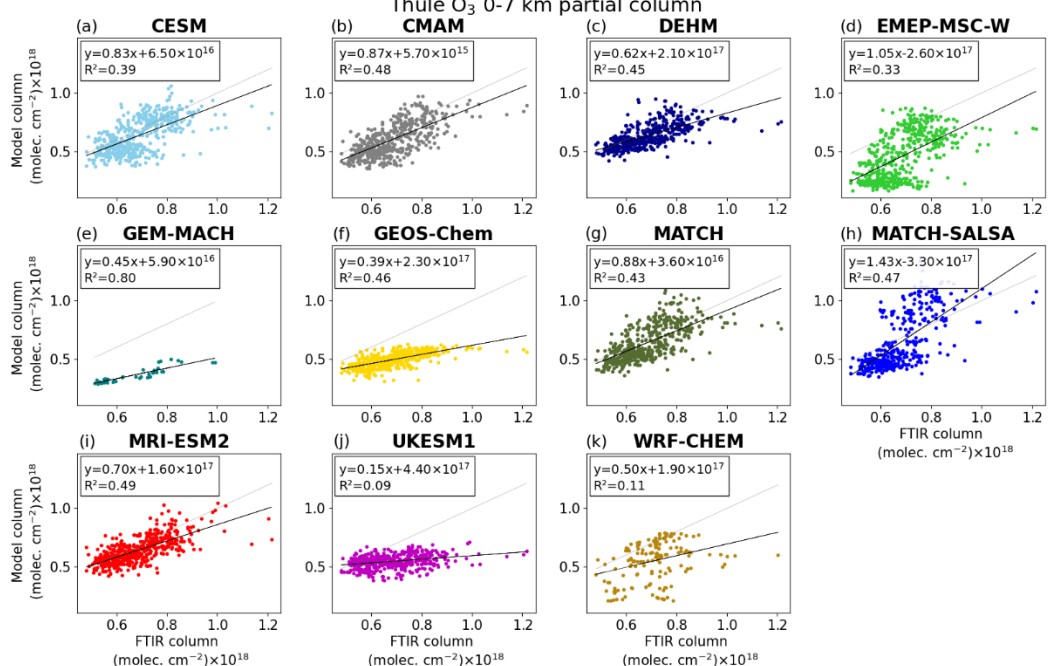

**Figure C10: Same as Fig. C9 but for Thule.**




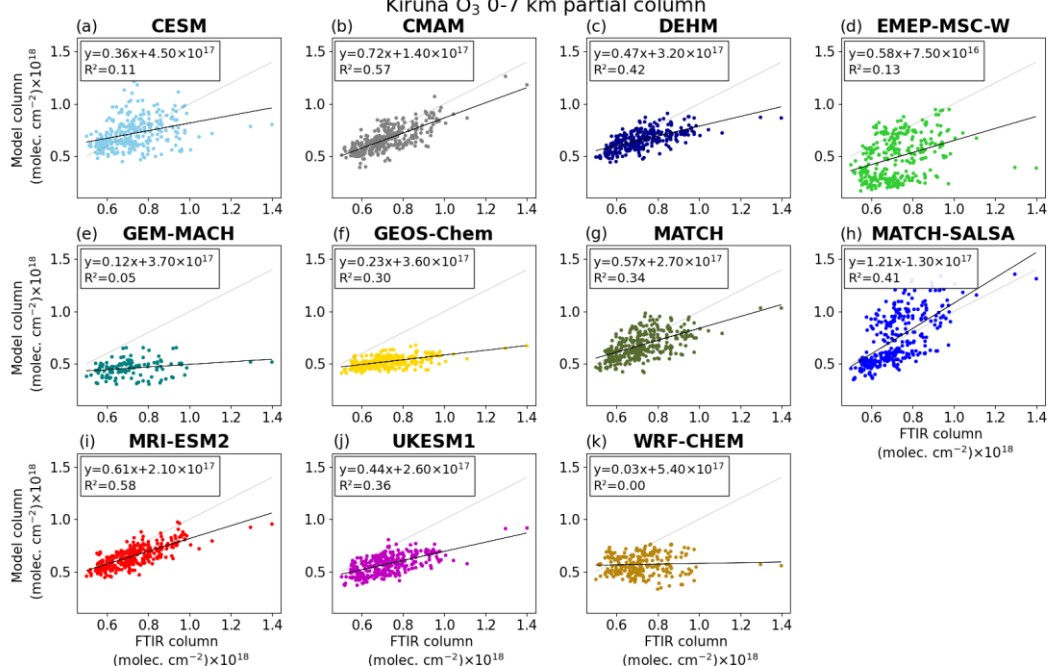

**Figure C11: Same as Fig. C9 but for Kiruna.**



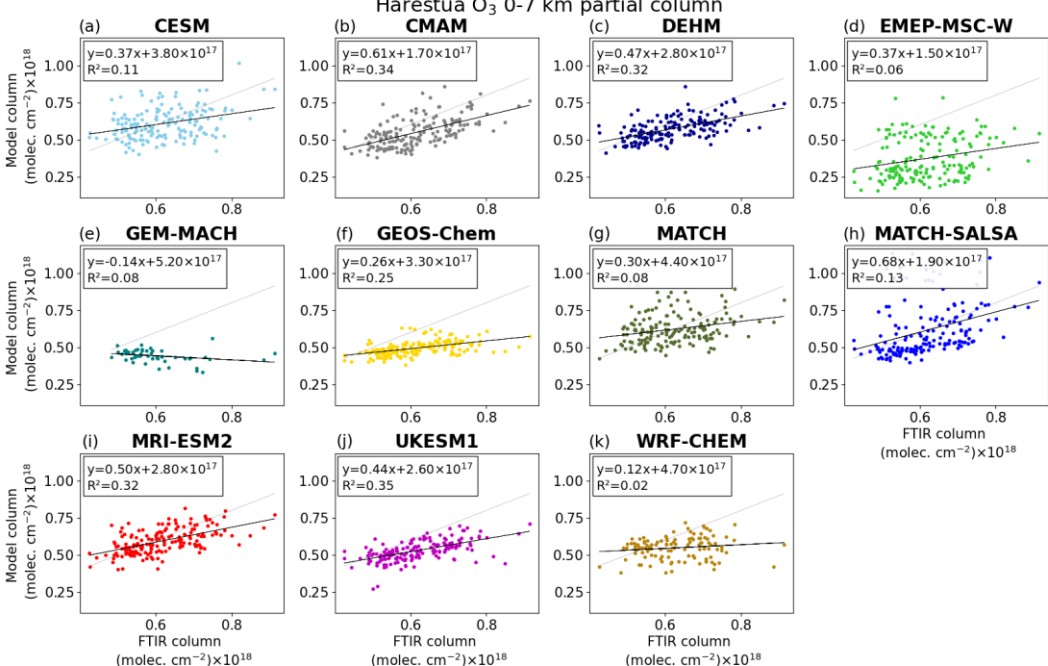

Figure C12: Same as Fig. C9 but for Harestua.





## Appendix D – MMM summary table

**Table D1: The multi-model mean percent difference for each species at each location, including the overall average percent difference for each species and the standard deviation of the mean.**


| Gas | Location | MMM Percent Difference |
|---|---|---|
| CH4 | Eureka | -9.9 ± 0.7 |
| | Ny Ålesund | -10.2 ± 0.7 |
| | Thule | -7.5 ± 2.0 |
| | Kiruna | -11.6 ± 0.5 |
| | Harestua | -8.8 ± 3.2 |
| | Average | -9.6 |
| CO | Eureka | -17.6 ± 5.6 |
| | Ny Ålesund | -16.7 ± 7.9 |
| | Thule | -24.4 ± 6.5 |
| | Kiruna | -23.7 ± 5.2 |
| | Average | -20.6 |
| O3 | Eureka | -20.1 ± 10.2 |
| | Ny Ålesund | -28.5 ± 8.3 |
| | Thule | -17.6 ± 9.8 |
| | Kiruna | -14.6 ± 8.7 |
| | Harestua | -9.6 ± 9.5 |
| | Average | -18.1 |



**Data availability**

The FTIR data are publicly available on the NDACC data repository ([https://www-air.larc.nasa.gov/missions/ndacc/data.html](https://www-air.larc.nasa.gov/missions/ndacc/data.html) (last access: 14 Feb 2023), and the model data are available at
[https://doi.org/10.18164/e0a0ac5c-d851-45b9-b6d9-4abc29d7d419](https://doi.org/10.18164/e0a0ac5c-d851-45b9-b6d9-4abc29d7d419) (last access: 14 Feb 2023).

**Author contribution**

KAW, CHW, and KS conceived this project. VF performed the formal analysis, including the comparisons between datasets, formation of plots and tables, and writing the manuscript. KS, CHW, and KAW provided scientific guidance and support throughout the work in addition to comments and edits to the manuscript. TB, JWH, JM, JN, and MP provided advice on the
FTIR data and facilitated the operations and data management for their respective instruments. SA, SB, RYC, JC, MD, XD, JSF, MG, WG, JL, KL, LM, NO, DAP. LP, JCR, MT, ST and ST provided the model outputs for the AMAP report and guidance on the analysis of models. All co-authors provided feedback on the paper.

**Competing interests**

At least one of the co-authors is a member of the editorial board of Atmospheric Measurement Techniques. The peer-review
process was guided by an independent editor, and the authors also have no other competing interests to declare.

**Disclaimer**

Publisher's note: Copernicus Publications remains neutral with regard to jurisdictional claims in published maps and institutional affiliations.

**Acknowledgements**

Funding for this work was provided by the Canadian Space Agency under the Earth System Science Data Analyses Program (grant 21SUASABBC).

The authors acknowledge the use of model datasets from the 2021 AMAP SLCF Assessment Report. The authors would like to thank all the people involved with the data collection and maintenance of the high-latitude NDACC FTIR instruments used in this study.

The Eureka FTIR measurements were made at the Polar Environment Atmospheric Research Laboratory (PEARL) by the Canadian Network for the Detection of Atmospheric Composition Change (CANDAC), which has been supported by the Atlantic Innovation Fund/Nova Scotia Research Innovation Trust, Canada Foundation for Innovation, Canadian



Foundation for Climate and Atmospheric Sciences, Canadian Space Agency, Environment and Climate Change Canada (ECCC), Government of Canada International Polar Year funding, Natural Sciences and Engineering Research Council,
Northern Scientific Training Program, Ontario Innovation Trust, Polar Continental Shelf Program, and Ontario Research Fund. We thank former CANDAC/PEARL PI James Drummond, PEARL Site Manager Pierre Fogal, CANDAC Data Manager Yan Tsehtik, the CANDAC operators, and the staff at ECCC's Eureka Weather Station for their contributions to data acquisition, and for logistical and on-site support.

We gratefully acknowledge funding from the Transregional Collaborative Research Centre TR 172 -- Arctic
Amplification: Climate Relevant Atmospheric and Surface Processes (AC)[3], project E02: Ny-Ålesund Column Thermodynamic Structure, Clouds, Aerosols, Trace Gases and Radiative Effects.  We also thank the Senate of Bremen for financial support. The AWI Bremerhaven provided logistical support for measurements in Ny Ålesund.

KIT would like to thank Uwe Raffalski from the Swedish Institute of Space Physics (IRF) for their continuing support of the NDACC FTIR site Kiruna.
MG and ST received financial support from the Arctic Monitoring and Assessment Programme (AMAP).

KSL, JCR, LM, and TO (LATMOS) acknowledge support from EU iCUPE (Integrating and Comprehensive Understanding on Polar Environments) project (grant agreement no. 689443) under the European Network for Observing our Changing Planet (ERA-Planet) and from access to IDRIS HPC resources (GENCI allocation A009017141) as well as the IPSL mesoscale computing center (CICLAD: Calcul Intensif pour le CLimat, l'Atmosphère et la Dynamique) for model
simulations. KSL also acknowledges support from French Space Agency (CNES) MERLIN (contract 7752).

MD and NO were supported by the Environment Research and Technology Development Fund (JPMEERF20202003 and JPMEERF20205001) of the Environmental Restoration and Conservation Agency Provided by the Ministry of Environment of Japan, the Arctic Challenge for Sustainability II (ArCS II), Program Grant Number JPMXD1420318865, and a grant for the Global Environmental Research Coordination System from the Ministry of the
Environment, Japan (MLIT2253).

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
