# Peer review of "Evaluating modelled tropospheric columns of CH4, CO and O3 in the Arctic using ground-based FTIR measurements"

_EGUsphere, 2023_

## Referee Comment (RC1)

**Overview:**

Flood, et. al. has submitted a manuscript comparing ground-based mid-infrared FTIR measurements of tropospheric patrial column O3, CH4 and CO at five arctic sites to 11 model simulations. Daily and monthly averages are compared. Comparisons are conducted for the years 2008, 2009, 2014 and 2015 with the aim to test and comment on model validity, i.e., can the models reproduce the measurements, and if not, why?

The authors build upon prior University of Toronto research into Arctic atmospheric composition using FTIR data. The novelty of this study is that it is the first-time tropospheric patrial column O3, CH4 and CO measurements at the five arctic sites have been compared to this suite of 11 models. As mentioned in the manuscript, in situ and satellite data have been used in past studies to evaluate the performance of these 11 models, but not the FTIR datasets. The FTIR datasets provide an integrated partial column abundance that is quite different in footprint (spatial and temporal) and altitude sensitivity to the datasets used in previous studies hence bringing a new product to assist in model evaluation. This manuscript illustrates the benefits of using such partial column data in model evaluation and should be viewed as another standard dataset (along with in situ and satellite remote sensing) in future model comparison activities.

The manuscript is logically structured and well referenced. The writing is clear and ,in most instances, unambiguous. The analysis is robust and easily understood. The content is well within the scope of this journal. Information on data availability is given. The single conflict of interest is minor, stated up front and will be easily dealt with by the journal editors.

I recommend publication of the manuscript after some changes in the manuscript to mainly improve the clarity of content and the context of the investigation.

**General comments:**

G1/ The Introduction needs more detail to set the context of this research.

In the introduction, the AMAP SLCF assessment report (2021) was used as the basis for setting the context of this research into Arctic SCLFs and the importance of model validation. It is only in subsequent sections that the priori model validation work within Whaley, et al (2022 & 2023), Emmons et al. (2015) and POLARCAT/POLMIP were mentioned. Such past studies should be mentioned in the introduction to assist the reader in knowing where this current study fits in and what this study is to achieve that the past studies did not. It is only at line 89 where a single sentence states the aim of the study: "This study builds upon the model-measurement comparisons presented in the 2021 AMAP SLCF Assessment Report using an additional Arctic dataset that was not included in the original report.". I view this current research as a natural extension of the work by Whaley, et al., 2022, but using a new dataset (FTIR site measurements) with a different temporal and spatial footprint to that of the in situ and satellite measurements.

The paragraph starting line 75 which introduces the FTIR measurement dataset should also be expanded to give examples of how such measurements from these 5 arctic sites have been used in past Arctic model validation studies. As it currently reads, it is unclear if this is the first time ever such measurements have been compared to models.

I think these changes will be easy to instigate and hopefully improvement context of this current research.

G2/ There is no mention of why column integrated measurements are used to validate/compared to the model simulations. This is one of the main novelties of this study.

At line 518 there is the statement: "NDACC FTIR spectrometers were selected for this project because of the wide range of species measured, high spectral resolution, multiple high-latitude sites, and publicly available data ", which seems the main justification of using the FTIR data (along with a brief contextual reference at line 72: "All of these factors lead to a scarcity of monitoring stations and a limited representation of atmospheric vertical information").

I think these are secondary reasons, the main reason being a (partial) column integrated data product that has a spatial and temporal footprint which is more presentative of the tropospheric free atmosphere than in situ and satellite measurements.

I recommend adding a statement (in the Introduction) focusing on the benefits that validating models using partial column data (that FTIR can provide). The advantages and disadvantages of using column integrated data needs to be explained and how such data allows comparison to models in  way in situ and satellite remotely sensed data cannot. It fills a gap.

G3/ It would be good to explicit state why CH4, CO and O3 were the selected species as both the models and measurements have other SLCFs products for which comparisons could be performed.

G4/ There is no mention why the four selected years (2008, 2009, 2014 and 2015) were chosen for the comparison activity. Reason/s why these years were selected need to be stated. Two other points should also be addressed: given this manuscript was submitted in 2023, why was the most recent year 2015? and why long-term trend comparison analysis , i.e., a continuous time series period, was not performed. I suspect model simulation temporal constraints, but this should be stated.

G5/  The partial column range used in comparisons is ground level to 7km. A prior study used 0-9km (Wespes, et al. 2012). Please state why the 0-7km range was selected.

G6/ There is no mention of the tropopause heights at the measurement sites. Even if the selected partial column upper boundary (7km) is less than the tropopause height, the averaging kernels might indicate 0-7km partial column measurement sensitivity to above the tropopause. How would it effect model measurement comparisons? Are stratospheric intrusions of major concern?

G7/ Reorganization of site-specific figures.

For CH4: seasonal daily and monthly time series plots along with daily model measurement scatter plots are given for a single site, Eureka, i.e., figures 3,4 and 5 and the other four similar FTIR site data plots are in the appendix. Then all site data/metrics for CH4 are displayed in figures 6 and 7. This is repeated for CO and O3.

I found I was continually being referred to figures in the appendix, especially when it came to interpretation of the model measurement results at the end of each species section. I would like the authors to consider rearranging figures. I suggest that all the daily/individual measurement plots, e.g., figures 3,8, 13 be moved to the appendix. The monthly plot , e.g., figure 4 include all stations, in a 2x3 panel plot.

Also, figure 5 to include all stations. For CH4 this would be a 3x5 panel plot.

For figures 10 and 15 I don't think all stations scatter plots can be plotted is a reasonable way in a single figure , thus still relegated to the appendix. If the author can think of another way to concisely display the mentioned data in the main body of the manuscript it could be worth investigating.

G8/ Analysis interpretation of CH4.

Compared to CO and O3, the discussion and interpretation of the CH4 partial column measurement model comparison results are very short. Example: line 266 "satellite 265 instrument and finds that the models are biased low in the vicinity of the tropopause (300hPa) (Whaley et al., 2022)." What height is 300hPa? How much biased low? Is this expected? acceptable?

Please expand and include a greater discussion of the results in comparison to findings from Whaley, et. al., 2022, esp. in context of surface CH4 in situ measurements.

**Specific comments:**

S1/ Following on from G1, the first two sentences in the abstract could be improved. They currently do not add any specific information about this study. The abstract should also mention the novelty of this study, i.e., what has been done here that hasn't been done before.

S2/ line 51. "…causing most of the pollutants to remain predominantly localised" but throughout the manuscript there is multiple references indicating long range transport [of pollutants] (as at line 370) are a possible cause of measurement model differences. Can this disparity be rectified.

S3/ line 125. Degrees of freedom and the partial column averaging kernel (PC AVK): Could figure 1 altitude range be expanded to ~ 20km to see 'what happens above 8km'. If the PC AVK above 7km is ~ 1.0 this means retrieval information above 7 km is incorporated into the 0-7km PC. If so, please comment upon, and implications of.

S4/ line 124 and Table 2. Please expand commentary and implications for DOFs < 1.0. For CO and O3 the PC DOFs are ~1.0, but for CH4 the DOFs are < 1.0, and from figure 1, there is less sensitivity to near surface CH4. What are the implications of this for model comparisons?

S5/ Section 2.2. Relative to other manuscripts the section describing model simulations is brief, but I think it is justified as detailed model descriptions (and forcings) are given in Whaley, et. al., (2022). I see no need to repeat information that is already readily available.

Could the authors make sure that any model output that is used in this current study that differs from model output used in the study by Whaley, et al., (2022) be stated and the reasons for the change (e.g., an updated model or forcings) also be stated. This may seem a logical statement, but if the authors are going to heavily defer to Whaley, et al., (2022) to provide details then it is very important there are no changes or changes are identified.

S6/ The first two sentences in section 3 are not needed, as it is covered in the section 'Data Availability', or if the authors want to retain it in the manuscript, then relocate to section 2.

S7/ Figure 2. The flow chart alludes to that the 'nearest' model grid point (to a measurement site) is used. This should be mentioned in the text. To clarify, is there any spatial weighting of localised grid points? I.e., weighting/kriging of the closet model points/cells to the FITR location? Have tests been done concerning a geolocation weighted average model value? I gather any differences will be minimal but would be good to confirm, even if for a single site.

S8/ I think another paragraph is needed at the end section 3 concerning the type of analysis that is going to be performed using eqn. 1 and 2 as the quantification metrics. Are you going to investigate, diurnal, daily, monthly, or seasonal differences? Long term trends? Basically, what are you going to look at.

S8/ Best line fits: linear regression. Do the best line fits in all the analysis also take into account the uncertainty in the abscissa (measurements) as well as the ordinate (model)? If so, please state so, if not, then maybe prudent to perform a few tests to assess the effect on the linear fit. Since measurement and model uncertainties are of comparable magnitude, abscissa uncertainty could have a large effect on the calculated linear fit.

S9/ line 244:

"For all models, the R2 values for Ny Ålesund and Harestua are significantly smaller, while the overall mean percent difference is comparable to the other locations. The discrepancy is likely attributed to the smaller number of measurement points, causing outliers to have more weight in the linear regression, which is reflected in the elevated NRMSE for Ny Ålesund across all models."

I do not think a lack of lack of measurement points is a cause. Both Figs A9 and A12 show there are plenty of data points. Fig A12 clearly shows there are outlier measurements at Harestua. I would attribute this to either measurement/retrieval error that was not filtered out thus should be removed from the comparison datasets, or anomalous atmospheric events which if at fine temporal or spatial scale the models would be able to reproduce, thus this measurement period should also be omitted as the model would not be able to replicate it. Given that the anomalous measurements are both too high and too low I suspect measurement error. I recommend omitting such outliers (across all data sets, unless it can be accounted for) using a defined filtering method and perform analysis again.

This approach will not account for the low R^2 at Ny Ålesund and I cannot easily see why the R^2 is lower than at other sites.

S10/ line338: "Similar trends have been found in other Arctic model-measurement comparison studies." Please reference this statement, also do you mean trends or findings? As temporal trends are not investigated in this study. I think would also be helpful to quantitatively state the amount of underprediction in prior studies and then relative to this study (referring to table D1 would be a good idea when comparing the results from this study to prior studies ).

S11/ line 355: "Further, the tracer investigation shows that OH differences account for more variability between the models than the transport mechanisms within the individual models."

Could this statement please be referenced.

S12/ line 366: "The results of the model-FTIR comparisons presented here support this reasoning, as the model with a positive bias (GEM-MACH) has a different emissions input, with possibly more complete emissions in the Arctic, as this was a high-resolution Arctic version."

This conjecture could be quite easily solved by looking at the model simulation parameters to see if this is true.

S13/ line 381: "In addition to atmospheric chemistry, its production is highly sensitive to meteorological conditions. Therefore, it is difficult for models to accurately simulate tropospheric O3." Ozone also can have a significant diurnal cycle due to photochemistry, complicating comparisons when measurements and model differ in time. Please include this cause as well.

S14/ line: 452. " To supplement the aircraft and satellite campaigns undertaken for the POLARCAT study, daily mean O3 measurements from the FTIR instruments at Eureka and Thule were compared to MOZART-4 simulations in Wespes, et al. (2012)".

Due to the daily diurnal cycle of ozone, comparisons of daily FTIR averaged ozone measurements would be biased high to model output (that uses daytime and nighttime values as I gather nighttime FTIR measurements are not taken). Can you confirm daily average MOZART ozone was used or matched to FTIR measurement times.

S15/ line 518:

"NDACC FTIR spectrometers were selected for this project because of the wide range of species measured, high spectral resolution, multiple high-latitude sites, and publicly available data."

As stated in G3, a better reason for using FTIR datasets should be given. This relates back to a general comment of the overall benefits of using column integrated measurements.

S16/ Defining FTIR uncertainty. This term (or variations of) is found within the text (e.g., lines 248, 323, 514) but not clearly defined. Is it the uncertainty of individual measurements as in table 2 , or the 1-sigma standard deviation of the daily/monthly measurement means?

S17/ The table 4 caption states:

"Summary of mean percent difference for each model and location by species. MMM is the multi-model mean. The colour scale indicates the mean percent difference relative to the FTIR measurements, from blue (-50%) to red (+50%). A square marker indicates that the mean difference is within the FTIR uncertainty. A triangle marker indicates that the mean difference is within the FTIR uncertainty combined with the standard deviation of the monthly mean percent difference."

It is difficult to understand what is being compared (and significance of the metric ) when FTIR uncertainty is not clearly defined. Is FTIR uncertainty the monthly measurement 1-sigma S.D. or the uncertainties of a single measurement as given in table 2?

There is no explanation of why a double metric is used, could this be explained in the text. What does it mean if "the mean difference is within the FTIR uncertainty" but not "within the FTIR uncertainty combined with the standard deviation of the monthly mean percent difference".

S18/ Table D1: Is an important table. I recommend putting this in the main body of the manuscript and referred to in each species section.

**Technical comments:**

T1/ line 81. Arctic is not defined, are you implying >60N? Maybe define what 'Arctic' is.

T2/ Table 1 and Table 2 colour key columns are not needed.

T3/ Paragraph starting line 114 concerning technical details about the FTIR data and retrieval strategies. I think there is a need to mention the vertical grid spacing of the retrieval, i.e., how many layers, esp. in the troposphere, and from 0-7km.

T4/ Figure 1. The term 'mean column'. Do you mean total or partial (0-7km) column? Please make this clear in the label. If it is total column, then I recommend replotting as 0-7km partial column.

T5/ Figure 1. The abscissa axis (Partial? column AVK) needs units. [unitless] or [relative] would suffice if not [ppb/ppb].

T6/ line 166. (+/- 1.5 hours): I think it needs to be explicitly stated why this time frame was chosen (from previous model comparison studies?), just to make it clear why , say , +/-24h cannot be used. A tight time constraint is required for ozone due to diurnal photochemistry.

T7/ line 173. Partial column averaging kernel I gather? Maybe add 'partial column'.

T8/ line 176. "ratio between the trace gas VMR and layer airmass (molec cm^-2)". Best to add the term 'layer airmass' for clarity.

T9/ line 180. To clarify, is the station altitude is also used as the lower model partial column layer boundary in analysis? If so, then I think this needs to be stated.

T10/ line 206. Please replaced 'important' with a more specific descriptor. Important is too subjective (important in what context?).

T11/ line 213. 'concentration' should be replaced with 'partial column'. The models are forced with concentrations (vmr), but the quantities under investigation are partial columns (molec. cm^-2).

T12/ Line 217. 'Little variance'. Sorry, I found this unclear, do you mean between the models or intra-model (within a month or day)?

T13/ line 219. 'uniformity', of what?

T14/ Figure 3, 8 and 13: The measurement symbols are extremely hard to differentiate between. Can you make them easier to differentiate?

T15/ All figures: In all the figures, when model data is plotted, I gather it is modelled smoothed partial columns? If so, please add 'smoothed' to all 'model data' just to make it clear.

T16/ line 380. "It is a secondary pollutant" replace with "In the troposphere, ozone is a secondary…"

T17/ line 395. "However, the FTIR O3 seasonal cycle does not have a springtime minimum from surface ODEs, as one might expect from surface measurements". Sorry, this does make sense. As it reads FTIR partial measurements are surface measurements? Can you please rewrite to make it clearer.

T18/ line 422. remove the word 'difficult'.

T19/ line 425. remove 'and as such recommended for future work'. I can understand what is trying to be conveyed, but nearly instance of a model measurement disagreement warrants future work.

T20/ line 497. remove the word 'historical'.

---

## Referee Comment (RC2)

**General comments:**

The manuscript by Flood et al. evaluates the accuracy of model estimates of partial atmospheric columns of CH4, CO, and O3 in Arctic regions by comparing against FTIR ground-based measurements from five northern high latitude sites in the Network for Detection of Atmospheric Composition Change (NDACC). The study considers 11 models and expands on previous analysis published in the Arctic Monitoring and Assessment Programme (AMAP) 2021 assessment report. That report included comparisons of 18 models to surface in situ, aircraft, and satellite-based measurements, but did not compare to ground-based FTIR measurements of partial column concentrations. Time-series of FTIR measurements and model estimates, correlations between FTIR measurements and model estimates, comparisons of seasonal cycles, and summarizing statistics of comparisons of model estimates and FTIR measurements are presented for each NDACC site. The site Eureka is the primary focus of most of the discussion in the paper, while comparisons at the remaining NDACC sites are mostly presented in the Appendices. There is also a balanced discussion of how the results of the comparison may or may not support previous analyses of this type.

Overall, the value of this analysis is clear. Comparing model estimates of greenhouse gas concentrations to FTIR retrievals is useful and provides a wealth of additional information that cannot be provided by comparing to surface-based measurements, and with much greater temporal frequency than aircraft-based measurements. Furthermore, this analysis is thorough and the interpretation of the results is reinforced by comparisons to previous literature. That being said, there are critical ways that the paper could and should be improved before publication. First, while references for each model considered are provided in the paper, the differences in model frameworks, assumptions, boundary conditions, and set up are not discussed much in the paper. The authors seem to expect the reader to conduct a high degree of background research or already have a very thorough understanding of all of these models. This is particularly relevant in the failure to state the spatial resolutions of the models being used and the failure to sight which models do or do not simulate the stratosphere (particularly in the context of the O3 discussion). There is certainly some discussion of factors that may be driving specific model biases, but these could be discussed more in the context of specific models. In addition, I think that some generalizations or conclusions are not as consistent with the plotted results as the authors suggest; however, many of the plots are difficult to read and I think reformatting some of the figures would really help.

**Specific comments and suggestions:**

In general, the paper would greatly benefit from a reorganization of how data is presented in the figures. Specifically, Figures 3, 4, 8, 9, 13, and 14 would be better presented as separate panelled subplots for each model, similar in format to Figures 5, 10, and 15. As they are, these figures are very difficult to read with many overlapping lines that make it hard to differentiate the behaviour of individual models.

Line 84, why are you only considering 11 of the 18 models used in the AMAP report?

In section 2, line 109, you say that SFIT4 is used to retrieve VMR profiles from NDACC FTIR measurements at all sites except Kiruna, which uses PROFFIT. There needs to be some discussion of how these two retrieval algorithms differ and how this may impact the resulting retrievals. Alternatively, you can cite a paper that compares the two. It seems unlikely that using different retrieval algorithms would not have some effect. Furthermore, in section 2, you should state what the typical temporal frequency of NDACC FTIR observations are. This becomes relevant in section 3 because it is

useful to know approximately how many FTIR observations are included in the 3-hour averages that are compared to model estimates. Similarly, some information about the spatial resolutions of the models being considered and the distances between the NDACC sites and the referenced location of the model estimates is important. Do all the model data have the same spatial resolutions and do distances between site locations and model estimates vary among sites or among models? Could any of this variability explain differences seen in the model comparisons?

In Section 3, specifically the flow chart in Fig. 2, since the FTIR measurements are collected with greater temporal frequency than the model estimates, it would seem that the temporal matching would involve finding the FTIR measurements closest in time to model estimates, rather than the other way around. Also, please clarify whether averaging kernels and a priori values are retrieved for each FTIR observation or based on a general reference for the instrument. If they do vary with each observation, how is this handled when applying the corrections.

It seems like the second paragraph of section 4.1 belongs in methods. By extension, it would make sense to talk about how many models you are including in the comparison analysis for each gas earlier in the paper (in section 2 or 3), though the gases covered by each model are shown in Table 3 the number of models that estimate each gas should be summarized in the text as well.

Can you speculate on why GEOS-Chem has lower % differences (better accuracy), but also lower correlation coefficients (poor precision or more scatter) in the CH4 comparisons?

Line 262, you say that the FTIRs show good sensitivity to surface CH4, but Fig.1 shows that the instruments are more sensitive to higher altitudes in the partial column than they are to the surface. I also wonder if variations in the tropopause height or a poor representation of this in the models or in the FTIR retrievals could affect the accuracy of your CH4 partial columns in the comparisons.

In section 4.2, could the increased discrepancies between FTIR retrievals and model estimates of CO in spring also be at least partly explained by errors or biases in the FTIR observations due to low solar zenith angle or cloud cover?

Line 294, I think the seasonal shifts in bias for EMEP-MSC-W and WRF-Chem are more remarkable than for MATCH (at least at Eureka, which seems to be the site that the results discussion is primarily focused on), but these are not mentioned.

On page 17, when discussing Fig. 11, why not comment on the fact that WRF-Chem has correlation coefficients near zero and very high NRMSE relative to the other models?

Line 353-355, is this describing a global effect in which European emissions have a greater influence on surface CO everywhere, or were the studies conducted in Europe and the surface CO is more affected by local emissions?

In section 4.3, please clarify which months are included in "springtime". It seems that most models agree better relative to other models as well as FTIR in February and March than all other months except September. If these months are part of springtime that does not support your claim that springtime O3 concentrations are poorly characterized in the models. Furthermore, if April is part of springtime, WRF-Chem should be mentioned on Lines 400-401, along with UKESM1, GEM-MACH, and GEOS-Chem.

Line 401-402, please elaborate on the reasoning behind this conclusion ("may be attributed to a low bias in the models' lateral boundary condition, inaccuracies im model water vapour and/or a lack of O3 transported from mid-latitudes.").

Line 403-404, why is MRI-ESM2 not mentioned here? That model seems to track very well with the FTIR measurements in Figures 13 and 14.

Line 409-410, this claim does not seem to be as consistently relevant for O3 as it is with CO. There are a number of models, including EMEP-MSC-W, DEHM, and CESM, that appear to exhibit a high degree of scatter for lower O3 partial column concentrations (at least for Eureka).

Line 416, if the Wespes et al. study mentions stratospheric influence as a major driver in tropospheric concentrations of O3, why is there no further discussion of which models in the current study simulate the stratosphere and how this may or may not influence errors in the partial column model estimates when compared to FTIR measurements?

Line 441-444, it should also be mentioned that WRF-Chem and GEM-MACH do not have the same temporal coverage as the other models.

Line 446-447, the statement, "although again are largely underpredicted" needs more context. Do you mean to say, "models largely underpredict FTIR measurements"?

Line 457-458, drawing this conclusion in relation to the Wespes et al. study seems a bit tenuous because they only compared to one model and it was not one of the models considered in the current study.

**Minor editing suggestions:**

Do Figures 5, 10, and 15 need to have legends to indicate which line is 1:1 and which is the linear fit? I think Figures 7, 12, and 17 also need legends indicating FTIR data and model data points

Line 178, change "FITR," to "FTIR retrieval," or "FTIR retrieved partial column,"

Line 282, change "provided below" to "provided in Fig. 8-10"

Line 299, missing period between "2021)" and "Further". Change "comparison of" to "correlations between"

Line 300, change "1:1 comparison" to "1:1 correlation"

Line 326, change "FTIR comparisons" to "FTIR measurements" or "FTIR retrievals"

Line 330, suggest mentioning that this is eight pairs out of 36.

Line 366, change "emission fluxes" to "anthropogenic emissions" or "anthropogenic fluxes"

Line 388 and Line 391, change "show" to "shows" and "reduce" to "reduces", respectively. Go through the paper and make sure verb tenses when referring to a single figure are singular, the verb should fit the subject outside the parentheses.

---

## Author Comment (AC1)

**Flood et al., Evaluating modelled tropospheric columns of CH$_4$, CO and O$_3$ in the Arctic using ground-based FTIR measurements,**
https://doi.org/10.5194/egusphere-2023-1161

**Response to Reviewers**

**Review 1**

We thank the reviewer for their comments, which have helped us improve the manuscript. Our author responses are given in a blue font, while the italicized text in the indented bullet points has been added in the manuscript.

Overview:

Flood, et. al. has submitted a manuscript comparing ground-based mid-infrared FTIR measurements of tropospheric patrial column O3, CH4 and CO at five arctic sites to 11 model simulations. Daily and monthly averages are compared. Comparisons are conducted for the years 2008, 2009, 2014 and 2015 with the aim to test and comment on model validity, i.e., can the models reproduce the measurements, and if not, why?

The authors build upon prior University of Toronto research into Arctic atmospheric composition using FTIR data. The novelty of this study is that it is the first-time tropospheric patrial column O3, CH4 and CO measurements at the five arctic sites have been compared to this suite of 11 models. As mentioned in the manuscript, in situ and satellite data have been used in past studies to evaluate the performance of these 11 models, but not the FTIR datasets. The FTIR datasets provide an integrated partial column abundance that is quite different in footprint (spatial and temporal) and altitude sensitivity to the datasets used in previous studies hence bringing a new product to assist in model evaluation. This manuscript illustrates the benefits of using such partial column data in model evaluation and should be viewed as another standard dataset (along with in situ and satellite remote sensing) in future model comparison activities.

The manuscript is logically structured and well referenced. The writing is clear and, in most instances, unambiguous. The analysis is robust and easily understood. The content is well within the scope of this journal. Information on data availability is given. The single conflict of interest is minor, stated up front and will be easily dealt with by the journal editors.

I recommend publication of the manuscript after some changes in the manuscript to mainly improve the clarity of content and the context of the investigation.

General comments:

G1/ The Introduction needs more detail to set the context of this research.

In the introduction, the AMAP SLCF assessment report (2021) was used as the basis for setting the context of this research into Arctic SCLFs and the importance of model validation. It is only in subsequent sections that the priori model validation work within Whaley, et al (2022 & 2023), Emmons et al. (2015) and POLARCAT/POLMIP were mentioned. Such past studies should be

mentioned in the introduction to assist the reader in knowing where this current study fits in and what this study is to achieve that the past studies did not. It is only at line 89 where a single sentence states the aim of the study: "This study builds upon the model-measurement comparisons presented in the 2021 AMAP SLCF Assessment Report using an additional Arctic dataset that was not included in the original report.". I view this current research as a natural extension of the work by Whaley, et al., 2022, but using a new dataset (FTIR site measurements) with a different temporal and spatial footprint to that of the in situ and satellite measurements.

The paragraph starting line 75 which introduces the FTIR measurement dataset should also be expanded to give examples of how such measurements from these 5 arctic sites have been used in past Arctic model validation studies. As it currently reads, it is unclear if this is the first time ever such measurements have been compared to models.

I think these changes will be easy to instigate and hopefully improvement context of this current research.

Moved up the sentence "this study builds on…" to earlier in the Introduction and moved the description of the POLARCAT/POLMIP study to the Introduction.

Also added the following text to Section 1:

- *These observations are used to assess what processes need to be revised in the models and how these shortcomings impact the further application of the models, such as for climate and health predictions. Other chapters explore emissions, measurement advances, trends, climate air quality impacts, health ecosystem impacts, and next steps.*

- *Previous studies have used FTIR data to examine model biases in the Arctic (e.g., Wespes et al., 2015; Zhou et al., 2019; Mahieu et al., 2021).*

- *These factors have led to initiatives like the AMAP SLCF Assessment and the POLARCAT (Polar Study using Aircraft, Remote Sensing, Surface Measurements and Models, of Climate, Chemistry, Aerosols and Transport) Model Intercomparison Project (POLMIP) which, in part, aim to assess model performance in the Arctic region. Where POLMIP examined 11 atmospheric models in relation to a variety of Arctic observations taken as part of the International Polar Year in 2008 (Emmons et al., 2015). These studies, in addition to the subsequent complementary publications (i.e., Wespes et al., 2012; Emmons et al., 2015; Monks et al., 2015; Whaley et al., 2022; 2023) provide a valuable point of reference for the modelling of $CH_4$, CO and $O_3$ in the Arctic, which is explored in this paper. This allows for the results presented here to be appraised relative to results from the same models compared to other instruments, with differing temporal frequency and altitude ranges (i.e., Whaley et al., 2022; 2023), with different models / model parameters and Arctic FTIR measurements (i.e., Wespes et al., 2015), and to generally assess the similarities/differences that arise within Arctic SLCF modelling.*

G2/ There is no mention of why column integrated measurements are used to validate/compared to the model simulations. This is one of the main novelties of this study.

At line 518 there is the statement: "NDACC FTIR spectrometers were selected for this project because of the wide range of species measured, high spectral resolution, multiple high-latitude sites, and publicly available data ", which seems the main justification of using the FTIR data (along with a brief contextual reference at line 72: "All of these factors lead to a scarcity of monitoring stations and a limited representation of atmospheric vertical information").

I think these are secondary reasons, the main reason being a (partial) column integrated data product that has a spatial and temporal footprint which is more presentative of the tropospheric free atmosphere than in situ and satellite measurements.

I recommend adding a statement (in the Introduction) focusing on the benefits that validating models using partial column data (that FTIR can provide). The advantages and disadvantages of using column integrated data needs to be explained and how such data allows comparison to models in way in situ and satellite remotely sensed data cannot. It fills a gap.

Added the following text to Section 1:

- *These high-latitude NDACC FTIR instruments provide a valuable set of long-term measurements of multiple species of interest in the Arctic. Compared to surface in situ or satellite observations, the column-integrated FTIR measurements have a spatial and temporal footprint that is more representative of the free troposphere. Performing model-measurement comparisons with partial column data thus complements the assessments presented in the 2021 AMAP Report.*

G3/ It would be good to explicit state why CH4, CO and O3 were the selected species as both the models and measurements have other SLCFs products for which comparisons could be performed.

Added the following text to Section 2.2:

- *The gases $CH_4$, CO, and $O_3$ were chosen for this study as model output for these species was available at 3-hourly intervals, and the FTIR measurements have good sensitivity for them throughout the 0-7 km altitude range, as discussed in the previous section. Note, that not every model has provided all three gases, there are three which have $CH_4$, nine with CO. and 11 with $O_3$ (see Table 3).*

- *While more models participated in the AMAP SLCF Assessment (18 total) and other species were simulated, these were not included in the current study because either the models did not have 3-hourly outputs or the FTIR retrievals had insufficient tropospheric sensitivity (e.g., $NO_2$).*

G4/ There is no mention why the four selected years (2008, 2009, 2014 and 2015) were chosen for the comparison activity. Reason/s why these years were selected need to be stated. Two other points should also be addressed: given this manuscript was submitted in 2023, why was the most recent year 2015? and why long-term trend comparison analysis , i.e., a continuous time series period, was not performed. I suspect model simulation temporal constraints, but this should be stated.

The most recent year was 2015 because emissions development and model simulations take time (on the order of 1-2 years), with analysis subsequent. For example, work began for the 2021 AMAP Assessment Report in 2017, when 2015 was chosen and still "recent"

Added the following text to the Section 2.2:

- *These four years were selected for the 2021 AMAP SLCF Assessment; 2008 and 2009 were previously evaluated in the 2015 Report and 2014 and-2015 were added to include more recent results from years for which Arctic measurements were available at the time (AMAP, 2021).*

G5/ The partial column range used in comparisons is ground level to 7km. A prior study used 0-9km (Wespes, et al. 2012). Please state why the 0-7km range was selected.

Wespes et al., 2012 chose a partial column range from the surface to 300 hPa "both to limit as much as possible the stratospheric and the tropopause height variation influence and to contain the altitude range of maximum sensitivity in the troposphere". Here, we chose a more conservative range from 0 -7 km.

Added the following text to Section 3:

- *Partial columns from 0-7 km were calculated given AMAP's focus on SLCFs in the troposphere, with the cap at 7 km chosen to limit any stratospheric influence.*

G6/ There is no mention of the tropopause heights at the measurement sites. Even if the selected partial column upper boundary (7km) is less than the tropopause height, the averaging kernels might indicate 0-7km partial column measurement sensitivity to above the tropopause. How would it effect model measurement comparisons? Are stratospheric intrusions of major concern?

The tropopause height varies with the time of year and atmospheric conditions, and in the Arctic it ranges from approximately 8 to 12 km (see Fig. 2, for example, from Ny Alesund in Hall et al., Tropopause height at 78° N 16° E: average seasonal variation 2007–2010, Atmos. Chem. Phys., 11, 5485–5490, https://doi.org/10.5194/acp-11-5485-2011, 2011). As stated for G5, 0-7 km was chosen to limit stratospheric influence. Further impacts of the averaging kernels are addressed in response to S3 / S4.

G7/ Reorganization of site-specific figures.

For CH4: seasonal daily and monthly time series plots along with daily model measurement scatter plots are given for a single site, Eureka, i.e., figures 3,4 and 5 and the other four similar FTIR site data plots are in the appendix. Then all site data/metrics for CH4 are displayed in figures 6 and 7. This is repeated for CO and O3.

I found I was continually being referred to figures in the appendix, especially when it came to interpretation of the model measurement results at the end of each species section. I would like the authors to consider rearranging figures. I suggest that all the daily/individual measurement plots, e.g., figures 3,8, 13 be moved to the appendix. The monthly plot , e.g., figure 4 include all stations, in a 2x3 panel plot.

Also, figure 5 to include all stations. For CH4 this would be a 3x5 panel plot.

For figures 10 and 15 I don't think all stations scatter plots can be plotted is a reasonable way in a single figure , thus still relegated to the appendix. If the author can think of another way to concisely display the mentioned data in the main body of the manuscript it could be worth investigating.

Several ways of displaying the data were explored. We aim to present the data in a way that demonstrates the differences within the models and between the different locations. We have moved the full data seasonal plots with the percent difference to the appendix (previously Fig. 3). Additionally, we have moved the Eureka monthly mean with percent difference to the appendix (previously Fig. 4), and replaced it with a multi-panel monthly mean plot, where the model monthly means are shown for each location and the bottom right panel has all of the percent differences on one plot. We acknowledge that elements in this may be difficult to distinguish. The goal of these plots are to provide an overview of what all the models are doing, and where the outliers stand out the most. The monthly mean plots as shown in the original manuscript are still available in the appendix for further examination between sites, if the reader requires a closer look. For the purpose of discussing the summary statistics of all the sites, we have decided to leave the Eureka model vs, FTIR plots in the main text (previously Fig. 5). For conciseness and uniformity between sections, these plots for the other locations remain in the appendix.

G8/ Analysis interpretation of CH4.

Compared to CO and O3, the discussion and interpretation of the CH4 partial column measurement model comparison results are very short. Example: line 266 "satellite 265 instrument and finds that the models are biased low in the vicinity of the tropopause (300hPa) (Whaley et al., 2022)." What height is 300hPa? How much biased low? Is this expected? acceptable?

Please expand and include a greater discussion of the results in comparison to findings from Whaley, et. al., 2022, esp. in context of surface CH4 in situ measurements.

Whaley et al. (2022) show a supplementary plot to indicate the level of AMAP model bias for the ACE-FTS CH4 comparisons and it is described as "low". The bias varies over time and location and is ~100 ppbv near the tropopause.

Added/modified the following text in Section 4.1:

- *The surface in situ CH4 comparison in Whaley et al. (2022) showed that measured surface CH4 VMRs are much more variable than the modelled VMRs. However, in the 0-7 km partial columns in this study, CH4 is well-mixed and more homogenous, resulting in better agreement between the models and the FTIR measurements.*

- *Further, due to the sharp decrease in CH4 above the tropopause (Whaley et al., 2022), a poor representation of the tropopause height may contribute to the low bias in the modelled 0-7 km partial columns, as shown from O3 data in Whaley et al. (2023). The AMAP Report also includes a comparison with upper-troposphere/lower-stratosphere*

*(UTLS) CH$_4$ VMRs as measured by the ACE-FTS (Atmospheric Chemistry Experiment - Fourier Transform Spectrometer) satellite instrument and finds that the models are biased low by ~100 ppb in the vicinity of the tropopause (300 hPa / ~8-9 km), indicating that the modelled tropopause may be too low (Whaley et al., 2022). The results found here are consistent with Whaley et al. (2022), in that that the model simulations of both the lower troposphere (0-7 km partial columns) and the UTLS are biased low, and models with north-south CH$_4$ gradients (here, only GEOS-Chem) have smaller biases than those that do not.*

Specific comments:

S1/ Following on from G1, the first two sentences in the abstract could be improved. They currently do not add any specific information about this study. The abstract should also mention the novelty of this study, i.e., what has been done here that hasn't been done before.

Added/modified the following text in the Abstract:

- *This study evaluates the tropospheric columns of methane, carbon monoxide, and ozone in the Arctic simulated by 11 models. The Arctic warming at nearly four times the global average rate, and with changing emissions in and near the region, it is important to understand Arctic atmospheric composition and how it is changing. Both measurements and modelling of air pollution in the Arctic are difficult, making model validation with local measurements valuable. Evaluations are performed using data from five high-latitude ground-based Fourier transform infrared (FTIR) spectrometers in the Network for the Detection of Atmospheric Composition Change (NDACC). The models were selected as part of the 2021 Arctic Monitoring and Assessment Programme (AMAP) Report on Short-Lived Climate Forcers. This work augments the model-measurement comparisons presented in that report by including a new data source: column-integrated FTIR measurements whose spatial and temporal footprint is more representative of the free troposphere than in situ and satellite measurements.*

S2/ line 51. "…causing most of the pollutants to remain predominantly localised " but throughout the manuscript there is multiple references indicating long range transport [of pollutants] (as at line 370) are a possible cause of measurement model differences. Can this disparity be rectified.

Removed this statement.

S3/ line 125. Degrees of freedom and the partial column averaging kernel (PC AVK): Could figure 1 altitude range be expanded to ~ 20km to see 'what happens above 8km'. If the PC AVK above 7km is ~ 1.0 this means retrieval information above 7 km is incorporated into the 0-7km PC. If so, please comment upon, and implications of.

The figure has been changed to include the partial column and total column averaging kernels, with the altitude range expanded to 20 km (as per comment T4).

Note that as per Table 2, the DOFs for the 0-7 km partial column are near 1 for all species, meaning that the retrieval of this tropospheric partial column are sufficiently sensitive to the troposphere and can be reliably discerned the from the upper retrieval layers other levels.

S4/ line 124 and Table 2. Please expand commentary and implications for DOFs < 1.0. For CO and O3 the PC DOFs are ~1.0, but for CH4 the DOFs are < 1.0, and from figure 1, there is less sensitivity to near surface CH4. What are the implications of this for model comparisons?

Added the following text to Section 2.1:

- *The degrees of freedom for signal (DOFS) is calculated by taking the trace of the averaging kernel; this indicates the number of independent pieces of information coming from each retrieval, or inversely, the number of components not constrained by the a priori.*

- *The number of measurements, mean DOFS, and mean percent error of the 0-7 km partial columns of $CH_4$, CO, and $O_3$ for 2008, 2009, 2014, and 2015, for each station, are listed in Table 2. The mean partial column (0-7 km and 7-20 km) and total column averaging kernels for $CH_4$, CO, and $O_3$ for 2008, 2009, 2014, and 2015, are shown in Fig. 1. The DOFS and averaging kernels are indicators of the vertical information within a retrieval. Fig. 1 shows the mean partial column averaging kernels for 0-7 km and 7-20 km are distinguishable, with maxima at different altitudes. The mean total column averaging kernels for all three species appear smooth around 1.0, which indicates that contributions from all altitudes have similar weights in the total columns. By altitude, the sensitivity of each species is >0.5 in the partial column examined (not shown), meaning that more than half of the retrieved profile information comes from the measurement (Vigouroux et al., 2009). The average DOFS vary by species and somewhat by station, given the reduced column height of 0-7 km, some of the values are less than one, meaning the retrieval is somewhat constrained by the a priori. However, it should be noted the comparisons presented in this paper account for the vertical sensitivity of the FTIR measurements by smoothing the model data with the averaging kernels. This process is described in Sect. 3.*

S5/ Section 2.2. Relative to other manuscripts the section describing model simulations is brief, but I think it is justified as detailed model descriptions (and forcings) are given in Whaley, et. al., (2022). I see no need to repeat information that is already readily available.

Could the authors make sure that any model output that is used in this current study that differs from model output used in the study by Whaley, et al., (2022) be stated and the reasons for the change (e.g., an updated model or forcings) also be stated. This may seem a logical statement, but if the authors are going to heavily defer to Whaley, et al., (2022) to provide details then it is very important there are no changes or changes are identified.

Added the following text to Section 2.2:

- *The model simulations are the same as those discussed in Whaley et al. (2022; 2023), and the 2021 AMAP SLCF Report, however, the analyses there were performed with the*

*monthly-mean output, while the analysis here is with the 3-hourly output, all of which is available at http://crd-data-donnees-rdc.ec.gc.ca/CCCMA/products/AMAP/.*

S6/ The first two sentences in section 3 are not needed, as it is covered in the section 'Data Availability', or if the authors want to retain it in the manuscript, then relocate to section 2.

Moved to Sect. 2, as suggested.

S7/ Figure 2. The flow chart alludes to that the 'nearest' model grid point (to a measurement site) is used. This should be mentioned in the text. To clarify, is there any spatial weighting of localised grid points? I.e., weighting/kriging of the closet model points/cells to the FITR location? Have tests been done concerning a geolocation weighted average model value? I gather any differences will be minimal but would be good to confirm, even if for a single site.

Added the following text to Section 3 to reflect Fig. 2:

- *The date/time and volume-mixing-ratio profiles from the model output are extracted from the grid point that is closest to the FTIR location.*

There was no spatial weighting or interpolation of the local model grid cells when the location matching was done, following the approach of Whaley et al. (2022; 2023). Given that the FTIR measures along a slant path towards the sun, that measurement covers more than just the point location at the ground and is thus more representative of the larger model grid cell. We also don't expect large gradients in concentrations at these remote locations in the Arctic.

S8/ I think another paragraph is needed at the end section 3 concerning the type of analysis that is going to be performed using eqn. 1 and 2 as the quantification metrics. Are you going to investigate, diurnal, daily, monthly, or seasonal differences? Long term trends? Basically, what are you going to look at.

Added the following text to Section 3:

- *These steps are taken to establish the modelled seasonal cycles, and quantify the differences between the models and measurements, by month and season. Further, assessing the MMM by month allows for a general overview of when and where models diverge from measurements and can help suggest shortcomings in the models. There are not enough measurements per day to evaluate a diurnal cycle, although it is expected to be small in the Arctic, and there are not enough years available in the 3-hourly dataset used here to examine long-term trends.*

S8/ Best line fits: linear regression. Do the best line fits in all the analysis also take into account the uncertainty in the abscissa (measurements) as well as the ordinate (model)? If so, please state so, if not, then maybe prudent to perform a few tests to assess the effect on the linear fit. Since measurement and model uncertainties are of comparable magnitude, abscissa uncertainty could have a large effect on the calculated linear fit.

The line is simply a line of best fit, as the model values do not have an associated error value.

Added the following text to the manuscript to clarify:

- *A regression line is fit to the raw scatter-plot data of the model output versus FTIR measurements using all the available data points, where each plot lists the equation of this line and the correlation coefficient, $R^2$.*

As a test (see example plot below for MATCH model and Eureka $O_3$), a subset of residuals (the difference between the regression line and model values) were plotted against the corresponding FTIR partial column values, this showed that the residuals are randomly distributed across the range of partial column values. Further, examining the FTIR uncertainty (as it is described in the paper), as a percentage of the FTIR partial column value shows that the percent uncertainty in the 0-7 km partial column does not increase linearly with the value of the partial column, and as such the residuals in the non-weighted regression do not exhibit heteroscedasticity.

[Figure]

S9/ line 244:

"For all models, the R2 values for Ny Ålesund and Harestua are significantly smaller, while the overall mean percent difference is comparable to the other locations. The discrepancy is likely attributed to the smaller number of measurement points, causing outliers to have more weight in the linear regression, which is reflected in the elevated NRMSE for Ny Ålesund across all models."

I do not think a lack of lack of measurement points is a cause. Both Figs A9 and A12 show there are plenty of data points. Fig A12 clearly shows there are outlier measurements at Harestua. I would attribute this to either measurement/retrieval error that was not filtered out thus should be removed from the comparison datasets, or anomalous atmospheric events which if at fine temporal or spatial scale the models would be able to reproduce, thus this measurement period should also be omitted as the model would not be able to replicate it. Given that the anomalous measurements are both too high and too low I suspect measurement error. I recommend omitting such outliers (across all data sets, unless it can be accounted for) using a defined filtering method and perform analysis again.

This approach will not account for the low R^2 at Ny Ålesund and I cannot easily see why the R^2 is lower than at other sites.

Upon re-evaluation and conferring with the instrument PI, the date range of November 2014-March 2015 (42 points total) from Harestua have been removed due to instrument error (this period includes the outlier points).

When the points are removed. the $R^2$ for Harestua for all three models increased from 0.15 to 0.57-0.58, the mean bias went from -9.9 ± 3.1 to -10.6 ± 1.5 for CMAM, -6.2 ± 3.2 to -6.8 ± 1.2 for GEOS-Chem and -9.4 ± 3.1 to -10.1 ± 1.4 for MRI-ESM2. This is because some points in early 2015 were removed that were previously reducing the mean bias. The NRMSE changes from 1.73, 1.35, and 1.67 to 4.29, 2.87, and 4.07, respectively.  These numbers have been updated in the manuscript.

The comment regarding number of measurement points at Ny Ålesund and Harestua has been removed, and the following text has been added to Section 4.1 in its place:

- *The mean differences for each model across sites are relatively consistent, while the results vary more when comparing $R^2$ and NRMSE. Particularly, when comparing between the same model, the $R^2$ for Ny Ålesund is the lowest and the NRMSE is the highest. The data from Ny Ålesund show less of a seasonal cycle than the other locations, and the FTIR uncertainty for $CH_4$ at Ny Ålesund is more than twice that of the other sites (see Table 4). The larger uncertainty may lead to reduced sensitivity to small changes, and increased variability masking seasonal changes, which can contribute to the discrepancy between the models and observations.*

S10/ line338: "Similar trends have been found in other Arctic model-measurement comparison studies." Please reference this statement, also do you mean trends or findings? As temporal trends are not investigated in this study. I think would also be helpful to quantitatively state the amount of underprediction in prior studies and then relative to this study (referring to table D1 would be a good idea when comparing the results from this study to prior studies ).

This sentence was meant to act as a bridge from the statement of the pattern observed in the figure to the sentences that follow which discuss the similar findings from other studies.  It has been rewritten in Section 4.2 to clarify:

- *This highlights the general tendency of the models to underpredict tropospheric CO more in the spring than in the summer, which has been observed by other Arctic model-measurement comparison studies.*

The results in Whaley et al. (2022) for CO from MOPITT are given as seasonal averages for each model at the 600 hPa pressure level, shown over the global domain. Unfortunately, they did not provide a MMM averaged over the Arctic region for the CO MOPITT comparisons, and even if they did, it would be for the 600 hPa pressure level only and in ppbv units, which can't be directly compared to our 0-7 km partial columns. Therefore, we can't be quantitative when we compare our values to theirs.

Added mmm % difference given in Whaley et al. (2023) in Section 4.2:

- *The change from a negative winter-spring bias to a positive summer bias was observed in model comparisons to surface CO measurements at two additional Arctic sites, Zeppelin, Norway and Utqiagvik/Barrow, USA , with a -20-30% bias in the first six months of the year (Whaley et al., 2023), which is compatible with results shown in Fig.10(e).*

S11/ line 355: "Further, the tracer investigation shows that OH differences account for more variability between the models than the transport mechanisms within the individual models."

Could this statement please be referenced.

The sentence follows a sentence which is cited, the "further" is to indicate it is an extension of the pervious sentence, additionally the following sentence is cited for the same paper.  The sentence in Section 4.2 has been revised to:

- *Furthermore, the tracer investigation in that study shows showed that OH differences account for more variability between the models than the transport mechanisms within the individual models.*

S12/ line 366: "The results of the model-FTIR comparisons presented here support this reasoning, as the model with a positive bias (GEM-MACH) has a different emissions input, with possibly more complete emissions in the Arctic, as this was a high-resolution Arctic version."

This conjecture could be quite easily solved by looking at the model simulation parameters to see if this is true.

The sentence in Section 4.2 has been modified to resolve:

- *The results of the model-FTIR comparisons presented here support this reasoning, as the only model with a positive bias (GEM-MACH) has additional local Arctic emissions (Gong et al, 2018).*

S13/ line 381: "In addition to atmospheric chemistry, its production is highly sensitive to meteorological conditions. Therefore, it is difficult for models to accurately simulate tropospheric O3." Ozone also can have a significant diurnal cycle due to photochemistry, complicating comparisons when measurements and model differ in time. Please include this cause as well.

Text has been added to Section 4.3, however it is pertinent to note that because the model and measurements only differ in time by a maximum of 1.5 hours, during which time large changes in Arctic $O_3$ would not be expected, particularly with the elongated daylight / darkness at these high latitudes.

- *In the troposphere, $O_3$ is a secondary pollutant, produced by photochemical oxidation of volatile organic compounds in the presence of $NO_x$. In addition to atmospheric photochemistry, its production is highly sensitive to meteorological conditions. Diurnal impacts on $O_3$ production are minimal in the Arctic, relative to lower latitudes, due to the gradual and prolonged change in solar altitude/angle throughout the year.*

S14/ line: 452. " To supplement the aircraft and satellite campaigns undertaken for the POLARCAT study, daily mean O3 measurements from the FTIR instruments at Eureka and Thule were compared to MOZART-4 simulations in Wespes, et al. (2012)".

Due to the daily diurnal cycle of ozone, comparisons of daily FTIR averaged ozone measurements would be biased high to model output (that uses daytime and nighttime values as I gather nighttime FTIR measurements are not taken). Can you confirm daily average MOZART ozone was used or matched to FTIR measurement times.

The Wespes paper states that they compare FTIR and MOZART daily mean ozone columns; nothing is stated about matching the model output to FTIR measurement times. So it is possible that there is a resulting bias as the reviewer notes. However, since the comparisons are from April-July, there should be minimal diurnal variation due to 24-hour sunlight at the latitudes of Eureka and Thule.

S15/ line 518:

"NDACC FTIR spectrometers were selected for this project because of the wide range of species measured, high spectral resolution, multiple high-latitude sites, and publicly available data."

As stated in G3, a better reason for using FTIR datasets should be given. This relates back to a general comment of the overall benefits of using column integrated measurements.

Added the following text to Section 5:

- *in addition, the column-integrated FTIR measurements used in this study have a spatial and temporal footprint that is more representative of the free troposphere than in situ and satellite measurements.*

S16/ Defining FTIR uncertainty. This term (or variations of) is found within the text (e.g., lines 248, 323, 514) but not clearly defined. Is it the uncertainty of individual measurements as in table 2, or the 1- sigma standard deviation of the daily/monthly measurement means?

To reduce ambiguity, the FTIR error/uncertainty has been renamed as "Mean Percent Uncertainty" in Table 2 and referred to as uncertainty rather than error elsewhere in the manuscript.

Also added/modified the following text and equations to Section 3 to clearly define all the terms used, and added symbols to figure captions throughout:

- *To compare the model and FTIR partial columns, a model-measurement percent difference ($\Delta_i$) is calculated, as defined by Eq. 2 for a single model-measurement pair (i), where $PC_{M,i}$ and $PC_{F,i}$ are the 0-7 km partial columns for the model and FTIR, respectively:*

$$\Delta_i = \left( \frac{PC_{M,i} - PC_{F,i}}{PC_{F,i}} \right) \times 100 . \hspace{2cm} (2)$$

  *A regression line is fit to the raw scatter-plot data of the model output versus FTIR measurements using all the available data points, where each plot includes the equation*

*of this line and the correlation coefficient, $R^2$. The normalized root mean square error (NRMSE), given by Eq. 3, is presented for each model and location, where N is the total number of model-measurement pairs (Kärnä and Baptista, 2016). The root mean square error is normalized to the standard deviation of the FTIR data ($\sigma_F$) used in the respective analysis:*

$$NRMSE = \frac{1}{\sigma_F} \sqrt{\left[\sum_{i=1}^{N}(PC_{M,i} - PC_{F,i})^2\right]} . \qquad (3)$$

*In addition to evaluating the models using every available FTIR data point in the analysis years, the monthly mean annual cycles are also presented. The monthly mean partial columns ($PC_{F,monthly,j}$) are calculated by taking the mean of every measurement in a given month (j), where $N_j$ is the number of points included in the month for all years considered. The monthly model mean partial columns ($PC_{M,monthly,j}$) are made in the same manner, using only the smoothed partial columns that have a corresponding matching FTIR measurement, as defined above. Equation 4 outlines the calculation of a monthly mean partial column for month j for a: the FTIRs ($PC_{F,monthly,j}$), and b: the models ($PC_{M,monthly,j}$):*

$$PC_{M,monthly,j} = \frac{1}{N_j}\sum_{i=1}^{Nj} PC_{M,i}. \qquad (4a)$$

$$PC_{F,monthly,j} = \frac{1}{N_j}\sum_{i=1}^{Nj} PC_{F,i}. \qquad (4b)$$

*The model-measurement monthly mean percent difference ($\Delta_{monthly,j}$), shown by Eq. 5, follows the same process as the monthly-mean partial column, and is the mean value from Eq. 2 for each month (j) across the years, where the error bars on the monthly mean plots represent the standard deviation of this mean:*

$$\Delta_{monthly,j} = \frac{1}{N_j}\sum_{i=1}^{Nj} \Delta_i. \qquad (5)$$

*The mean of these monthly mean differences is used to calculate the overall mean percent difference ($\Delta_O$) for each model, sometimes referred to as model bias, where $N_{months}$ is the number of measurement months in a calendar year at that location (see Table 1), and the uncertainty given is the standard deviation of this mean:*

$$\Delta_O = \frac{1}{N_{months}}\sum_{j=1}^{N_{months}} \Delta_{monthly,j}. \qquad (6)$$

*Finally, the monthly multi-model mean (MMM) partial column for month j ($PC_{MMM,monthly,j}$) is calculated by taking the mean $PC_{M,monthly,j}$ for all models, at a given location, calculated with Eq. 4b, and the MMM monthly mean difference ($\Delta_{MMM,monthly,j}$) is the mean of $\Delta_{monthly,j}$ for all models, at a given location calculated with Eq. 5. The overall percent difference of the MMM-measurement ($\Delta_{O,MMM}$) is given by Eq. 7:*

$$\Delta_{O,MMM} = \frac{1}{N_{months}} \sum_{j=1}^{N_{months}} \Delta_{MMM,monthly,j}. \tag{7}$$

S17/ The table 4 caption states:

"Summary of mean percent difference for each model and location by species. MMM is the multi- model mean. The colour scale indicates the mean percent difference relative to the FTIR measurements, from blue (-50%) to red (+50%). A square marker indicates that the mean difference is within the FTIR uncertainty. A triangle marker indicates that the mean difference is within the FTIR uncertainty combined with the standard deviation of the monthly mean percent difference."

It is difficult to understand what is being compared (and significance of the metric ) when FTIR uncertainty is not clearly defined. Is FTIR uncertainty the monthly measurement 1-sigma S.D. or the uncertainties of a single measurement as given in table 2?

There is no explanation of why a double metric is used, could this be explained in the text. What does it mean if "the mean difference is within the FTIR uncertainty" but not "within the FTIR uncertainty combined with the standard deviation of the monthly mean percent difference".

Added the following text to Section 3:

- *These steps are taken to establish the modelled seasonal cycles, and quantify the differences between the models and measurements, by month and season. Further, assessing the MMM by month allows for a general overview of when and where models diverge from measurements and can help suggest shortcomings in the models. There are not enough measurements per day to evaluate a diurnal cycle, although it is expected to be small in the Arctic, and there are not enough years available in the 3-hourly dataset used here to examine long-term trends.*
  *When discussing FTIR uncertainty, this refers to the mean uncertainty per gas and station, as listed in Table 2. When discussing the mean difference between the model and measurements, this refers to the overall mean difference ($\Delta_O$) as described by Eq. 6. In Sects. 4 and 5, these two parameters are used to assess model performance: if $\Delta_O$ is within measurement (FTIR) uncertainty, the model can be considered in general agreement with the FTIR; if $\Delta_O \pm$ the standard deviation of the mean is within the measurement uncertainty, then the model is sometimes in agreement with the measurements; and if the uncertainty and $\Delta_O$ do not overlap then the model and measurements do not agree.*

S18/ Table D1: Is an important table. I recommend putting this in the main body of the manuscript and referred to in each species section.

This table has been added to the main text as Table 5.

Technical comments:

T1/ line 81. Arctic is not defined, are you implying >60N? Maybe define what 'Arctic' is. T2/ Table 1 and Table 2 colour key columns are not needed.

Added the following text to Section 1:

- *Five of the 28 NDACC FTIR stations are located at latitudes north of 60°N, for the purpose of this study, these will all be referred to as Arctic sites. The five sites are: Eureka, Canada; Ny Ålesund, Norway; Thule, Greenland; Kiruna, Sweden; and Harestua, Norway.*

T3/ Paragraph starting line 114 concerning technical details about the FTIR data and retrieval strategies. I think there is a need to mention the vertical grid spacing of the retrieval, i.e., how many layers, esp. in the troposphere, and from 0-7km.

All sites have 11 layers, except for Ny Ålesund which is at a lower altitude (as listed in Table 1) and has 12 layers in the partial column. The layers thickness change with altitude, but are more less consistent (within rounding) between stations.

Added the following text to the manuscript:

- *The partial column examined here (0-7 km) encompasses 11 vertical layers for all sites, except Ny Ålesund, which has an additional (12$^{th}$) layer given the lower altitude of its location (see Table 1).*

T4/ Figure 1. The term 'mean column'. Do you mean total or partial (0-7km) column? Please make this clear in the label. If it is total column, then I recommend replotting as 0-7km partial column.

As mentioned in the reply to S3, Figure 1 has been revised to include partial and total column averaging kernels.

T5/ Figure 1. The abscissa axis (Partial? column AVK) needs units. [unitless] or [relative] would suffice if not [ppb/ppb].

The units have been added to the axis and caption of this figure.

T6/ line 166. (+/- 1.5 hours): I think it needs to be explicitly stated why this time frame was chosen (from previous model comparison studies?), just to make it clear why , say , +/-24h cannot be used. A tight time constraint is required for ozone due to diurnal photochemistry.

Added/ modified text to clarify:

- *The FTIR measurements are matched with the 3-hourly model measurement closest in time (±<1.5 hours), this is done to minimize the time difference between the two points, such that no measurement is greater than 1.5 hours from a modelled output.*

T7/ line 173. Partial column averaging kernel I gather? Maybe add 'partial column'.

Modified the sentence in Section 3 to clarify:

- *The calculation for the smoothing is shown in Eq. 1, where $x_a$ is the FTIR a priori VMR vertical profile, $A$ is the VMR averaging kernel matrix from the corresponding FTIR measurement, and $x_{model}$ is the modelled VMR vertical profile.*

T8/ line 176. "ratio between the trace gas VMR and layer airmass (molec cm^-2)". Best to add the term 'layer airmass' for clarity.

Modified the sentence in Section 3 to clarify:

- *The model VMR profile is then transformed to a layer profile in units of molecules per centimeter squared using the ratio between the VMR and layer partial column (in molecules per centimeter squared) in the retrieved FTIR profile as the conversion factor.*

T9/ line 180. To clarify, is the station altitude is also used as the lower model partial column layer boundary in analysis? If so, then I think this needs to be stated.

Yes, the models' lowest layer follows the surface topography, and their native levels are interpolated onto the FTIR altitude grid. This is indicated by the statement "Note that "0 km" is used as proxy for the minimum altitude, but this varies, based on location, with the altitude of each instrument listed in Table 1.".

T10/ line 206. Please replaced 'important' with a more specific descriptor. Important is too subjective (important in what context?).

Changed to "powerful greenhouse gas".

T11/ line 213. 'concentration' should be replaced with 'partial column'. The models are forced with concentrations (vmr), but the quantities under investigation are partial columns (molec. cm^-2).

Changed as suggested.

T12/ Line 217. 'Little variance'. Sorry, I found this unclear, do you mean between the models or intra- model (within a month or day)?

Statement has been removed to prevent confusion, it was repetitive in nature.

T13/ line 219. 'uniformity', of what?

Added the following text to Section 4.1 to clarify:

- *The uniformity between the years (see A1-A5 for full data timeseries plots) and consistency of the model biases between sites is likely a consequence of being prescribed in the models, in addition to the longer lifetime of CH4, relative to the other SLCFs.*

T14/ Figure 3, 8 and 13: The measurement symbols are extremely hard to differentiate between. Can you make them easier to differentiate?

These plots have been moved to the appendix (Figures A1, B1, and C1) as suggested in another comment. The difference between the years is not discussed, and so the symbols have been removed.

T15/ All figures: In all the figures, when model data is plotted, I gather it is modelled smoothed partial columns? If so, please add 'smoothed' to all 'model data' just to make it clear.

It is stated in Section 3 that all of the model data presented is smoothed, but this has been added to the relevant figure captions to reiterate.

T16/ line 380. "It is a secondary pollutant" replace with "In the troposphere, ozone is a secondary…"

Added suggested text.

T17/ line 395. "However, the FTIR O3 seasonal cycle does not have a springtime minimum from surface ODEs, as one might expect from surface measurements". Sorry, this does make sense. As it reads FTIR partial measurements are surface measurements? Can you please rewrite to make it clearer.

Modified text for clarity:

- *However, the 0-7 km partial column FTIR $O_3$ seasonal cycle, shown here, is dominated by the free troposphere and stratospheric processes, and does not have a springtime minimum from surface ODEs, as one might expect from surface measurements (Solberg et al., 1996; Berg et al., 2003; Skov et al., 2006; Eneroth et al., 2007; Whaley et al, 2023).*

T18/ line 422. remove the word 'difficult'.

Removed text, as suggested.

T19/ line 425. remove 'and as such recommended for future work'. I can understand what is trying to be conveyed, but nearly instance of a model measurement disagreement warrants future work.

Removed text, as suggested.

T20/ line 497. remove the word 'historical'.

Removed text, as suggested.

---

## Author Comment (AC2)

**Flood et al., Evaluating modelled tropospheric columns of CH₄, CO and O₃ in the Arctic using ground-based FTIR measurements,**
[https://doi.org/10.5194/egusphere-2023-1161](https://doi.org/10.5194/egusphere-2023-1161)

**Response to Reviewers**

**Review 2**

We thank the reviewer for their comments, which have helped us improve the manuscript. Our author responses are given in a blue font, while the italicized text in the indented bullet points has been added in the manuscript.

General comments:

The manuscript by Flood et al. evaluates the accuracy of model estimates of partial atmospheric columns of CH4, CO, and O3 in Arctic regions by comparing against FTIR ground-based measurements from five northern high latitude sites in the Network for Detection of Atmospheric Composition Change (NDACC). The study considers 11 models and expands on previous analysis published in the Arctic Monitoring and Assessment Programme (AMAP) 2021 assessment report. That report included comparisons of 18 models to surface in situ, aircraft, and satellite-based measurements, but did not compare to ground-based FTIR measurements of partial column concentrations. Time-series of FTIR measurements and model estimates, correlations between FTIR measurements and model estimates, comparisons of seasonal cycles, and summarizing statistics of comparisons of model estimates and FTIR measurements are presented for each NDACC site. The site Eureka is the primary focus of most of the discussion in the paper, while comparisons at the remaining NDACC sites are mostly presented in the Appendices. There is also a balanced discussion of how the results of the comparison may or may not support previous analyses of this type.

Overall, the value of this analysis is clear. Comparing model estimates of greenhouse gas concentrations to FTIR retrievals is useful and provides a wealth of additional information that cannot be provided by comparing to surface-based measurements, and with much greater temporal frequency than aircraft-based measurements. Furthermore, this analysis is thorough and the interpretation of the results is reinforced by comparisons to previous literature. That being said, there are critical ways that the paper could and should be improved before publication. First, while references for each model considered are provided in the paper, the differences in model frameworks, assumptions, boundary conditions, and set up are not discussed much in the paper. The authors seem to expect the reader to conduct a high degree of background research or already have a very thorough understanding of all of these models. This is particularly relevant in the failure to state the spatial resolutions of the models being used and the failure to sight which models do or do not simulate the stratosphere (particularly in the context of the O3 discussion). There is certainly some discussion of factors that may be driving specific model biases, but these could be discussed more in the context of specific models. In addition, I think that some generalizations or conclusions are not as consistent with the plotted results as the authors

suggest; however, many of the plots are difficult to read and I think reformatting some of the figures would really help.

Specific comments and suggestions:

In general, the paper would greatly benefit from a reorganization of how data is presented in the figures. Specifically, Figures 3, 4, 8, 9, 13, and 14 would be better presented as separate panelled subplots for each model, similar in format to Figures 5, 10, and 15. As they are, these figures are very difficult to read with many overlapping lines that make it hard to differentiate the behaviour of individual models.

After considering the feedback from both reviewers, the figures in the text have been modified and moved to help with flow, but retain clarity. Figures 3, 8, and 13 have been moved to the appendix, and since we don't discuss the difference between each of the four years, we have removed the different symbols for them. We feel that Figures 4, 9, and 14 show a better view of the models in relation to each other when they are presented on the same axis and prefer to keep them in one. These have been moved to the appendix and a subpanel plot with all of the monthly mean results at all locations has been added in the text.

Line 84, why are you only considering 11 of the 18 models used in the AMAP report?

Added the following text to Section 2.2:

- *While more models participated in the AMAP SLCF Assessment (18 total) and other species were simulated, these were not included in the current study because either the models did not have 3-hourly outputs or the FTIR retrievals had insufficient tropospheric sensitivity (e.g., $NO_2$).*

In section 2, line 109, you say that SFIT4 is used to retrieve VMR profiles from NDACC FTIR measurements at all sites except Kiruna, which uses PROFFIT. There needs to be some discussion of how these two retrieval algorithms differ and how this may impact the resulting retrievals.

Alternatively, you can cite a paper that compares the two. It seems unlikely that using different retrieval algorithms would not have some effect. Furthermore, in section 2, you should state what the typical temporal frequency of NDACC FTIR observations are. This becomes relevant in section 3 because it is useful to know approximately how many FTIR observations are included in the 3-hour averages that are compared to model estimates. Similarly, some information about the spatial resolutions of the models being considered and the distances between the NDACC sites and the referenced location of the model estimates is important. Do all the model data have the same spatial resolutions and do distances between site locations and model estimates vary among sites or among models? Could any of this variability explain differences seen in the model comparisons?

Added the following text to Section 2.1:

- *All sites included in this paper use SFIT4, except Kiruna, which uses a comparable retrieval code called PROFFIT, which has been shown to agree well with SFIT (Hase et. al, 2004).*

Table 1 lists the operational season of the FTIR instruments at each site and Table 2 indicates how many measurements were taken in the years being investigated. The temporal frequency of measurements would depend on operations of each location, including instrument downtime and weather.  There are very few instances (<10 across sites) of multiple FTIR measurements being averaged from falling closest to the same model interval.

The NDACC locations are listed within Table 1.

The model spatial resolution has been added to Table 3.

In Section 3, specifically the flow chart in Fig. 2, since the FTIR measurements are collected with greater temporal frequency than the model estimates, it would seem that the temporal matching would involve finding the FTIR measurements closest in time to model estimates, rather than the other way around. Also, please clarify whether averaging kernels and a priori values are retrieved for each FTIR observation or based on a general reference for the instrument. If they do vary with each observation, how is this handled when applying the corrections.

Although multiple FTIR measurements can occur within a 3-hour period, there were very few instances where this occurred. When it did (as stated in the text) the partial columns were averaged. Given that there are far more modelled results, the FTIR measurements were matched with the model output that was closest in time to the measurement.

This statement in Section 3 indicates that each FTIR measurement has its own averaging kernel and that the smoothing applied to the model profile uses the averaging kernel of the relevant FTIR measurement:

> *"Then, the model VMR profile is smoothed using the respective FTIR measurement's averaging kernel and a priori profile."*

This statement in Section 2.1 indicates that each station has a single a priori profile, but that the pressure and temperature profiles correspond to the conditions near the time of each measurement:

> *"The a priori information for the modelled spectra is provided by 40-year-average profiles from the Whole Atmosphere Community Climate Model (WACCM) (Marsh et al., 2013), with spectroscopic absorption parameters from the HITRAN 2008 line-list (Rothman et al., 2009) and daily pressure and temperature profiles from the U.S. National Centers for Environmental Prediction (NCEP) (Kalnay et al., 1996)."*

It seems like the second paragraph of section 4.1 belongs in methods. By extension, it would make sense to talk about how many models you are including in the comparison analysis for each gas earlier in the paper (in section 2 or 3), though the gases covered by each model are

shown in Table 3 the number of models that estimate each gas should be summarized in the text as well.

Added the following text to the manuscript in Section 2.2:

- *Note that not every model has provided all three gases; there are three which have $CH_4$, nine with CO, and 11 with $O_3$ (see Table 3).*

Moved text in Section 4.1 regarding $CH_4$ prescribed in models to Section 2.2, as suggested.

Can you speculate on why GEOS-Chem has lower % differences (better accuracy), but also lower correlation coefficients (poor precision or more scatter) in the CH4 comparisons?

Added the following text to Section 4.1:

- *GEOS-Chem does simulate a north-south gradient, which is reflected in the smaller overall model-measurement percent difference, compared to other models, in all locations (note Fig. 6 in Whaley et al., 2022). However, the $R^2$ of GEOS-Chem vs. FTIR is smaller than that for the other models at some locations (Eureka and Kiruna), which can be attributed to the increase in variability the gradient introduces – including some instances of overestimation.*

Line 262, you say that the FTIRs show good sensitivity to surface CH4, but Fig.1 shows that the instruments are more sensitive to higher altitudes in the partial column than they are to the surface. I also wonder if variations in the tropopause height or a poor representation of this in the models or in the FTIR retrievals could affect the accuracy of your CH4 partial columns in the comparisons.

Added /modified the text in Section 4.1:

- *The FTIR retrievals show good sensitivity to tropospheric $CH_4$ (sensitivity >0.5), however, as these column measurements average out $CH_4$ biases over the tropospheric column, they are not expected to exactly match the surface measurement comparisons. Furthermore, due to the sharp decrease in $CH_4$ above the tropopause (Whaley et al., 2022), a poor representation of the tropopause height may contribute to the low bias in the 0-7 km partial columns, as shown from $O_3$ data in Whaley et al. (2023).*

We have also included the partial column averaging kernels for 0-7 km and 7-20 km to show the difference between the altitude ranges in the partial columns.

In section 4.2, could the increased discrepancies between FTIR retrievals and model estimates of CO in spring also be at least partly explained by errors or biases in the FTIR observations due to low solar zenith angle or cloud cover?

We do not believe these factors account for the discrepancies as the other Arctic modelling papers discussed found similar results for CO comparisons using in situ and satellite measurements (e.g., Whaley et al., 2022).

Line 294, I think the seasonal shifts in bias for EMEP-MSC-W and WRF-Chem are more remarkable than for MATCH (at least at Eureka, which seems to be the site that the results discussion is primarily focused on), but these are not mentioned.

Added the following text to Section 4.2:

- *WRF-Chem is biased low in the spring and summer, but agrees better with the observations from August onwards, in contrast to EMEP-MSC-W, which tends to diverge from the measurements in the mid- to late summer.*

On page 17, when discussing Fig. 11, why not comment on the fact that WRF-Chem has correlation coefficients near zero and very high NRMSE relative to the other models?

The following text has been appended to the statements which were already included in the text on this topic:

- *WRF-Chem shows better agreement with the FTIR measurements from Eureka, where the NRMSE is comparable to CESM, CMAM and GEOS-Chem. This is likely a result of the increased density of measurement points in August and September, when WRF-Chem exhibits a minimum bias compared to the FTIR data, and because the comparison only includes data points from 2014 and 2015. The large negative biases earlier in the year lead to low $R^2$ and high NRMSE at all sites. This appears to be linked to negative biases in modelled surface CO over mid-latitude source regions, and in the free troposphere compared to MOPITT data, as reported by Whaley et al. (2022).*

Line 353-355, is this describing a global effect in which European emissions have a greater influence on surface CO everywhere, or were the studies conducted in Europe and the surface CO is more affected by local emissions?

The results discussed in relation to the POLMIP study are regarding the Arctic. This is noted in the preceding sentence: *"Using an idealized tracer, POLMIP examined anthropogenic and biomass burning influences in Arctic regions, demonstrating a seasonal dependence of transport efficiency"*.

In section 4.3, please clarify which months are included in "springtime". It seems that most models agree better relative to other models as well as FTIR in February and March than all other months except September. If these months are part of springtime that does not support your claim that springtime O3 concentrations are poorly characterized in the models. Furthermore, if April is part of springtime, WRF-Chem should be mentioned on Lines 400-401, along with UKESM1, GEM-MACH, and GEOS-Chem.

We have added (late February - May) to the text on first mention of "spring" to define it. We agree that the MMM has little to no bias in the springtime O3 at Eureka, however, there is a large spread in springtime O3 values across models. While we discussed each of the models' behavior in the springtime, we did not state that overall it is poorly characterized, just that it is quite variable.

Added WRF-Chem to the statement, as suggested.

Line 401-402, please elaborate on the reasoning behind this conclusion ("may be attributed to a low bias in the models' lateral boundary condition, inaccuracies in model water vapour and/or a lack of O3 transported from mid-latitudes.").

Added/modified text in Section 4.3:

- *The discrepancies may arise from inaccuracies in model water vapor leading to an increase in $O_3$ destruction and/or a lack of $O_3$ transported from mid-latitudes, which is a substantial source of tropospheric $O_3$ in the Arctic (Hirdman et al., 2010; Whaley et al., 2023). In the case of the regional GEM-MACH model, low biases in $O_3$ or precursor species at the lateral boundary conditions may also be contributing.*

Line 403-404, why is MRI-ESM2 not mentioned here? That model seems to track very well with the FTIR measurements in Figures 13 and 14.

Added as suggested.

Line 409-410, this claim does not seem to be as consistently relevant for O3 as it is with CO. There are a number of models, including EMEP-MSC-W, DEHM, and CESM, that appear to exhibit a high degree of scatter for lower O3 partial column concentrations (at least for Eureka).

Reworded in Section 4.3 to better describe the results shown in the figures:

- *The general underprediction towards the largest values could be related to the underestimation in precursor species (such as CO or $NO_x$), a lack of long-range transport, an underestimation of ozone production in air masses during long-range transport to the Arctic, or a combination thereof.*

Line 416, if the Wespes et al. study mentions stratospheric influence as a major driver in tropospheric concentrations of O3, why is there no further discussion of which models in the current study simulate the stratosphere and how this may or may not influence errors in the partial column model estimates when compared to FTIR measurements?

We have added a column to Table 3 to indicate the level of stratospheric chemistry for each model.

Further text was added in Section 4.3 to support this:

- *The model-FTIR comparisons reveal that the spatial resolution and inclusion of stratospheric chemistry in the models does not necessarily improve results (refer to Table 3 for horizontal resolution and stratospheric chemistry). For example, WRF-Chem, EMEP MSC-W, and GEM-MACH show a low $R^2$ and higher NRMSE (varying between sites and models), although contributing to this for WRF-Chem and GEM-MACH could be the limited number of analysis years (two and one, respectively). These air-quality focused models have detailed chemistry and were run at higher spatial resolutions, whereas for example CMAM, a climate-focused model, has a coarser resolution with simplified tropospheric chemistry and demonstrates larger $R^2$ and smaller mean percent differences (Fig. 13). However, when considering the stratosphere, CMAM, which*

*includes comprehensive stratospheric chemistry, has comparable metrics in Fig. 13 to DEHM, which uses prescribed climatologies for the stratosphere. Similarly, Whaley et al. (2022) stated that the degree of stratospheric chemistry in the models did not reveal a consistent benefit or handicap when comparing the models with surface measurements.*

Line 441-444, it should also be mentioned that WRF-Chem and GEM-MACH do not have the same temporal coverage as the other models.

Added the following text to Section 4.3 to reiterate:

- *For example, WRF-Chem, EMEP MSC-W, and GEM-MACH show a low $R^2$ and higher NRMSE (varying between sites and models), although contributing to this for WRF-Chem and GEM-MACH could be the limited number of analysis years (two and one, respectively).*

Line 446-447, the statement, "although again are largely underpredicted" needs more context. Do you mean to say, "models largely underpredict FTIR measurements"?

Changed text to suggestion.

Line 457-458, drawing this conclusion in relation to the Wespes et al. study seems a bit tenuous because they only compared to one model and it was not one of the models considered in the current study.

Modified text in Section 4.3:

- *Results here are similar to those presented in Wespes et al. (2012), where across all the locations and models, 24 of the 55 model-measurement mean percent differences were within ±15% (see Table 4).*

Minor editing suggestions:

Do Figures 5, 10, and 15 need to have legends to indicate which line is 1:1 and which is the linear fit? I think Figures 7, 12, and 17 also need legends indicating FTIR data and model data points

The 1:1 line and linear fit are described in the figure caption. Previous iterations had more detailed legends, however these were reduced to allow for larger text of the current legend, and avoid additional clutter on the plots.

A legend has been added to the MMM plots.

Line 178, change "FITR," to "FTIR retrieval," or "FTIR retrieved partial column," Line 282, change "provided below" to "provided in Fig. 8-10"

Changed as per suggestion.

Line 299, missing period between "2021)" and "Further". Change "comparison of" to "correlations between"

Changed as per suggestion.

Line 300, change "1:1 comparison" to "1:1 correlation"

Changed as per suggestion.

Line 326, change "FTIR comparisons" to "FTIR measurements" or "FTIR retrievals" Line 330, suggest mentioning that this is eight pairs out of 36.

Added as per suggestion.

Line 366, change "emission fluxes" to "anthropogenic emissions" or "anthropogenic fluxes"

Changed as per suggestion.

Line 388 and Line 391, change "show" to "shows" and "reduce" to "reduces", respectively. Go through the paper and make sure verb tenses when referring to a single figure are singular, the verb should fit the subject outside the parentheses.

Edits made.